# RELATION EDITING FOR LARGE LANGUAGE MODELS

## ABSTRACT

Knowledge editing is a critical technique for the routine updating and mainte-
nance of LLMs. Existing research predominantly assumes changes only to the
object within subject-relation-object triples, with minimal exploration into tech-
niques for editing the relation. We term this task Relation Editing (distinct from
the established "Object Editing" paradigm). We first construct a dedicated rela-
tion editing dataset and benchmark existing algorithms, revealing a critical flaw:
even with successful edits, prominent methods suffer from the persistent retention
of outdated information, with rates reaching as high as 98.20%. Editing failures
stem primarily from two sources: the persistent retention of outdated relationships
and the presence of challenging editing samples. To address the first issue, we
propose a novel relation editing framework called Forgetting-and-Editing (`FE`).
We theoretically show that existing forgetting methods (i.e., model unlearning)
are unsuitable for this purpose and, to this end, introduce a new target assign-
ment strategy within our framework. To mitigate the second challenge, we intro-
duce a self-paced learning strategy, instantiated in a new algorithm named self-
paced AlphaEdit (`SPaEdit`). We conduct extensive experiments on our com-
piled relation-editing dataset and established object-editing benchmarks. Results
demonstrate that our proposed relation editing strategy achieves satisfactory per-
formance on the relation editing task. In addition, SPaEdit outperforms existing
SOTA methods on object-editing benchmarks. Our research also suggests further
study is warranted in relation editing, particularly on forgetting existing relations.

## 1 INTRODUCTION

Knowledge editing has emerged as a critical technique for precisely modifying factual associations
within LLMs without costly full retraining (Cao et al., 2021). This capability addresses the fun-
damental challenge posed by the static nature of LLMs, providing an efficient mechanism to up-
date their knowledge base with new facts or correct existing inaccuracies. The field has converged
on two distinct architectural strategies: weight-space editors that surgically alter transformer pa-
rameters (e.g., MEMIT's layer-wise scaling (Meng et al., 2023)), versus non-invasive approaches
employing external memory or prompt-based adaptation (e.g., MELO's dynamic LoRA (Yu et al.,
2024)). While effective in isolation, these methods face inherent trade-offs between edit precision
and knowledge retention stability. The recent breakthrough AlphaEdit (Fang et al., 2025) introduces
a novel null-space constrained approach that theoretically guarantees knowledge preservation while
enabling precise edits.

A knowledge triple takes the form $(s, r, o)$, for subject $s$, relation $r$, and object $o$. Most current
research on knowledge editing focuses on changing the object $((s, r, o) \rightarrow (s, r, o^*))$ (Wang et al.,
2024b), but pays little attention to changing the relation $((s, r, o) \rightarrow (s, r^*, o))$, even though such
updates are common in practice. For instance, changing "Zinedine Zidane is a player for Real
Madrid" to "Zinedine Zidane is a coach of Real Madrid" means updating the relation while keeping
the subject and object the same. This is a frequent type of change that existing methods overlook.
We call editing that targets relation changes "Relation Editing". In contrast, standard knowledge
editing is called "Object Editing". The easiest way to handle relation editing is to simply give
the new triple $(s, r^*, o)$ to current object-editing methods. If successful, separate research would
be unnecessary. To test this idea, we created a relation-editing dataset named ReEditBench from
available object-editing benchmarks. We then evaluated popular object-editing techniques including

ROME (Meng et al., 2022), MEMIT (Meng et al., 2023), and AlphaEdit (Fang et al., 2025). All methods performed poorly: first, although models learn the new triple $(s, r^*, o)$, they still strongly recall the old one (e.g., models edited with AlphaEdit keep 98.20% of the original knowledge). Second, these methods perform especially poorly on hard-to-edit relations. Our initial analysis shows that the editing success rate decreases as the difference grows between the model's knowledge of $(s, r^*)$ and the object $o$.

To bridge the gap, we propose a Forgeting-and-Editing (FE) strategy that enables models to learn new relations while forgetting old ones, thereby allowing existing object-editing algorithms to be adapted for relation-editing tasks. However, our theoretical analysis shows that current model unlearning strategies are inapplicable for direct use. Therefore, we propose a new target assignment scheme for old relation forgetting. For the hard relation editing problem, we draw inspiration from the classical self-paced learning concept (Kumar et al., 2010) and propose a self-paced knowledge editing algorithm called Self-paced AlphaEdit (SPaEdit). This method first learns easier samples based on the difficulty levels of knowledge tuples, then progressively incorporates more challenging ones for iterative optimization, ultimately selecting the optimal solution through validation.

Tests on our constructed relation-editing dataset demonstrate that our FE strategy significantly enhances the performance of object-editing methods beyond their standalone application. Specifically, on the Success metric, the FE strategy led to an average performance improvement of 10.07%, with a peak improvement of 34.49%. Notably, combining the FE strategy with the proposed SPaEdit yielded the best relation-editing performance. We also directly applied SPaEdit to existing object-editing benchmark datasets (Levy et al., 2017; Meng et al., 2022) and found that it outperformed the representative SOTA methods, including AlphaEdit. Additionally, a series of ablation experiments and sensitivity analyses consistently demonstrated the superiority of the proposed FE strategy and SPaEdit method.

## 2 PROBLEM DESCRIPTION AND ANALYSIS

### 2.1 PROBLEM DESCRIPTION

Knowledge editing aims to update factual triples stored in LLMs through single or sequential edits (Wang et al., 2024b). Unlike existing knowledge editing which modifies the object $o$ in a fact tuple $(s, r, o)$ (referred to object editing in this study), relation editing alters the relation $r$ rather than object $o$, resulting in a new tuple $(s, r^*, o)$[1]. In the locate-then-edit paradigm (Zhang et al., 2025; Pan et al., 2025), each edit applies a perturbation $\boldsymbol{\Delta}$ to the model parameters $\mathbf{W} \in \mathbb{R}^{d_1 \times d_0}$, where $d_0$ and $d_1$ denotes the dimensions of the FFN's intermediate and output layers. Specifically, for updating $h$ relation facts, let $\mathbf{K}_1 = [\boldsymbol{k}_1 \mid \boldsymbol{k}_2 \mid \cdots \mid \boldsymbol{k}_h] \in \mathbb{R}^{d_0 \times h}$ and $\mathbf{K}_1' = [\boldsymbol{k}_1' \mid \boldsymbol{k}_2' \mid \cdots \mid \boldsymbol{k}_h'] \in \mathbb{R}^{d_0 \times h}$ be the keys for the raw and the updated subject-relation pairs, respectively. The value matrix $\mathbf{V}_1 = [\boldsymbol{v}_1 \mid \boldsymbol{v}_2 \mid \cdots \mid \boldsymbol{v}_h] \in \mathbb{R}^{d_1 \times h}$ remains unchanged. To our knowledge, there is currently no dedicated research work focusing on relation editing. While RaKE (Wei et al., 2023) briefly touches upon it, the task itself remains largely overlooked by the research community.

Directly applying the solution approach of object editing, the following optimization objective minimizing the error for updated relations while preserving existing knowledge is obtained:

$$\boldsymbol{\Delta} = \arg\min_{\tilde{\boldsymbol{\Delta}}} \|(\mathbf{W} + \tilde{\boldsymbol{\Delta}})\mathbf{K}_1' - \mathbf{V}_1\|_F^2. \tag{1}$$

At first glance, existing object editing methods such as MEMIT and AlphaEdit can be directly applied in principle, essentially solving the problem based on variations of Eqn. 1. To verify whether object editing methods can be directly applied to relations, we constructed ReEditBench, a new benchmark for relation editing. It was built through a rigorous four-stage pipeline, with full details provided in Appendix A.1. First, we curate initial high-quality facts

Table 1: Statistics of ReEditBench.

| Data Source | New Relation | Conditional Relation | Total |
|---|---|---|---|
| ZsRE | 2,000 | 2,000 | 4,000 |
| Wikidata | 1,700 | 2,218 | 3,918 |
| **Total** | **3,700** | **4,218** | **7,918** |

---

[1]It should be noted that this study does not consider the scenario in which the user specifies a new object for the original subject and object; instead, we assume that no relevant information is provided by the user.

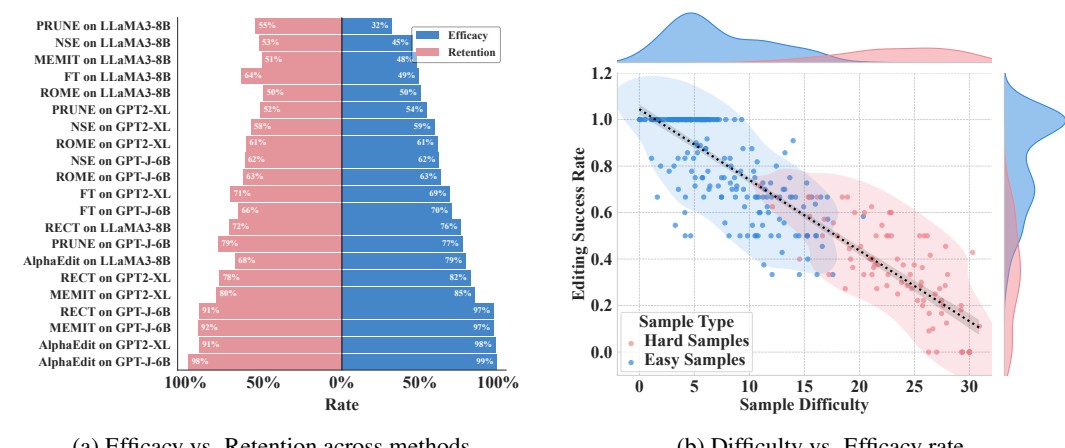

(a) Efficacy vs. Retention across methods

(b) Difficulty vs. Efficacy rate

Figure 1: Analysis of key challenges in relation editing. (a) The bar chart compares editing efficacy (blue) with Retention of the original fact (pink), showing that old knowledge persists. (b) The scatter plot shows a strong negative correlation between sample difficulty and Efficacy rate, indicating performance decay on challenging samples.

from established knowledge-intensive benchmarks, primarily ZsRE (Levy et al., 2017) and Wikidata (Vrandečić & Krötzsch, 2014). Second, a generator LLM (DeepSeekV3 (Liu et al., 2024a)) automatically reframes these facts into relation-editing tasks, guided by two distinct patterns: new relation and conditional relation. Third, all candidates are filtered automatically for structural integrity and semantic plausibility using scripts and a verifier LLM (DeepseekR1 (Guo et al., 2025)). Finally, to quantify the dataset's quality, we manually validated a 30% random sample and found that 98.5% of the instances were valid, confirming the high quality of our generation pipeline. This process yields 7,918 high-quality editing instances, with a detailed breakdown provided in Table 1.

## 2.2 RESULTS DIRECTLY WITH OBJECT EDITING

To investigate the direct applicability of existing object editing methods to the relation editing task, we conducted a series of empirical evaluations. Our analysis of the results reveals two distinct patterns. First, as shown in Fig. 1(a), while most editing methods achieve high success rates in acquiring the new knowledge (blue bars), they concurrently retain the original, conflicting knowledge at exceptionally high rates (pink bars). This creates a near-symmetrical visual pattern; for instance, AlphaEdit on GPT-J (Wang & Komatsuzaki, 2021) pairs a success rate of approximately 99% with a retention rate of 98%. Second, Fig. 1(b) demonstrates a strong negative correlation between the editing success rate and sample difficulty (measured as the magnitude of the initial residual, $\|v_i - \mathbf{W}k_i'\|_2^2$). The data clearly forms two distinct clusters, with "easy samples" (blue) concentrated in a high-success region and "hard samples" (pink) occupying a low-success region.

To delve deeper into the intrinsic nature of these "hard samples", we further investigated the role of semantic similarity between the original relation $r$ and the target $r^*$. Our analysis (detailed in Appendix C.5) uncovers an intriguing trade-off: relations with high semantic proximity are easier to learn but significantly harder to forget, whereas semantically divergent ones show the opposite trend. However, while semantic analysis offers valuable explanatory insights, we find that the computational residual remains the superior metric for quantifying difficulty in practice. This is because semantic similarity captures only the linguistic dimension of difficulty. In contrast, the computational residual acts as a **holistic proxy** that aggregates all latent influencing factors—including semantics, knowledge frequency, and structural complexity. It provides a **direct, quantifiable signal** of the actual optimization barrier the model faces, making it a more robust and computationally efficient standard for our curriculum learning than semantic metrics alone.

These observations point to two fundamental and distinct limitations of current approaches in relation editing. First, the near-symmetrical pattern of success and retention indicates that these methods perform an additive operation rather than a corrective overwrite, resulting in the problematic coexistence of both new and old knowledge. Second, they consistently fail on high-difficulty editing

samples. We therefore conclude that existing methods are ill-suited for this task, as they fail to properly erase outdated information and lack the efficacy required for challenging edits.

# 3 METHODOLOGY

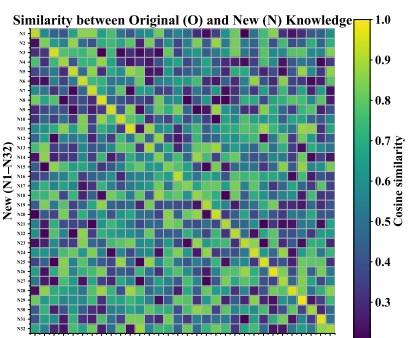

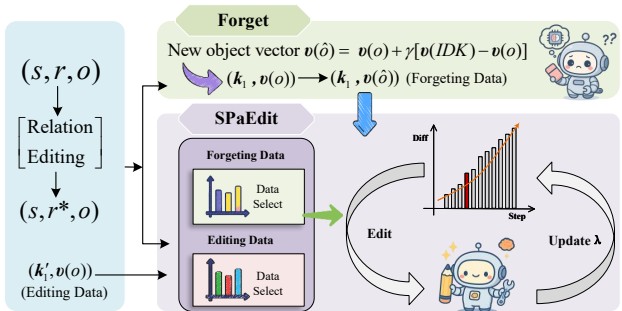

Figure 2: Similarity heatmap between original and new relation keys.

Figure 3: Overview of our proposed framework for relation editing, combining a novel forgetting-and-editing (FE) strategy with a Self-paced AlphaEdit (SPaEdit) algorithm.

The empirical findings from the previous section indicate that the retention of the old tuple $(s, r, o)$ constitutes the primary failure point when existing object editing methods are directly applied to relation editing. Hence, a highly intuitive approach involves first forgetting the old tuple and then incorporating new knowledge. This strategy has in fact been mentioned in several studies on object editing (Ni et al., 2024; Jung et al., 2025). The present study also adopts this general strategy; however, we will begin by introducing our theoretical analysis, which demonstrates that existing model unlearning strategies are largely ineffective when directly applied to old relation forgetting. Based on this analysis, we further propose a novel unlearning method. To mitigate degradation on difficult cases, we further develop Self-Paced AlphaEdit (SPaEdit), which performs editing under an easy-to-hard curriculum inspired by self-paced learning.

## 3.1 THEORETICAL INVESTIGATION

Conventionally, LLM unlearning methods (Yao et al., 2024; Wang et al., 2025) set the prediction target for data to be forgotten either to "I don't know" (IDK) or to a random response. However, as will be explained mathematically, both strategies are ill-suited for old relation forgetting under linear regression-based editing methods such as AlphaEdit and MEMIT.

Following prior studies (Meng et al., 2022; Fang et al., 2025) that formulate knowledge editing as a linear regression task, we model the forgetting of old relations within this framework to facilitate understanding. We consider a linear homogeneous regression problem ($y = \boldsymbol{w}^\top \boldsymbol{x}$) with a training set $\mathbb{D} = \{(\boldsymbol{x}_i, y_i)\}$ for $i = 1, \ldots, N$, assuming $y_i \in [0, 1]$. The set $\mathbb{D}$ is split evenly into $\mathbb{D}_g$ (normal data) and $\mathbb{D}_b$ (forgetting data), for which we examine two cases: (1) fixing each $y_i$ in $\mathbb{D}_b$ to a constant $\hat{y}$ (simulating all objects changed to IDK), and (2) setting each $y_i$ in $\mathbb{D}_b$ to a random value (simulating random object assignments). By minimizing MSE, we obtain:

$$\boldsymbol{w}^* = (\mathbf{X}^\top \mathbf{X})^{-1}(\mathbf{X}_g^\top \boldsymbol{y}_g + \mathbf{X}_b^\top \boldsymbol{y}_b), \quad (2)$$

where $\mathbf{X} \in \mathbb{R}^{N \times d}$ is the feature matrix and $\boldsymbol{y} \in \mathbb{R}^N$ is the label vector, $\mathbf{X}_g^\top \boldsymbol{y}_g$ represents the signal from the normal data, and $\mathbf{X}_b^\top \boldsymbol{y}_b$ represents the term from the forgetting data. The subsequent analysis will focus on how this term distorts the optimal solution $\boldsymbol{w}^*$. Let $\boldsymbol{w}_g^*$ be the solution achieved by only applying the normal data $\mathbb{D}_g$. In the first case, with mathematical deduction, Eqn. 2 yields:

$$\boldsymbol{w}_{\text{const}}^* = (\mathbf{X}^\top \mathbf{X})^{-1}(\mathbf{X}_g^\top \boldsymbol{y}_g + \frac{\hat{y}N}{2}\boldsymbol{u}) = \boldsymbol{w}_g^* + \frac{\hat{y}N}{2}(\mathbf{X}^\top \mathbf{X})^{-1}\boldsymbol{u}, \quad (3)$$

where $\boldsymbol{u} = \frac{1}{|\mathbb{D}_b|} \sum_{i \in \mathbb{D}_b} \boldsymbol{x}_i$. This implies that all predictions will be systematically distorted toward $\hat{y}$ (our theoretical conclusion closely aligns with a recent empirical observation in LLM unlearning (Yuan et al., 2025): when the target token is identical, the output probability of the target

token (i.e., IDK in the case) increases for both unlearning and normal inputs), and the degree of distortion depends on the correlation between the new input and $(\mathbf{X}^\top \mathbf{X})^{-1}\boldsymbol{u}$.

In the second case, with mathematical deduction, we can obtain the expected solution as follows:

$$\mathbb{E}[\boldsymbol{w}^*_{\text{rand}}] = (\mathbf{X}^\top \mathbf{X})^{-1}(\mathbf{X}_g^\top \boldsymbol{y}_g + \mathbb{E}[\mathbf{X}_b^\top \boldsymbol{y}_b]) = \boldsymbol{w}^*_g + (\mathbf{X}^\top \mathbf{X})^{-1}(0.5|\mathbb{D}_b|\mathbb{E}[\boldsymbol{x}]). \tag{4}$$

Similar to the first case, the random noise introduces a systematic bias in expectation in both normal and unlearning samples, pulling the solution toward a direction determined by the irrelevant feature mean, which forces the predicted values to skew toward 0.5 (average response in LLMs).

Our theoretical analysis shows that when using current model editing methods to forget old relations, standard unlearning strategies cause normal knowledge to become systematically distorted.

## 3.2 KNOWLEDGE FORGETTING VIA TARGET SMOOTHING

Theoretical analysis shows conventional target assignment strategies for LLM unlearning are ineffective for knowledge forgetting. Thus, the key to our approach is determining a suitable object, denoted as $\hat{o}$, for the triplet $(s, r, o)$ to be unlearned. This $\hat{o}$ should neither be uniform across all samples nor randomly assigned. Furthermore, statistical analysis of the vector representations of $(s, r)$ and $(s, r^*)$ reveals very high similarity across the dataset, as shown in Fig. 2. Given the properties of linear regression, an additional guideline for selecting $\hat{o}$ is that the difference between $\boldsymbol{v}(\hat{o})$ and $\boldsymbol{v}(o)$ is not large; otherwise, a significant disparity between these values, combined with the high similarity of $(s, r)$ and $(s, r^*)$, would make the optimization problem significantly harder to solve.

Based on these three considerations, we directly generate the vector for $\hat{o}$ in the following manner:

$$\boldsymbol{v}(\hat{o}) = \boldsymbol{v}(o) + \gamma[\boldsymbol{v}(IDK) - \boldsymbol{v}(o)], \gamma \in (0, 1). \tag{5}$$

Our assignment strategy, controlled by the hyperparameter $\gamma$, is designed to satisfy three criteria: nonconstant assignment, nonrandom assignment, and target vector proximity. As our analysis shows (Appendix B.1), compared with fixed constant targets (e.g., "I don't know") or random responses, it suppresses systematic bias, improves edit success, and reduces retention, while inducing smaller perturbations to normal knowledge and yielding more stable optimization. Nevertheless, our experiments reveal residual retention on some models, indicating that relation editing poses substantive new challenges and merits dedicated investigation.

## 3.3 THE PROPOSED FORGETTING-AND-EDITING STRATEGY

Building upon the target smoothing derived in Section 3.2, we propose the Forgetting-and-Editing (FE) strategy. This strategy serves as a comprehensive framework that integrates the "unlearning" of outdated relations with the injection of new knowledge. Fig. 3 provides an illustrative overview of this pipeline.

To achieve this dual objective, we construct a composite editing task. For a given batch of $N$ relation editing samples, where the $i$-th sample involves changing the relation from $(s_i, r_i, o_i)$ to $(s_i, r_i^*, o_i)$, the procedure operates in two stages combined into a single optimization step:

- **Stage 1: Constructing the Forgetting Pairs.** We first compute the interpolated target $\boldsymbol{v}(\hat{o}_i)$ using Eqn. 5. We then form the forgetting pair $(\boldsymbol{k}_i, \boldsymbol{v}(\hat{o}_i))$, where $\boldsymbol{k}_i$ is the key vector corresponding to the original subject-relation $(s_i, r_i)$. This pair instructs the model to shift the representation of the old relation toward a neutral state, effectively suppressing the activation of the outdated knowledge.

- **Stage 2: Constructing the Editing Pairs.** Simultaneously, we construct the standard editing pair $(\boldsymbol{k}_i', \boldsymbol{v}(o_i))$, where $\boldsymbol{k}_i'$ is the key vector for the new subject-relation $(s_i, r_i^*)$, and $\boldsymbol{v}(o_i)$ is the target value of the object. This pair ensures the model accurately captures the new relational association.

**Joint Optimization.** Finally, both the forgetting pairs and the editing pairs are concatenated to form the full training set for the current batch:

$$\mathcal{D}_{\text{total}} = \bigcup_{i=1}^{N} \{(\boldsymbol{k}_i, \boldsymbol{v}(\hat{o}_i)), (\boldsymbol{k}_i', \boldsymbol{v}(o_i))\}. \tag{6}$$

---

**Algorithm 1:** SPaEdit

---

**Input:** $\mathbf{K}_1 \in \mathbb{R}^{d \times n}$, $\mathbf{V}_1 \in \mathbb{R}^{d \times n}$, $\mathbf{W} \in \mathbb{R}^{d \times d}$, $\mathbf{P} \in \mathbb{R}^{d \times d}$, $\mathbf{K}_p \in \mathbb{R}^{d \times m}$, $\alpha, \beta, \mu, \lambda_0, T$
**Output:** sequence of edited matrices $\{\mathbf{W}^{(t)}\}_{t=1}^{T}$

$\lambda \leftarrow \lambda_0$;
**for** $t = 1$ **to** $T$ **do**
    **for** $i = 1$ **to** $n$ **do**
        $\ell_i \leftarrow \left\| (\mathbf{W} + \mathbf{\Delta P}) \, \boldsymbol{k}_i - \boldsymbol{v}_i \right\|_2^2$;
        $z_i \leftarrow \mathbb{1}(\ell_i \leq \lambda)$
    $\mathbf{Z} \leftarrow \mathrm{diag}(z_1, \ldots, z_n), \mathbf{R} \leftarrow \mathbf{V}_1 - \mathbf{W}\mathbf{K}_1$;
    $\mathbf{\Delta P} \leftarrow \mathbf{R}\mathbf{Z}\mathbf{K}_1^\top \mathbf{P} \left( \mathbf{K}_1 \mathbf{Z} \mathbf{K}_1^\top \mathbf{P} + \beta \mathbf{K}_p \mathbf{K}_p^\top \mathbf{P} + \alpha \mathbf{I} \right)^{-1}$;
    $\mathbf{W}^{(t)} \leftarrow \mathbf{W} + \mathbf{\Delta P}, \mathbf{W} \leftarrow \mathbf{W}^{(t)}, \lambda \leftarrow \mu\lambda$;
**return** $\{\mathbf{W}^{(t)}\}_{t=1}^{T}$;

---

This combined dataset $\mathcal{D}_{\text{total}}$ is then fed into the base editor (e.g., AlphaEdit or our SPaEdit). By jointly optimizing for both objectives, the algorithm updates the weights to simultaneously unlearn the old relation and acquire the new one, resolving the conflict inherent in relation editing.

## 3.4 IMPROVEMENT VIA SELF-PACED LEARNING

As demonstrated in our experimental analysis (Section 2.2), some knowledge edits are significantly more challenging than others. This motivates us to incorporate self-paced learning (SPL), an easy-to-hard curriculum, into the knowledge editing process. We integrate this strategy with SOTA AlphaEdit (Fang et al., 2025). To formulate our approach, we re-examine the original objective of AlphaEdit, which seeks an optimal perturbation $\mathbf{\Delta}$:

$$\arg\min_{\mathbf{\Delta}} \|(\mathbf{W} + \mathbf{\Delta P})\mathbf{K}_1 - \mathbf{V}_1\|_F^2 + \alpha\|\mathbf{\Delta P}\|_F^2 + \beta\|\mathbf{\Delta P}\mathbf{K}_p\|_F^2. \tag{7}$$

Here, $\mathbf{K}_1$ and $\mathbf{V}_1$ are the keys and values of the facts to be edited. The objective incorporates two regularizers: $\alpha$-term constrains the update within the null space of AlphaEdit via the projector $\mathbf{P}$, and $\beta$-term penalizes interference with previously edited knowledge $\mathbf{K}_p$. The baseline objective uses uniform instance weighting and ignores difficulty. We therefore recast editing as SPL, introducing binary selectors $z_i \in \{0, 1\}$ to build an adaptive curriculum, leading to the following objective:

$$\min_{\mathbf{\Delta}, \boldsymbol{z}} \mathcal{J}(\mathbf{\Delta}, \boldsymbol{z}; \lambda) = \sum_{i=1}^{n} z_i \ell_i(\mathbf{\Delta}) + \alpha\|\mathbf{\Delta P}\|_F^2 + \beta\|\mathbf{\Delta P}\mathbf{K}_p\|_F^2 - \lambda\sum_{i=1}^{n} z_i. \tag{8}$$

Here, $z_i = 1$ indicates that the $i$-th sample is included in the editing approach. $\lambda > 0$ is the pace parameter that controls the curriculum's difficulty. The sample-wise loss is the squared error for the $i$-th edit: $\ell_i(\mathbf{\Delta}) = \|(\mathbf{W} + \mathbf{\Delta P})\boldsymbol{k}_i - \boldsymbol{v}_i\|_2^2 = \|\mathbf{\Delta P}\boldsymbol{k}_i - \boldsymbol{r}_i\|_2^2$, where $\boldsymbol{r}_i = \boldsymbol{v}_i - \mathbf{W}\boldsymbol{k}_i$ is the residual for the $i$-th sample. We optimize Eqn. 8 via alternating minimization between $\mathbf{\Delta}$ and $\boldsymbol{z}$.

With $\boldsymbol{z}$ fixed, the problem reduces to a regularized least-squares objective over the subset of "easy" samples. Let $\bar{\mathbf{Z}} = \mathbf{Z}^{1/2} = \mathrm{diag}(\boldsymbol{z})$. We solve for $\mathbf{\Delta}$:

$$\min_{\mathbf{\Delta}} \left\| (\mathbf{\Delta P}\mathbf{K}_1 - (\mathbf{V}_1 - \mathbf{W}\mathbf{K}_1)) \mathbf{Z}^{1/2} \right\|_F^2 + \alpha\|\mathbf{\Delta P}\|_F^2 + \beta\|\mathbf{\Delta P}\mathbf{K}_p\|_F^2. \tag{9}$$

This is a convex problem whose closed-form solution for the update $\mathbf{\Delta}_{\text{SPaEdit}} = \mathbf{\Delta P}$ is:

$$\mathbf{\Delta}_{\text{SPaEdit}} = (\mathbf{V}_1 - \mathbf{W}\mathbf{K}_1)\mathbf{Z}\mathbf{K}_1^\top \mathbf{P} \left( \mathbf{K}_1 \mathbf{Z} \mathbf{K}_1^\top \mathbf{P} + \beta\mathbf{K}_p\mathbf{K}_p^\top \mathbf{P} + \alpha\mathbf{I} \right)^{-1}. \tag{10}$$

With $\mathbf{\Delta}$ fixed, we determine the optimal sample selection $z^*$ for each sample for the next iteration. This step realizes an easy-to-hard curriculum by adjusting the difficulty threshold $\lambda$ to progressively incorporate more challenging samples:

$$z_i^*(\lambda) = \begin{cases} 1, & \text{if } \ell_i(\mathbf{\Delta}) < \lambda \\ 0, & \text{otherwise} \end{cases}. \tag{11}$$

This two-step process is iterated, with $\lambda$ gradually increasing to incorporate more difficult samples over time. Once these optimization iterations conclude, we obtain a series of $\mathbf{W}^{(t)}$. We use a validation set for model selection, stopping the iterative process when the validation loss plateaus. Details on the validation set's construction are provided in the Appendix A.1.2. The entire algorithm is called SPaEdit as shown in Algorithm 1. Notably, compared to AlphaEdit, our approach incurs minimal structural overhead, requiring only the introduction of the diagonal matrix $\mathbf{Z}$ to dynamically control the optimization order of the samples.

## 4 EXPERIMENTS AND ANALYSIS

### 4.1 EXPERIMENTAL SETUP

**Base LLMs & Baseline Methods.** We evaluate knowledge editing across three representative LLMs: LLaMA3 (8B) (Meta, 2024), GPT-J (6B), and GPT2-XL (1.5B) (Radford et al., 2019). Seven parametric editing methods are compared: **MEMIT** (Meng et al., 2023), **RECT** (Gu et al., 2024), **NSE** (Jiang et al., 2024), **ROME** (Meng et al., 2022), **Fine-Tuning (FT)** (Zhu et al., 2020), **PRUNE** (Ma et al., 2025), and **AlphaEdit** (Fang et al., 2025). Detailed descriptions of these baseline methods are provided in Appendix A.2. We have ensured the accessibility of our work; all experiments can be replicated from start to finish on a single, commonly available NVIDIA L40S(48G).

**Metrics.** Our evaluation metrics are chosen based on the specific task. For Relation Editing, we focus on **Success** (holistic replacement) and **Retention** (forgetting), alongside **Efficacy** and **Generalization**. For the standard Object Editing task, we use the canonical set of **Efficacy**, **Generalization**, and **Specificity** for ZsRE, and expand this set with **Fluency** and **Consistency** for the generative CounterFact benchmark. Detailed definitions are available in Appendix A.3.

**Datasets.** We evaluate our methods on **ReEditBench**, our novel benchmark constructed for the Relation Editing task. To select the optimal model from our iterative algorithm, we use a validation set to minimize a weighted loss that balances three key objectives: forgetting the old fact, learning the new one, and generalization, with respective weights of 0.4, 0.4, and 0.2. The detailed construction of this validation set is described in Appendix A.1.2. To further assess the universality and generalization capabilities of our proposed SPaEdit algorithm, we also evaluate its performance on the two object editing benchmarks: **ZsRE** (Levy et al., 2017) and **CounterFact** (Meng et al., 2022). ReEditBench will be available once acceptance and codes are in the attachment.

### 4.2 EFFICACY OF THE FORGETTING-AND-EDITING STRATEGY ON RELATION EDITING

**Setup.** The results are shown under the standard sequential editing setting, where a total of 2000 samples were randomly drawn from the dataset for updates, with each edit consisting of 100 samples. For the relevant experimental runs, the forgetting parameter $\lambda$ was set to 0.6, and the update regularization coefficients $\alpha$ and $\beta$ were set to 10 and 1, respectively.

**Results.** In our evaluation framework, we prioritize Success and Retention, since relation editing must ensure that new knowledge reliably replaces old. Table 2 shows that our Forgetting-and-Editing (FE) strategy consistently improves performance across methods and models: by markedly lowering Retention (up to **40.85**% reduction), it raises Success by up to **34.49**%, while typically improving Efficacy and Generalization. The seemingly high retention of some baselines is misleading, stemming from low editing success that fails to challenge original knowledge and thus yields deceptively low interference. In contrast, our strategy genuinely alters the knowledge relationship by pairing high editing success with effective forgetting of outdated facts. We further show that replacing fixed targets with our interpolation-based assignment yields substantially better unlearning than assigning "I don't Know" or random answers (see Appendix C.1), confirming the effectiveness of our design. Nevertheless, Retention remains nontrivial in absolute terms, often around 50% in difficult settings, indicating that fully clean forgetting is still unsolved and merits further study.

**Analysis of the Forgetting Strategy.** We present an empirical comparison of four unlearning strategies in Fig. 4, with results that clearly validate our theoretical analysis from Section 3.1. The experiments show that conventional unlearning strategies, which set the prediction target for outdated knowledge to either a generic "I don't know" response or a random value, are ineffective at reducing the model's knowledge retention. On the GPT-J model, for instance, these approaches yield retention

Table 2: Main Results on the Relational Editing Task

| LLMs | Method | Success↑ | | Retention↓ | | Efficacy↑ | | Generalization↑ | |
|---|---|---|---|---|---|---|---|---|---|
| | | Original | +FE | Original | +FE | Original | +FE | Original | +FE |
| LLaMA3 | MEMIT | 33.77 | 68.26 (+34.49) | 51.70 | 58.82 (-7.12) | 48.43 | 70.93 (+22.50) | 49.09 | 67.00 (+17.91) |
| | RECT | **59.41** | 66.83 (+7.42) | 72.78 | 59.45 (+13.33) | 66.78 | 69.70 (+2.92) | 54.63 | 58.96 (+4.33) |
| | NSE | 43.20 | 54.30 (+11.10) | 53.73 | 52.24 (+1.49) | 45.00 | 58.53 (+13.53) | 59.26 | 58.55 (-0.71) |
| | ROME | 31.39 | 44.91 (+13.52) | 60.47 | 56.36 (+4.11) | 50.91 | 56.64 (+5.73) | 50.93 | 56.80 (+5.87) |
| | FT | 48.88 | 63.45 (+14.57) | 64.49 | 63.57 (+0.92) | 49.96 | 71.01 (+21.05) | 69.16 | 67.31 (-1.85) |
| | PRUNE | 29.40 | 29.81 (+0.41) | **44.68** | 30.46 (+14.22) | 44.04 | 34.25 (-9.79) | 43.86 | 42.97 (-0.89) |
| | AlphaEdit | 52.18 | 78.46 (+26.28) | 78.34 | 67.12 (+11.22) | 79.17 | 83.24 (+4.07) | **76.62** | 80.03 (+3.41) |
| | **SPaEdit(Ours)** | 54.45 | **81.71** (+27.26) | 68.56 | 62.77 (+5.79) | **83.23** | **87.37** (+4.14) | 75.88 | **81.14** (+5.26) |
| GPT2-XL | MEMIT | 56.31 | 57.79 (+1.48) | 80.26 | 57.21 (+23.05) | 85.23 | 84.67 (-0.56) | 80.68 | 85.21 (+4.51) |
| | RECT | 54.60 | 54.72 (+0.12) | 78.10 | 61.62 (+16.48) | 82.35 | 84.08 (+1.73) | 78.37 | 77.12 (-1.25) |
| | NSE | 45.00 | 45.45 (+0.45) | 58.53 | 58.24 (+0.29) | 59.26 | 59.99 (+0.73) | 58.55 | 59.43 (+0.88) |
| | ROME | 45.74 | 45.82 (+0.08) | 61.71 | 61.49 (+0.22) | 61.70 | 61.39 (-0.31) | 61.19 | 61.78 (+0.59) |
| | FT | 49.96 | 51.32 (+1.36) | 71.01 | 67.25 (+3.76) | 69.16 | 69.93 (+0.77) | 67.31 | 67.58 (+0.27) |
| | PRUNE | 37.88 | 38.04 (+0.16) | 52.62 | 39.14 (+13.48) | 54.49 | 55.71 (+1.22) | 52.99 | 52.60 (-0.39) |
| | AlphaEdit | **65.31** | 75.93 (+10.62) | 91.31 | 50.46 (+40.85) | 86.83 | 87.36 (+0.53) | 84.51 | 85.50 (+0.99) |
| | **SPaEdit(Ours)** | 62.00 | **83.93** (+21.93) | 68.55 | 48.78 (+19.77) | 85.93 | **88.46** (+2.53) | 87.36 | **87.50** (+0.14) |
| GPT-J | MEMIT | 72.55 | 82.36 (+9.81) | 92.98 | 71.94 (+21.04) | **87.14** | 87.80 (-0.76) | 84.69 | 84.89 (+0.20) |
| | RECT | 72.54 | 77.63 (+5.09) | 91.67 | 74.54 (+17.13) | 82.12 | 82.42 (+0.30) | 81.90 | 82.10 (+0.20) |
| | NSE | 45.65 | 45.95 (+0.30) | **62.13** | 61.12 (+1.01) | 62.03 | 60.94 (-1.09) | 61.52 | 61.63 (+0.11) |
| | ROME | 46.38 | 47.79 (+1.41) | 63.34 | **29.27** (+34.04) | 63.32 | 61.49 (-1.83) | 63.24 | 63.78 (+0.54) |
| | FT | 51.19 | 61.10 (+9.91) | 66.24 | 43.50 (+22.74) | 70.79 | 78.72 (+7.97) | 67.31 | 68.67 (+1.34) |
| | PRUNE | 55.71 | 63.05 (+7.34) | 79.12 | 59.87 (+19.25) | 77.25 | 77.00 (-0.25) | 75.41 | 76.62 (-1.21) |
| | AlphaEdit | 65.99 | 89.98 (+23.99) | 98.20 | 63.84 (+34.36) | 85.53 | 85.64 (+0.11) | 86.87 | 87.80 (+0.93) |
| | **SPaEdit(Ours)** | **78.46** | **91.02** (+12.56) | 88.24 | 59.84 (+28.40) | 75.93 | **88.08** (+12.15) | 87.36 | **88.58** (+1.22) |

rates as high as **77.2**% and **77.9**% respectively, which confirms that their inherent systematic biases impede effective forgetting. In contrast, our proposed strategy, which works by interpolating the value vector of the outdated fact towards a neutral state, performs exceptionally well and achieves the best trade-off between the success and retention rates across all tested models (LLaMA3, GPT2-XL, and GPT-J). Specifically, not only does our method rank among the highest in Success rate, but more critically, it consistently achieves the lowest Retention rate in all cases.

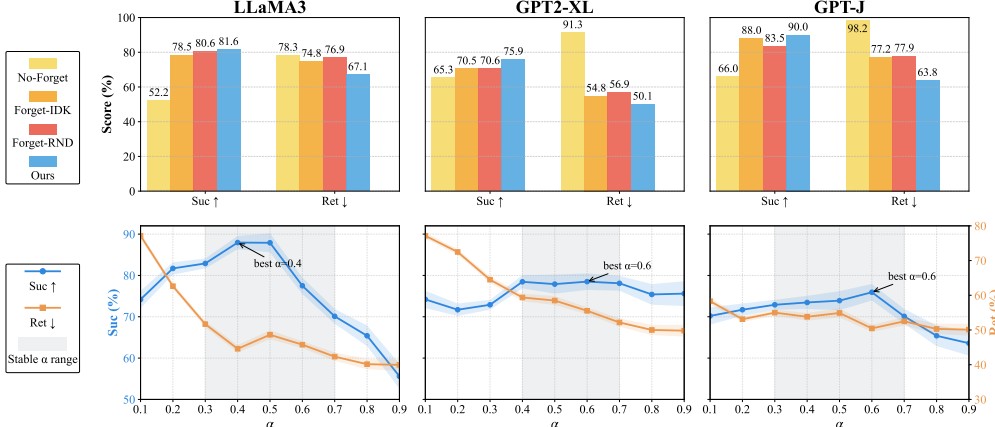

Figure 4: Ablation and Sensitivity Analysis of the Forgetting-and-Editing Strategy.

**Sensitivity Analysis on Hyperparameter** $\lambda$**.** Our sensitivity analysis for the interpolation factor $\lambda$, shown in Fig. 4, reveals a clear trade-off between forgetting and learning. While a larger $\lambda$ leads to more effective forgetting (a monotonic decrease in the Retention rate), it also produces a concave trajectory for the Success rate, which first increases and then decreases. We identify a broad optimal window, $\lambda \in [0.3, 0.7]$, where the Success rate is maximized without a significant compromise in forgetting. The existence of such a wide effective range underscores the robustness of our FE strategy and its low sensitivity to hyperparameter tuning, which is vital for practical deployment.

### 4.3 GENERALIZATION AND PERFORMANCE ON OBJECT EDITING BENCHMARKS

To assess generality, we use 100-example hard subsets from ZsRE and CounterFact. This section focuses on the ZsRE results, which demonstrate state-of-the-art performance. The complete results for CounterFact and an analysis of the sample difficulty distributions are available in Appendix C.2.

**Results on ZsRE.** As demonstrated in Table 3, SPaEdit consistently establishes a new state-of-the-art on the ZsRE benchmark across all tested models. Its commanding lead in Efficacy is particularly notable: achieving **92.32**% on LLaMA3 (a significant improvement over AlphaEdit's 81.87%) and a near-perfect **99.97**% on GPT-J. This superiority extends to Generalization, where SPaEdit achieves the top score of 89.89% on GPT2-XL, and also leads in Specificity on GPT-J with 28.61%. The experimental findings reveal that the hard sample subset poses a considerable challenge, causing notable performance degradation even for strong methods like AlphaEdit that rely on single-pass optimization. In stark contrast, SPaEdit not only withstands this challenge but excels by maintaining superior performance where other methods falter. This highlights the advantage of SPaEdit's strategic, staged learning process: it first builds a robust foundation on easier edits before progressively incorporating more chal-

Table 3: Object Editing Performance on ZsRE

| LLM | Method | Efficacy↑ | Generalization↑ | Specificity↑ |
|---|---|---|---|---|
| LLaMA3 | ROME | 31.87 | 32.4 | 32.26 |
| | MEMIT | 86.07 | 82.39 | 33.33 |
| | AlphaEdit | 81.87 | 78.11 | 33.03 |
| | SPaEdit | **92.32** | **82.6** | 32.11 |
| GPT2-XL | ROME | 15.87 | 16.98 | 7.74 |
| | MEMIT | 71.47 | 63.14 | 7.37 |
| | AlphaEdit | 92.17 | 82.68 | 7.72 |
| | SPaEdit | **98.96** | **89.89** | 7.23 |
| GPT-J | ROME | 23.69 | 27.9 | 24.12 |
| | MEMIT | 94.86 | 90.02 | 28.22 |
| | AlphaEdit | 96.26 | 90.46 | 28.15 |
| | SPaEdit | **99.97** | **91.3** | **28.61** |

lenging ones, thereby avoiding optimization pitfalls that arise when attempting to resolve high-residual errors simultaneously. Consequently, this approach not only provides an effective solution for relation editing but also establishes a new state-of-the-art on traditional object editing tasks. Furthermore, we report results on more comprehensive datasets in the Appendix C.3; despite the near-saturation of performance metrics on these benchmarks, our method still maintains a slight but consistent advantage over current field methods.

## 4.4 MECHANISTIC INSIGHT INTO SPAEDIT

To elucidate the mechanisms driving SPaEdit's advantage, we visualize its internal curriculum dynamics and resulting cost-benefit profile in Fig. 5.

**(a) Curriculum dynamics.** This figure traces how the sample-difficulty distribution evolves under our self-paced learning framework as parameters are updated. At the start (e.g., $(t = 1)$), the distribution is right-skewed toward high difficulty, indicating many hard samples. As training proceeds and parameters are optimized, proficiency increases and the mass shifts from the hard (right) to the easy (left) region. By later iterations (e.g., $(t = 13)$), the distribution is left-skewed, meaning most samples are easy. This progression demonstrates the effectiveness of the parameter updates.

**(b) Cost-benefit analysis.** The adaptive nature of SPaEdit is validated by its cost-benefit trade-off. On tasks with a low proportion of hard samples, SPaEdit incurs negligible overhead, matching the execution time of baselines while achieving superior efficacy. As task difficulty increases, it strategically invests modest additional computation time, which yields a substantial gain in editing success, in stark contrast to baselines whose performance degrades sharply. This favorable trade-off demonstrates that SPaEdit efficiently allocates resources, ensuring both robustness and high performance across a wide spectrum of difficulties.

## 5 RELATED WORK

**Parameter-Based Knowledge Editing.** Methods split into two families: meta-learning (KE (Cao et al., 2021), MEND (Mitchell et al., 2022)) and locate-then-edit, which identifies fact-related weights and applies a closed-form update (ROME (Meng et al., 2022), MEMIT (Meng et al., 2023)). Subsequent work increases granularity to neurons or heads (LoFiT (Yin et al., 2024), FiNE (Pan et al., 2025)), while AlphaEdit (Fang et al., 2025) improves safety and efficacy by projecting updates into a knowledge-preserving null space. Despite implementation differences, prior art largely defines editing as modifying the object $o$ in $(s, r, o)$; the relation $r$ has been systematically overlooked. We present the first systematic study of relation editing to fill this gap.

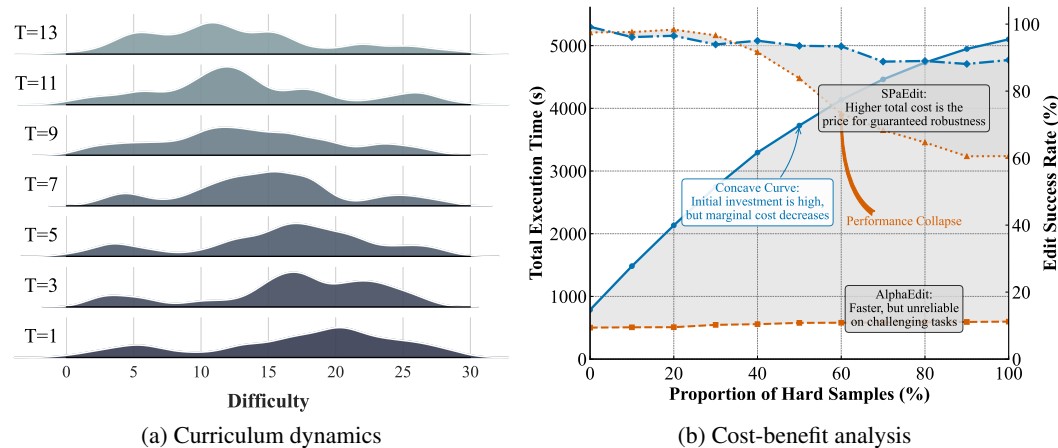

(a) Curriculum dynamics    (b) Cost-benefit analysis

Figure 5: (a) shows easy-to-hard self-paced curriculum dynamics. (b) shows the cost–benefit trade-off: modest extra time yields large efficacy gains on hard samples.

**Temporal Adaptation and Unlearning.** Machine unlearning seeks reliable removal of obsolete or private knowledge from LLMs. Gradient based approaches include forgetting losses (Yao et al., 2024), orthogonal projection updates (Hoang et al., 2024), and Fisher weighted masking (Cha et al., 2024). Memory centric methods externalize edits to ensure isolation after editing (GRACE (Hartvigsen et al., 2023), T-Patcher (Huang et al., 2023), KV scrubbing (Wang et al., 2024a)). These methods assign fixed forget-set targets (e.g., "I don't know" or random answers). Because locate-then-edit is fundamentally a linear-regression update, such targets can induce systematic bias. We therefore propose an interpolation-based unlearning strategy tailored to this setting.

**Curriculum and Self-Paced Learning.** The principle of ordering samples from easy to hard is central to Curriculum Learning (CL), which uses heuristics (Bengio et al., 2009), and Self-Paced Learning (SPL), which automates selection with regularized weights (Kumar et al., 2010). These concepts have since been extended to modern deep learning, from being automated by RL controllers (Graves et al., 2017) to being adapted for LLM instruction-tuning and continual learning (Ke et al., 2022; Liu et al., 2024b; Ge et al., 2025). Despite this broad applicability, these principles have not yet been systematically applied to knowledge editing. Our work bridges this gap by introducing a self-paced learning framework tailored for this task, which yields substantial improvements on difficult edits.

# 6    CONCLUSIONS

In this work, we formalize Relation Editing and expose a key weakness of existing methods: they retain outdated information and fail on difficult edits. We address this with two contributions: the Forgetting-and-Editing (FE) framework, which introduces a targeted unlearning strategy to resolve knowledge conflicts, and SPaEdit, a self paced algorithm for edits of varying difficulty. Our experiments validate both: FE is effective on our new relation editing benchmark, and SPaEdit achieves state of the art on this task and on standard object editing benchmarks. Despite these gains, fully and permanently erasing obsolete relations remains challenging, so future work will develop more effective unlearning mechanisms for relation editing.

## ETHICS STATEMENT

Our research aims to enhance the capability of updating and maintaining knowledge within Large Language Models (LLMs), which is crucial for ensuring their accuracy and timeliness in real-world applications. Our proposed method for Relation Editing, particularly SPaEdit, significantly improves the precision and reliability of knowledge correction in these models.

However, we recognize that any technology capable of directly modifying a model's internal knowledge carries potential risks. For instance, such techniques could be misused to introduce erroneous, harmful, or biased information. We therefore strongly urge researchers in both academia and indus-

try to establish rigorous validation, oversight, and review mechanisms to ensure the ethical deployment and use of these techniques.

Despite these challenges, the original intent of model editing technology is positive, with the core objective of facilitating efficient and effective updates for large models in the future. We encourage researchers to leverage this technology responsibly and with care, collectively guiding its development in a socially beneficial direction.

## REPRODUCIBILITY STATEMENT

To ensure reproducibility, Appendix A details our experimental setup, baselines, dataset construction, and evaluation metrics. All source code and data used in this study, including the SPaEdit implementation and the ReEditBench dataset, are available at `https://anonymous.4open.science/r/RelEdit-7677`. These resources enable independent verification and replication of our results and encourage further research.

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

## USE OF LARGE LANGUAGE MODELS

Per ICLR policy, we report that Large Language Models (LLMs) were used to assist with grammar correction and language polishing for this paper. The human authors conceived all core ideas and analysis, and take full responsibility for the final content.

# A EXPERIMENTAL SETUP

## A.1 DATASET CONSTRUCTION DETAILS

### A.1.1 CONSTRUCTION OF TRAINING DATASET.

This section provides a detailed breakdown of the four-stage pipeline used to construct our **ReEditBench** benchmark. The overall construction process is illustrated in Fig. 6. Our pipeline, which yields 7,918 high-quality relation editing instances, is detailed below.

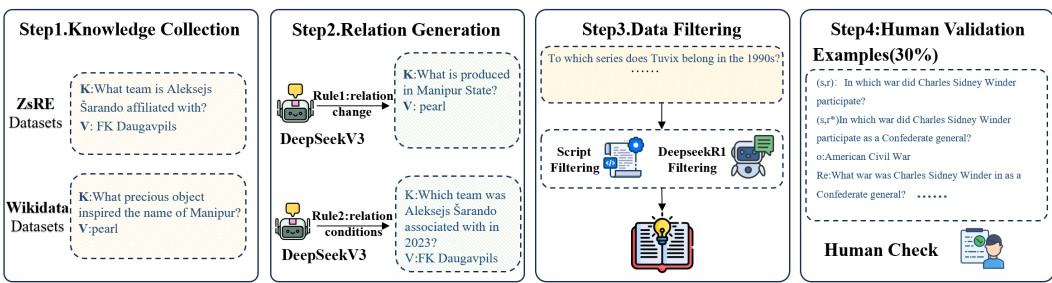

Figure 6: The construction process of our dataset.

**Stage 1: Knowledge Collection.** We began by sourcing our initial knowledge pool from established, high-quality, knowledge-intensive benchmarks, primarily ZsRE and a curated subset of Wikidata. These sources were chosen because they provide structured, fact-checked subject-relation-object triplets $(s, r, o)$, ensuring a factually grounded foundation for our benchmark. This step avoids the noise and ambiguity often associated with sourcing facts directly from raw web text.

**Stage 2: LLM-based Relation Generation.** With the curated facts, we employed a powerful generator LLM, DeepSeekV3, to automatically reframe each fact into a plausible relation-editing task. The generation was guided by a structured prompt that instructed the model to produce a new, related fact by modifying the relation $r$ to $r^*$ while keeping the subject $s$ and object $o$ fixed. The prompt encouraged the generation of two distinct types of relation edits to ensure diversity:

- **New Relation:** This involves a direct modification of the core relationship. For example, the fact ⟨Parag Agrawal, CEO of, Twitter⟩ could be reframed into a new target edit of ⟨Parag Agrawal, CTO of, Twitter⟩.

- **Conditional Relation:** This involves adding a new contextual or temporal constraint to the original relation. For instance, ⟨Joe Biden, President of, USA⟩ could be reframed to target ⟨Joe Biden, 46th President of, USA⟩.

The output of this stage was a large pool of candidate relation edits, each consisting of an original triplet $(s, r, o)$ and a target edited triplet $(s, r^*, o)$.

**Stage 3: Automated Filtering Pipeline.** To ensure the structural and semantic quality of the generated candidates, we implemented a rigorous two-phase automated filtering process.

1. **Script-based Filtering:** An initial pass was conducted using scripts to validate the structural integrity of all generated instances. This phase automatically discarded any malformed outputs, such as those with empty fields, incorrect formatting, or structural deviations from the required triplet format.

2. **LLM-based Verification:** To mitigate the risk of the generator model favorably evaluating its own outputs, we used a separate, independent verifier LLM, DeepseekR1. This verifier was prompted to assess the factual and semantic plausibility of each candidate edit. It was tasked with flagging and removing any instances that contained logical contradictions, factual hallucinations, or were semantically incoherent, ensuring that only high-quality, believable edits proceeded.

**Stage 4: Human Validation.** As the final and most critical quality assurance step, we performed a manual validation on a randomly sampled subset of the data. We sampled 30% of the automatically filtered instances and presented them to human annotators. The annotators were tasked with verifying the factual accuracy, logical consistency, and semantic coherence of each relation edit. The high agreement rate among annotators and the low error rate observed during this process certified the overall high quality and reliability of our automated generation and filtering pipeline.

### A.1.2 CONSTRUCTION AND USAGE OF THE VALIDATION SET FOR MODEL SELECTION

Since our SPaEdit algorithm is an iterative process, a principled method is required to select the optimal model checkpoint. We achieve this by constructing a dedicated validation set and evaluating a composite loss function at each iteration.

**1. Validation Set Construction.** The validation set is created by randomly holding out 20% of the editing instances from the full training dataset before the main editing process begins. For each instance in this validation set, which consists of an original fact $(s, r, o)$ and a target fact $(s, r^*, o)$, we define the following key-value pairs for evaluation:

- **Original Key-Value Pair**: $(\boldsymbol{k}_{\text{org}}, \boldsymbol{v}_{\text{org}})$, where $\boldsymbol{k}_{\text{org}}$ is the key vector corresponding to the original subject-relation pair $(s, r)$, and $\boldsymbol{v}_{\text{org}}$ is the value vector for the object $o$.

- **New Key-Value Pair**: $(\boldsymbol{k}_{\text{new}}, \boldsymbol{v}_{\text{org}})$, where $\boldsymbol{k}_{\text{new}}$ is the key for the new relation $(s, r^*)$. Note that the target value vector $\boldsymbol{v}_{\text{org}}$ remains the same.

- **Paraphrased Key**: $\boldsymbol{k}_{\text{re}}$, which is a semantic rephrasing of the new key $\boldsymbol{k}_{\text{new}}$. This is generated using an external LLM to test for generalization.

- **Forget Target**: $\boldsymbol{v}_{\text{forget}}$, which is the target value for the unlearning process, computed via interpolation as defined in Eq. (5): $\boldsymbol{v}_{\text{forget}} = \boldsymbol{v}_{\text{org}} + \gamma(\boldsymbol{v}_{\text{IDK}} - \boldsymbol{v}_{\text{org}})$.

**2. Iterative Evaluation with a Weighted Loss Function.** At each iteration $t$ of the SPaEdit algorithm, we compute the perturbation $\boldsymbol{\Delta}_t$ and obtain the intermediate edited model weights $\mathbf{W}_t = \mathbf{W} + \boldsymbol{\Delta}_t \mathbf{P}$. We then evaluate this model on the validation set by calculating three distinct loss components based on the squared L2-norm loss function $\ell(\boldsymbol{k}, \boldsymbol{v}) = \left\| \mathbf{W}_t \boldsymbol{k} - \boldsymbol{v} \right\|_2^2$:

1. **Forgetting Loss ($\mathcal{L}_{\textbf{forget}}$):** This loss measures how successfully the model unlearns the original, outdated fact. It is calculated by ensuring the output for the original key $\boldsymbol{k}_{\text{org}}$ moves towards the forget target $\boldsymbol{v}_{\text{forget}}$. This addresses two of your requirements: that the model learns to associate $\boldsymbol{k}_{\text{org}}$ with $\boldsymbol{v}_{\text{forget}}$, and consequently, that its association with $\boldsymbol{v}_{\text{org}}$ is suppressed.

$$\mathcal{L}_{\text{forget}}(t) = \mathbb{E}_{(\boldsymbol{k}_{\text{org}}, \boldsymbol{v}_{\text{forget}})} \left[ \left\| \mathbf{W}_t \boldsymbol{k}_{\text{org}} - \boldsymbol{v}_{\text{forget}} \right\|_2^2 \right] \tag{12}$$

2. **Efficacy Loss ($\mathcal{L}_{\textbf{efficacy}}$):** This loss assesses the direct acquisition of the new knowledge. It is the error between the model's output for the new key $\boldsymbol{k}_{\text{new}}$ and the correct value $\boldsymbol{v}_{\text{org}}$.

$$\mathcal{L}_{\text{efficacy}}(t) = \mathbb{E}_{(\boldsymbol{k}_{\text{new}}, \boldsymbol{v}_{\text{org}})} \left[ \left\| \mathbf{W}_t \boldsymbol{k}_{\text{new}} - \boldsymbol{v}_{\text{org}} \right\|_2^2 \right] \tag{13}$$

3. **Generalization Loss ($\mathcal{L}_{\textbf{gen}}$):** This loss evaluates whether the model can apply the new knowledge to paraphrased prompts, ensuring semantic understanding rather than superficial memorization.

$$\mathcal{L}_{\text{gen}}(t) = \mathbb{E}_{(\boldsymbol{k}_{\text{re}}, \boldsymbol{v}_{\text{org}})} \left[ \left\| \mathbf{W}_t \boldsymbol{k}_{\text{re}} - \boldsymbol{v}_{\text{org}} \right\|_2^2 \right] \tag{14}$$

where $\mathbb{E}[\cdot]$ denotes the average loss over all samples in the validation set.

**3. Final Model Selection.** The total validation loss at iteration $T$, denoted $\mathcal{L}_{\text{val}}(t)$, is a weighted sum of these three components.

$$\mathcal{L}_{\text{val}}(t) = w_{\text{forget}} \cdot \mathcal{L}_{\text{forget}}(t) + w_{\text{efficacy}} \cdot \mathcal{L}_{\text{efficacy}}(t) + w_{\text{gen}} \cdot \mathcal{L}_{\text{gen}}(t) \tag{15}$$

We use an early stopping strategy with a patience of 3 iterations to select the final model. We track the minimum validation loss, $\mathcal{L}_{\text{val}}^*$, observed so far. Training is terminated at the first iteration $T$ where the validation loss has not improved (i.e., decreased by more than a threshold $\epsilon$) for 3 consecutive iterations. The final model is the checkpoint that corresponds to the best observed validation loss $\mathcal{L}_{\text{val}}^*$ up to iteration $T$. In our experiments, the weights are set as hyperparameters to balance the trade-offs between these objectives. For instance, we might use $w_{\text{forget}} = 0.4$, $w_{\text{efficacy}} = 0.4$, and $w_{\text{gen}} = 0.2$.

## A.2 BASELINE METHOD

**ROME** (Meng et al., 2022) Introduces a one-shot, locate-then-edit causal framework that rewrites factual knowledge in large language models without disturbing unrelated parameters. The method first pinpoints the single feed-forward key–value subspace storing the target fact, then derives the optimal rank-one perturbation $\mathbf{\Delta W}$ via gradients, embedding the new key–value mapping while preserving the overall distribution. A subsequent KL-divergence minimization enforces that the edited model behaves identically to the original on general text, yielding "local rewrite, global preservation." Experiments on GPT and BART show that ROME persistently and reliably updates knowledge in a single edit, outperforming prior global fine-tuning or explicit memory approaches with negligible side effects on unrelated facts and downstream tasks.

**Fine-Tuning (FT)** (Zhu et al., 2020) Formalizes knowledge editing as constrained fine-tuning of a minimal parameter subset within the transformer. It freezes all weights except the up- and down-projection matrices of a single MLP layer that gradient analysis identifies as causally critical for the target fact. A small, fact-only dataset is constructed by cloze-style prompts, and standard cross-entropy fine-tuning is performed with an additional $L_2$ proximity term that penalizes deviation from the original parameters. A trust-region optimizer keeps parameter drift within a preset radius, ensuring that the update remains locally confined while the new association is encoded. This lightweight fine-tuning paradigm yields reliable edits without the need for custom architectural modules.

**MEMIT** (Meng et al., 2023) Scales causal model editing from single facts to thousands by exploiting the linear key–value associative memory implicit in feed-forward layers. It jointly identifies a small set of critical layers and simultaneously applies rank-one updates to their MLP up-projection matrices, rewriting all target associations in one forward pass. An under-determined least-squares objective with $\ell_2$ and locality-of-edit regularizers ensures that the new memories satisfy output constraints while minimizing disturbance to unrelated knowledge, and a closed-form solution avoids expensive iterative optimization.

**RECT** (Gu et al., 2024) Reformulates model editing as a low-rank, layer-wise correction problem that explicitly accounts for causal traces of factual recall. Instead of a single update, RECT identifies a minimal set of $k$ contiguous MLP layers whose hidden representations are causally most responsible for a given fact, and applies rank-$r$ ($r \leq 4$) updates only to their down-projection matrices. A consistency loss that penalizes both output drift and internal representation shift is minimized, ensuring that edited knowledge is both effective and faithful to the original distribution. Traceability is enforced by an additional regularizer that collapses the updated subspace onto the principal component of the fact's context, enabling post-hoc verification.

**NSE** (Jiang et al., 2024) Reframes knowledge editing as neuron-level intervention within the feed-forward layers of transformer LMs. The algorithm first detects a sparse subset of neurons whose activations are maximally predictive of the target fact via integrated-gradients attribution. It then introduces fact-specific scaling vectors that multiplicatively modulate the output of these neurons, while additive bias terms shift their activation baselines to encode the new association. A two-stage optimization alternates between (i) closed-form least-squares fitting of the scaling/bias parameters to satisfy the editing objective and (ii) a distribution-preserving regularizer that minimizes KL divergence on held-out corpora. By confining changes to a handful of neuron-specific parameters, NSE achieves fine-grained edits without altering global layer weights.

**PRUNE** (Ma et al., 2025) Treats model editing as parameter-efficient subspace pruning within the MLP blocks of transformer LMs. For each fact to be updated, it first identifies a task-specific sparse

mask over the rows of the up-projection matrix by gradient-based saliency scoring; the unmasked weights are frozen. A subsequent low-rank adapter is then trained only on the pruned subspace to encode the new key–value association, while a KL-divergence regularizer penalizes any deviation in the model's distribution on unrelated contexts. This pruning-plus-adaptation pipeline yields localized, modular edits that can be independently stored, swapped, or revoked without re-touching the original parameters.

**AlphaEdit** (Fang et al., 2025) Augments the locate-then-edit pipeline with a null-space projection that prevents any parameter perturbation from disturbing previously stored knowledge. After obtaining the standard update $\boldsymbol{\Delta}$ via least-squares on the target key-value pairs, AlphaEdit multiplies $\boldsymbol{\Delta}$ by the projection matrix $\mathbf{P} = \mathbf{U}_0\mathbf{U}_0^\top$, where $\mathbf{U}_0$ spans the left null space of the covariance matrix built from keys of the preserved knowledge. The projected perturbation $\boldsymbol{\Delta}\mathbf{P}$ satisfies $\boldsymbol{\Delta}\mathbf{P}\mathbf{K}_0 = \mathbf{0}$, ensuring the edited model still outputs the original values for all preserved associations while focusing capacity on the new fact. The resulting closed-form update $\boldsymbol{\Delta}\mathbf{P} = (\mathbf{V}_1 - \mathbf{W}\mathbf{K}_1)\mathbf{K}_1^\top\mathbf{P}\big(\mathbf{K}_1\mathbf{K}_1^\top\mathbf{P} + \mathbf{K}_p\mathbf{K}_p^\top\mathbf{P} + \mathbf{I}\big)^{-1}$ plugs into existing editors with one line of code and negligible runtime overhead.

## A.3 METRICS

### A.3.1 ZsRE METRICS

Following the previous work, this section defines each ZsRE metric given a LLM $f_\theta$, a knowledge fact prompt $(s_i, r_i)$, an edited target output $o_i$, and the model's original output $o_c^i$:

- **Efficacy:** Efficacy is calculated as the average top-1 accuracy on the edit samples:

$$\mathbb{E}_i\left\{o_i = \arg\max_o \mathbb{P}_{f_\theta}(o|(s_i, r_i))\right\} \tag{16}$$

- **Generalization:** Generalization measures the model's performance on equivalent prompt of $(s_i, r_i)$, such as rephrased statements $N((s_i, r_i))$. This is evaluated by the average top-1 accuracy on these $N((s_i, r_i))$:

$$\mathbb{E}_i\left\{o_i = \arg\max_o \mathbb{P}_{f_\theta}(o|N((s_i, r_i)))\right\} \tag{17}$$

- **Specificity:** Specificity ensures that the editing does not affect samples unrelated to the edit cases $O(s_i, r_i)$. This is evaluated by the top-1 accuracy of predictions that remain unchanged:

$$\mathbb{E}_i\left\{o_i^c = \arg\max_o P_{f_\theta}(o|O((s_i, r_i)))\right\} \tag{18}$$

### A.3.2 COUNTERFACT METRICS

Following previous work, this section defines the evaluation metrics for the Counterfact dataset. To ensure a consistent and fair comparison with the ZsRE benchmark, we adopt its top-1 accuracy-based evaluation methodology for the Efficacy, Generalization, and Specificity metrics. Therefore, we only present the definitions for the remaining metrics unique to this generative task evaluation:

- **Fluency (generation entropy):** Measure for excessive repetition in model outputs. It uses the entropy of n-gram distributions:

$$-\frac{2}{3}\sum_k g_2(k)\log_2 g_2(k) + \frac{4}{3}\sum_k g_3(k)\log_2 g_3(k) \tag{19}$$

- **Consistency (reference score):** The consistency of the model's outputs is evaluated by computing the cosine similarity between the TF-IDF vectors of the model-generated text and a reference Wikipedia text.

### A.3.3 ReLEditBench METRICS

This section defines each ReLEditBench metric given an original fact $(s, r, o)$ and a new fact $(s, r^*, o)$:

- **Success:** The Success score is a joint metric that holistically verifies if a knowledge edit was successful. As formulated in Eqn. 20, it requires two conditions to be met simultaneously: (i) the model must no longer predict the original object $o$ for the original query $(s, r)$ and (ii) it must correctly predict the new object $o$ for the updated query $(s, r^*)$.

$$\mathbb{E}_{x \sim \mathcal{D}} \left[ \mathbf{1} \left\{ o_i = \arg \max_{\neg o} \mathbb{P}_{f_\theta}(o \mid (s, r)) \right\}, \ \mathbf{1} \left\{ o_i = \arg \max_{o} \mathbb{P}_{f_\theta}(o \mid (s, r^*)) \right\} \right] \quad (20)$$

- **Retention:** The Retain metric evaluates whether the model successfully retains the newly introduced knowledge after the edit. As defined in Eqn. 21, it measures the probability that the new object $o_i$ is the top prediction for the new prompt.

$$\mathbb{E}_i \left\{ o_i = \arg \max_{o} \mathbb{P}_{f_\theta}(o|(s, r)) \right\} \quad (21)$$

- **Efficient:** The Efficacy score measures the model's direct acquisition of the new fact. It is defined in Eqn. 22 as the probability that the new object $o_i$ is the top prediction for the new prompt $(s, r^*)$. A high score signifies that the new knowledge has been successfully instilled.

$$\mathbb{E}_i \left\{ o_i = \arg \max_{o} \mathbb{P}_{f_\theta}(o|(s, r^*)) \right\} \quad (22)$$

- **Generalization:** This metric evaluates if the model can apply the new knowledge beyond the specific prompt it was edited on. As shown in Eqn. 23, it measures the model's ability to predict the correct object $o'$ when presented with a set of paraphrased or semantically equivalent prompts $N((s, r^*))$:

$$\mathbb{E}_i \left\{ o = \arg \max_{o'} P_{f_\theta}(o'|N((s, r^*))) \right\} \quad (23)$$

### A.4 EXPERIMENTAL DETAILS

This appendix details the hyperparameters used in our experiments. It is divided into two parts: the first outlines the base configuration parameters for the Large Language Models (LLMs), and the second elaborates on the key hyperparameters for our proposed SPaEdit and Forgetting-and-Editing (FE) strategies.

#### A.4.1 MODEL CONFIGURATION PARAMETERS

The following table summarizes the main configuration parameters used for each of the three base models. These are primarily defined within their respective JSON configuration files.

Table 4: Base configuration parameters for the LLMs used in the experiments.

| Parameter | Value | Description |
|---|---|---|
| **model_name** | `EleutherAI_gpt-j-6B`, `gpt2-xl`, `Llama3-8B` | Specifies the pretrained language model. |
| **layers** | [3-8], [13-17], [4-8] | The target Transformer layers for editing. |
| **v_num_grad_steps** | 25 or 20 | Number of gradient steps for value vector computation. |
| **v_lr** | `5e-1` or `1e-1` | Learning rate used during value vector computation. |
| **v_loss_layer** | 27, 47, 31 | The specific model layer used to compute the edit loss. |
| **kl_factor** | `0.0625` | Weight of the KL-divergence regularization term. |
| **mom2_dataset** | `wikipedia` | Dataset for computing second-moment statistics. |
| **rewrite_module_tmp** | Varies by model | Template for the path to the module being rewritten. |

**Key Hyperparameters for the SPaEdit and FE Strategies** In addition to the base configurations, our proposed algorithms are governed by several key hyperparameters that control the editing and forgetting behavior.

- **Forgetting Interpolation Factor** ($\gamma$): This is the core hyperparameter of our FE strategy, as defined in Eqn. 5. It controls the degree of interpolation from the original fact's value vector, $v(o)$, towards a neutral "I don't know" state, $v(\text{IDK})$. A higher $\gamma$ value enforces a more thorough forgetting of the outdated information. In our experiments, this was set to 0.4 for GPT-J-6B and 0.6 for both LLaMA3-8B and GPT2-XL to achieve an optimal balance between forgetting and learning.

- **Update Regularization Coefficients ($\alpha$ and $\beta$)**: These coefficients in the SPaEdit objective function (Eqn. 7) regularize the update perturbation $\boldsymbol{\Delta}$ to maintain model stability.
  - $\alpha$ constrains the overall magnitude of the update, preventing large, potentially disruptive changes to the model's parameters.
  - $\beta$ minimizes the edit's impact on a set of preserved knowledge keys $\mathbf{K}_p$, ensuring that unrelated information is not corrupted.

  Throughout our experiments, we set $\alpha = 10$ and $\beta = 1$ to apply strong general regularization while precisely preserving prior knowledge.

- **Self-Paced Learning Curriculum Parameters ($\lambda_0$, $\mu$, and $T$)**: These parameters define the "easy-to-hard" curriculum for the SPaEdit algorithm, as outlined in Algorithm 1.
  - $\lambda_0$ (Initial Pace Parameter): The initial difficulty threshold, which determines the set of the "easiest" samples to be edited at the beginning of the process.
  - $\mu$ (Pace Growth Factor): The multiplicative factor by which the difficulty threshold $\lambda$ is increased in each iteration ($\lambda \leftarrow \mu\lambda$). This controls the pace at which more challenging samples are introduced into the training set.
  - $T$ (Max Iterations): The Max number of iterations in the curriculum, defining the overall length of the optimization process.

  For our experiments, we set the initial pace to $\lambda_0 = 10$, the growth factor to $\mu = 1.1$, and the total number of iterations to $T = 20$. This configuration allows the model to first converge on easy edits before gradually incorporating more difficult ones, enhancing overall robustness and success rate.

## B  IMPLEMENTATION DETAILS AND RELATED PROOFS

### B.1  THEORETICAL ANALYSIS OF THE FORGETTING-AND-EDITING STRATEGY

We analyze the proposed Forgetting-and-Editing (FE) strategy within the linear regression framework. The core of our strategy is to generate a modified representation for the target object through interpolation:

$$\boldsymbol{v}(\hat{o}) = \boldsymbol{v}(o) + \gamma[\boldsymbol{v}(\text{IDK}) - \boldsymbol{v}(o)], \quad \gamma \in (0, 1). \tag{24}$$

This operation is performed for each sample in the forgetting set $\mathbb{D}_b$. To analyze its effect within the regression framework, we translate this representation-level operation into label-space formulation. For any sample $i$ in $\mathbb{D}_b$, the modified target label becomes:

$$\boldsymbol{v}(\hat{o}_i) = (1 - \gamma)\boldsymbol{v}(o_i) + \gamma\boldsymbol{v}(\text{IDK}). \tag{25}$$

Extending this operation to the entire forgetting set of $M$ samples, we define:

- The original label vector: $\boldsymbol{y}_b = [\boldsymbol{v}(o_1), \boldsymbol{v}(o_2), ..., \boldsymbol{v}(o_M)]^\top$
- The IDK label vector: $\boldsymbol{y}_{\text{IDK}} = [\boldsymbol{v}(\text{IDK}), \boldsymbol{v}(\text{IDK}), ..., \boldsymbol{v}(\text{IDK})]^\top$

The FE strategy effectively applies the same linear interpolation to each corresponding element of these vectors, yielding the modified label vector:

$$\begin{aligned}
\boldsymbol{y}_b^{\text{FE}} &= [\boldsymbol{v}(\hat{o}_1), \boldsymbol{v}(\hat{o}_2), ..., \boldsymbol{v}(\hat{o}_M)]^\top \\
&= \boldsymbol{y}_b + \gamma(\boldsymbol{y}_{\text{IDK}} - \boldsymbol{y}_b).
\end{aligned} \tag{26}$$

Substituting this into the closed-form solution of the linear regression problem yields:

$$\begin{aligned}
\boldsymbol{w}_{\text{FE}}^* &= (\mathbf{X}^\top\mathbf{X})^{-1}(\mathbf{X}_g^\top\boldsymbol{y}_g + \mathbf{X}_b^\top\boldsymbol{y}_b^{\text{FE}}) \\
&= (\mathbf{X}^\top\mathbf{X})^{-1}\left(\mathbf{X}_g^\top\boldsymbol{y}_g + \mathbf{X}_b^\top\left[\boldsymbol{y}_b + \gamma(\boldsymbol{y}_{\text{IDK}} - \boldsymbol{y}_b)\right]\right) \\
&= \boldsymbol{w}_g^* + \gamma(\mathbf{X}^\top\mathbf{X})^{-1}\mathbf{X}_b^\top(\boldsymbol{y}_{\text{IDK}} - \boldsymbol{y}_b),
\end{aligned} \tag{27}$$

where $\boldsymbol{w}_g^* = (\mathbf{X}^\top \mathbf{X})^{-1} \mathbf{X}_g^\top \boldsymbol{y}_g$ is the solution trained solely on normal data.

**Comparative Advantages.**   The proposed Feature Editing (FE) strategy offers distinct advantages over methods that employ a fixed value (e.g., I don't know) or random answers for forgetting. Unlike the constant bias introduced by a fixed label or the uncontrolled bias from random assignment, our approach generates a non-constant, data-dependent bias term $\gamma(\mathbf{X}^\top \mathbf{X})^{-1} \mathbf{X}_b^\top (\boldsymbol{y}_{\text{IDK}} - \boldsymbol{y}_b)$. This key difference prevents systematic bias and preserves prediction diversity. Furthermore, the hyperparameter $\gamma$ provides precise and continuous control over the forgetting strength, a feature unavailable in conventional methods. The optimization process remains stable due to the proximity between the original and target features, avoiding large gradients. Finally, the adjustment is highly targeted, effectively removing specific information while minimizing distortion to the model's normal knowledge.

## B.2   FORMULATION OF THE MULTI-OBJECTIVE OPTIMIZATION PROBLEM

The fundamental goal of parameter-modifying knowledge editing is to find a minimal perturbation $\boldsymbol{\Delta}$, to a model's weight matrix, $\mathbf{W}$, such that the edited model $\mathbf{W}' = \mathbf{W} + \boldsymbol{\Delta}$ reflects new knowledge without catastrophically forgetting existing information. This can be framed as a multi-objective optimization problem.

Let us define the key components:

- **New Knowledge (Update Set):** A set of new facts to be incorporated, represented by key-value pairs $\{(\boldsymbol{k}_i, \boldsymbol{v}_i)\}$. We can stack these into matrices $\mathbf{K}_1$ (keys) and $\mathbf{V}_1$ (values). The objective is to make the model output $\mathbf{V}_1$ when given $\mathbf{K}_1$. The error for this is captured by the term $\mathcal{L}_{\text{update}} = \|(\mathbf{W} + \boldsymbol{\Delta})\mathbf{K}_1 - \mathbf{V}_1\|_F^2$.

- **Preserved Knowledge (Preservation Set):** The vast set of existing knowledge that must remain unchanged. This is represented by key-value pairs $\{(\boldsymbol{k}_j, \boldsymbol{v}_j)\}$ stacked into matrices $\mathbf{K}_0$ and $\mathbf{V}_0$. Since the pre-trained model $\mathbf{W}$ is assumed to already store this knowledge, we have $\mathbf{W}\mathbf{K}_0 \approx \mathbf{V}_0$. The objective is to minimize the change in output for these keys, giving an error term $\mathcal{L}_{\text{preserve}} = \|(\mathbf{W} + \boldsymbol{\Delta})\mathbf{K}_0 - \mathbf{V}_0\|_F^2$.

- **Regularization:** To prevent the perturbation $\boldsymbol{\Delta}$ from becoming excessively large and harming the model's general abilities, a regularization term on the perturbation itself is included, $\mathcal{L}_{\text{reg}} = \|\boldsymbol{\Delta}\|_F^2$.

Combining these objectives, we arrive at the standard optimization problem for knowledge editing:

$$\min_{\boldsymbol{\Delta}} \mathcal{L}(\boldsymbol{\Delta}) = \underbrace{\|(\mathbf{W} + \boldsymbol{\Delta})\mathbf{K}_1 - \mathbf{V}_1\|_F^2}_{\text{Update Error}} + \alpha \underbrace{\|(\mathbf{W} + \boldsymbol{\Delta})\mathbf{K}_0 - \mathbf{V}_0\|_F^2}_{\text{Preservation Error}} + \beta \underbrace{\|\boldsymbol{\Delta}\|_F^2}_{\text{Regularization}} \tag{28}$$

where $\alpha$ and $\beta$ are hyperparameters that balance the trade-off between the objectives.

We can simplify this expression. Since $\mathbf{W}\mathbf{K}_0 = \mathbf{V}_0$, the preservation term becomes $\|(\mathbf{W} + \boldsymbol{\Delta})\mathbf{K}_0 - \mathbf{W}\mathbf{K}_0\|_F^2 = \|\boldsymbol{\Delta}\mathbf{K}_0\|_F^2$. For the update term, we can define the **residual matrix** $\mathbf{R} = \mathbf{V}_1 - \mathbf{W}\mathbf{K}_1$, which represents the error that the edit must correct. The term thus becomes $\|\boldsymbol{\Delta}\mathbf{K}_1 - \mathbf{R}\|_F^2$. The simplified objective is:

$$\min_{\boldsymbol{\Delta}} \mathcal{L}(\boldsymbol{\Delta}) = \|\boldsymbol{\Delta}\mathbf{K}_1 - \mathbf{R}\|_F^2 + \alpha\|\boldsymbol{\Delta}\mathbf{K}_0\|_F^2 + \beta\|\boldsymbol{\Delta}\|_F^2 \tag{29}$$

**Incorporating the Null-Space Projection.**   A key innovation from methods like AlphaEdit is to constrain the update $\boldsymbol{\Delta}$ to the null-space of the preserved knowledge. This theoretically guarantees that the edit does not interfere with this knowledge. This is achieved using a projection matrix $\mathbf{P}$.

- Let $\mathbf{K}_p$ be the matrix of keys for the knowledge we wish to explicitly preserve. $\mathbf{K}_p$ is the concrete realization of the abstract $\mathbf{K}_0$ used to build the projector.

- The null-space projection matrix $\mathbf{P}$ is constructed such that for any matrix $\mathbf{A}$, the update $\mathbf{A}\mathbf{P}$ satisfies $(\mathbf{A}\mathbf{P})\mathbf{K}_p = \mathbf{0}$. $\mathbf{P}$ is symmetric ($\mathbf{P} = \mathbf{P}^\top$) and idempotent ($\mathbf{P}^2 = \mathbf{P}$).

By replacing the raw perturbation $\boldsymbol{\Delta}$ with the projected perturbation $\boldsymbol{\Delta}\mathbf{P}$, we enforce this non-interference constraint. The optimization objective is adapted to solve for an optimal update within

this safe subspace. The preservation term is now implicitly handled by the projection, but can be kept as a soft constraint, while the regularization term is applied to the projected update. This leads to an objective of the form:

$$\min_{\boldsymbol{\Delta}} \mathcal{L}(\boldsymbol{\Delta}) = \|\boldsymbol{\Delta}\mathbf{P}\mathbf{K}_1 - \mathbf{R}\|_F^2 + \alpha'\|\boldsymbol{\Delta}\mathbf{P}\|_F^2 + \beta'\|\boldsymbol{\Delta}\mathbf{P}\mathbf{K}_p\|_F^2 \tag{30}$$

Here, the hyperparameters $\alpha'$ and $\beta'$ now regularize the magnitude of the projected update and explicitly penalize any residual interference with the preserved set $\mathbf{K}_p$.

**Introducing Self-Paced Learning (SPL).** The final step in the paper's methodology (SPaEdit) is the introduction of a self-paced learning curriculum. This acknowledges that not all edits are equally difficult. The model should first learn from "easy" samples and gradually incorporate "harder" ones. This is implemented via a binary selection matrix $\mathbf{Z}$.

- $\mathbf{Z}$ is a diagonal matrix where each diagonal entry $z_i \in \{0, 1\}$.
- At each iteration, $z_i = 1$ if the $i$-th sample is deemed "easy" (i.e., its loss is below a certain threshold $\lambda$); otherwise, $z_i = 0$.
- This selection matrix is applied only to the **update error term**, effectively masking out the hard samples for the current iteration. To keep the objective quadratic, we use its square root, $\mathbf{Z}^{1/2}$ (which is equal to $\mathbf{Z}$ since its elements are 0 or 1).

By integrating the selection matrix $\mathbf{Z}$ into Eqn. 30, we arrive at the final optimization problem as formulated in the paper:

$$\min_{\boldsymbol{\Delta}} \mathcal{L}(\boldsymbol{\Delta}) = \|(\boldsymbol{\Delta}\mathbf{P}\mathbf{K}_1 - \mathbf{R})\mathbf{Z}^{1/2}\|_F^2 + \alpha\|\boldsymbol{\Delta}\mathbf{P}\|_F^2 + \beta\|\boldsymbol{\Delta}\mathbf{P}\mathbf{K}_p\|_F^2 \tag{31}$$

This final form is what the paper uses to derive a closed-form solution for the projected update $\boldsymbol{\Delta}_{\text{SPaEdit}} = \boldsymbol{\Delta}\mathbf{P}$. The key terms are:

- $\boldsymbol{\Delta}$: The raw perturbation matrix we are solving for.
- $\mathbf{P}$: The null-space projection matrix, which is symmetric ($\mathbf{P} = \mathbf{P}^\top$) and idempotent ($\mathbf{P}^2 = \mathbf{P}$).
- $\mathbf{Z}$: The diagonal selection matrix ($\mathbf{Z} = \mathbf{Z}^\top$, $\mathbf{Z}^2 = \mathbf{Z}$, $\mathbf{Z}^{1/2} = \mathbf{Z}$).
- $\mathbf{K}_1, \mathbf{R}, \mathbf{K}_p, \alpha, \beta$: As defined previously.

### B.3 DERIVATION OF THE CLOSED-FORM SOLUTION

The objective function $\mathcal{L}(\boldsymbol{\Delta})$ is convex with respect to $\boldsymbol{\Delta}$. We can find the minimum by taking the gradient with respect to $\boldsymbol{\Delta}$ and setting it to zero.

First, we expand the objective using the trace operator, as $\|\mathbf{X}\|_F^2 = \text{Tr}(\mathbf{X}^\top\mathbf{X})$.

$$\mathcal{L}(\boldsymbol{\Delta}) = \text{Tr}\left(((\boldsymbol{\Delta}\mathbf{P}\mathbf{K}_1 - \mathbf{R})\mathbf{Z})^\top((\boldsymbol{\Delta}\mathbf{P}\mathbf{K}_1 - \mathbf{R})\mathbf{Z})\right)$$
$$+ \alpha\text{Tr}\left((\boldsymbol{\Delta}\mathbf{P})^\top(\boldsymbol{\Delta}\mathbf{P})\right) + \beta\text{Tr}\left((\boldsymbol{\Delta}\mathbf{P}\mathbf{K}_p)^\top(\boldsymbol{\Delta}\mathbf{P}\mathbf{K}_p)\right) \tag{32}$$
$$= \text{Tr}\left(\mathbf{Z}(\mathbf{K}_1^\top\mathbf{P}^\top\boldsymbol{\Delta}^\top - \mathbf{R}^\top)(\boldsymbol{\Delta}\mathbf{P}\mathbf{K}_1\mathbf{Z} - \mathbf{R}\mathbf{Z})\right)$$
$$+ \alpha\text{Tr}\left(\mathbf{P}^\top\boldsymbol{\Delta}^\top\boldsymbol{\Delta}\mathbf{P}\right) + \beta\text{Tr}\left(\mathbf{K}_p^\top\mathbf{P}^\top\boldsymbol{\Delta}^\top\boldsymbol{\Delta}\mathbf{P}\mathbf{K}_p\right) \tag{33}$$

Using the properties $\mathbf{P} = \mathbf{P}^\top$ and the cyclic property of the trace, we can rewrite each term:

$$\mathcal{L}(\boldsymbol{\Delta}) = \text{Tr}(\boldsymbol{\Delta}\mathbf{P}\mathbf{K}_1\mathbf{Z}\mathbf{K}_1^\top\mathbf{P}\boldsymbol{\Delta}^\top) - 2\text{Tr}(\boldsymbol{\Delta}\mathbf{P}\mathbf{K}_1\mathbf{Z}\mathbf{R}^\top) + \text{Tr}(\mathbf{R}\mathbf{Z}\mathbf{R}^\top)$$
$$+ \alpha\text{Tr}(\boldsymbol{\Delta}\mathbf{P}\mathbf{P}\boldsymbol{\Delta}^\top) + \beta\text{Tr}(\boldsymbol{\Delta}\mathbf{P}\mathbf{K}_p\mathbf{K}_p^\top\mathbf{P}\boldsymbol{\Delta}^\top) \tag{34}$$

Now, we compute the gradient $\nabla_{\boldsymbol{\Delta}}\mathcal{L}(\boldsymbol{\Delta})$. Using the matrix calculus identities $\nabla_{\mathbf{X}}\text{Tr}(\mathbf{B}\mathbf{X}^\top) = \mathbf{B}$ and $\nabla_{\mathbf{X}}\text{Tr}(\mathbf{X}\mathbf{B}\mathbf{X}^\top\mathbf{C}) = \mathbf{C}\mathbf{X}\mathbf{B} + \mathbf{C}^\top\mathbf{X}\mathbf{B}^\top$:

$$\nabla_{\boldsymbol{\Delta}}\mathcal{L}(\boldsymbol{\Delta}) = 2(\boldsymbol{\Delta}\mathbf{P}\mathbf{K}_1\mathbf{Z}\mathbf{K}_1^\top\mathbf{P}) - 2(\mathbf{R}\mathbf{Z}\mathbf{K}_1^\top\mathbf{P})$$
$$+ 2\alpha(\boldsymbol{\Delta}\mathbf{P}\mathbf{P}) + 2\beta(\boldsymbol{\Delta}\mathbf{P}\mathbf{K}_p\mathbf{K}_p^\top\mathbf{P}) \tag{35}$$

Since $\mathbf{P} = \mathbf{P}^2$, we can simplify $\mathbf{\Delta PP} = \mathbf{\Delta P}$. Setting the gradient to zero to find the minimum:

$$2(\mathbf{\Delta P K_1 Z K_1^\top P}) - 2(\mathbf{R Z K_1^\top P}) + 2\alpha(\mathbf{\Delta P}) + 2\beta(\mathbf{\Delta P K_p K_p^\top P}) = 0 \qquad (36)$$

Dividing by 2 and rearranging to isolate terms with $\mathbf{\Delta}$:

$$(\mathbf{\Delta P K_1 Z K_1^\top P}) + \alpha(\mathbf{\Delta P}) + \beta(\mathbf{\Delta P K_p K_p^\top P}) = \mathbf{R Z K_1^\top P} \qquad (37)$$

Let $\mathbf{\Delta}_{\text{SPaEdit}} = \mathbf{\Delta P}$ represent the final projected update. We can factor $\mathbf{A}_{\text{SPaEdit}}$ out from the left-hand side:

$$\mathbf{\Delta}_{\text{SPaEdit}}(\mathbf{K_1 Z K_1^\top P} + \alpha\mathbf{I} + \beta\mathbf{K_p K_p^\top P}) = \mathbf{R Z K_1^\top P} \qquad (38)$$

Finally, by right-multiplying by the inverse of the term in the parenthesis, we obtain the closed-form solution for the effective update $\mathbf{\Delta}_{\text{SPaEdit}}$:

$$\mathbf{\Delta}_{\text{SPaEdit}} = (\mathbf{R Z K_1^\top P})(\mathbf{K_1 Z K_1^\top P} + \beta\mathbf{K_p K_p^\top P} + \alpha\mathbf{I})^{-1} \qquad (39)$$

This is the rigorous derivation for the update rule. The matrix $(\mathbf{K_1 Z K_1^\top P} + \beta\mathbf{K_p K_p^\top P} + \alpha\mathbf{I})$ is guaranteed to be invertible because $\mathbf{K_1 Z K_1^\top P}$ and $\beta\mathbf{K_p K_p^\top P}$ are positive semi-definite, and the addition of the regularizer $\alpha\mathbf{I}$ (for $\alpha > 0$) makes the entire matrix positive definite and thus invertible.

*Note:* Some papers may present a slightly simplified version of this formula. The version derived here is the one that follows directly and rigorously from the stated optimization objective. For instance, the final $\mathbf{P}$ in the term $\mathbf{R Z K_1^\top P}$ might be omitted in some implementations, but including it is mathematically consistent as the entire equation operates within the projected subspace.

## C  MORE EXPERIMENTAL RESULTS

### C.1  COMPREHENSIVE ABLATION STUDY ON FORGETTING STRATEGIES

To provide a comprehensive validation of our Forgetting-and-Editing (FE) design, we conducted an extensive ablation study. We compare the performance of several editing methods under four distinct conditions: 1) the Baseline method without any forgetting component; 2) using a naive IDK forgetting target; 3) using a Random forgetting target; and 4) using Our proposed interpolation-based FE strategy.

The results are detailed in Table 5. We present the core metrics of Retention ($\downarrow$) and Efficacy ($\uparrow$) to facilitate a direct comparison of the trade-offs involved in each strategy across all models and key methods.

Table 5: Side-by-side comparison of forgetting strategies across all models and key methods. For each strategy, we report Retention (lower is better) and Efficacy ( higher is better). Our proposed strategy consistently achieves the best balance, delivering the lowest retention while simultaneously maximizing efficacy.

| LLM | Method | No-Forgetting | | + FE (IDK) | | + FE (Random) | | + FE (Ours) | |
|---|---|---|---|---|---|---|---|---|---|
| | | Retention $\downarrow$ | Efficacy $\uparrow$ | Retention$\downarrow$ | Efficacy$\uparrow$ | Retention$\downarrow$ | Efficacy$\uparrow$ | Retention$\downarrow$ | Efficacy$\uparrow$ |
| **LLaMA3** | AlphaEdit | 88.34 | 89.17 | 76.11 | 75.23 | 76.90 | 78.19 | 74.50 | 83.24 |
| | **SPaEdit** | 88.56 | 83.23 | 75.92 | 83.48 | 70.41 | 82.17 | **68.56** | **87.37** |
| **GPT2-XL** | AlphaEdit | 91.31 | 88.83 | 60.25 | 83.45 | 65.81 | 84.90 | 50.46 | 87.36 |
| | **SPaEdit** | 68.55 | 85.93 | 55.18 | 80.15 | 61.33 | 81.82 | **48.78** | **88.46** |
| **GPT-J** | AlphaEdit | 98.20 | 99.53 | 81.67 | 89.12 | 85.43 | 81.30 | 77.84 | 85.64 |
| | **SPaEdit** | 88.24 | 85.93 | 65.40 | 88.31 | 72.88 | 89.04 | **59.84** | **88.08** |

**Analysis of Results.** The side-by-side comparison in Table 5 provides a clear and consistent picture across all experimental settings, revealing the critical impact of the chosen forgetting strategy:

- **Naive Strategies Lead to an Unfavorable Trade-off:** While applying naive forgetting targets like IDK and Random generally succeeds in lowering Retention compared to the No-Forgetting baseline, this benefit comes at a significant and often unacceptable cost. In most cases, particularly

on GPT2-XL, these strategies lead to a noticeable degradation in Efficacy. For instance, SPaEdit's Efficacy on GPT2-XL drops from 85.93% to 80.15% with IDK. Even in scenarios where Efficacy does not drop (e.g., SPaEdit on LLaMA3), the improvement is marginal and the Retention rate remains substantially higher than what our method achieves. This demonstrates that naive approaches force a difficult trade-off: one must sacrifice the model's ability to learn new facts in order to forget old ones.

- **Our FE Strategy is the Most Effective at Unlearning:** A key, unambiguous finding is that our proposed strategy is the most powerful tool for unlearning. Across every model and for both AlphaEdit and SPaEdit, our method consistently achieves the lowest Retention rate. On GPT2-XL, it reduces SPaEdit's Retention to just 48.78%, a figure far superior to any other strategy, proving its state-of-the-art capability in erasing outdated knowledge.

- **Synergistic Effect: A Superior Balance of Forgetting and Learning:** Most critically, our strategy is the only one that resolves the trade-off, creating a synergistic synergistic effect. It not only achieves the best unlearning (lowest Retention) but does so while consistently maintaining or significantly improving Efficacy. For our SPaEdit method, applying the "Ours" strategy boosted Efficacy from 83.23% to 87.37% on LLaMA3 and from 85.93% to 88.46% on GPT2-XL, all while achieving the lowest Retention scores. This stands in stark contrast to the compromised performance of naive approaches and confirms that our carefully designed forgetting targets do not disrupt learning but actually facilitate a cleaner, more effective integration of new knowledge.

In conclusion, this comprehensive ablation confirms that how a model is instructed to "forget" is as important as the instruction to "learn." Our interpolation-based target assignment provides a robust, effective, and non-destructive mechanism for unlearning, establishing a new SOTA for clean and efficient relational editing.

## C.2 ADDITIONAL EXPERIMENTAL RESULTS ON OBJECT EDITING

To provide a more comprehensive validation of our method, we present further results on the demanding CounterFact benchmark. This dataset is particularly challenging, focusing on counter-intuitive factual edits that require precise model updates.

Table 6: Object Editing Performance on the CounterFact Hard Subset. SPaEdit consistently outperforms prior methods, notably achieving state-of-the-art Fluency, indicating higher-quality text generation post-edit.

| LLM | Method | Efficacy↑ | Generalization↑ | Specificity↑ | Fluency↑ | Consistency↑ |
|---|---|---|---|---|---|---|
| LLaMA3 | ROME | 32.02 | 33.41 | 34.31 | 425.55 | 13.01 |
| | MEMIT | 69.22 | 65.61 | 30.54 | 629.68 | 53.15 |
| | AlphaEdit | 79.21 | 73.54 | 30.92 | 629.91 | 56.67 |
| | SPaEdit (Ours) | **92.80** | **95.21** | **42.51** | **631.11** | **56.78** |
| GPT2-XL | ROME | 39.42 | 30.01 | 5.82 | 592.64 | 65.09 |
| | MEMIT | 70.45 | 72.98 | 7.93 | 465.78 | 53.58 |
| | AlphaEdit | 83.22 | 83.91 | 8.54 | 621.76 | 55.62 |
| | SPaEdit (Ours) | **92.66** | **94.82** | **9.62** | **629.26** | 54.52 |
| GPT-J | ROME | 32.05 | 37.01 | 25.76 | 514.82 | 15.64 |
| | MEMIT | 79.22 | 78.27 | 27.58 | 618.93 | 57.84 |
| | AlphaEdit | 87.52 | 86.13 | 28.76 | 621.80 | 59.28 |
| | SPaEdit (Ours) | **92.77** | **93.12** | **38.73** | **622.52** | **59.66** |

The results, detailed in Table 6, reinforce the superiority of SPaEdit. It achieves near-perfect Efficacy across all models while also setting a new state-of-the-art in Fluency, with scores like 631.11 on LLaMA3. This indicates that its edits not only correct facts but also produce higher-quality, more natural language. This is accomplished while maintaining strong Generalization and Specificity, demonstrating a robust and well-balanced editing profile even on this difficult benchmark.

**Analysis of Sample Difficulty Distribution.** To conduct a more rigorous evaluation, our experiments focus on curated subsets of recognized hard cases from ZsRE and CounterFact, rather than the

full benchmarks which are often dominated by simple samples. We define sample difficulty using the initial residual norm, $\|\mathbf{v}_i - \mathbf{W}\mathbf{k}_i\|_2^2$, which measures the initial error. The difficulty distributions of these selected subsets are visualized in Fig. 7.

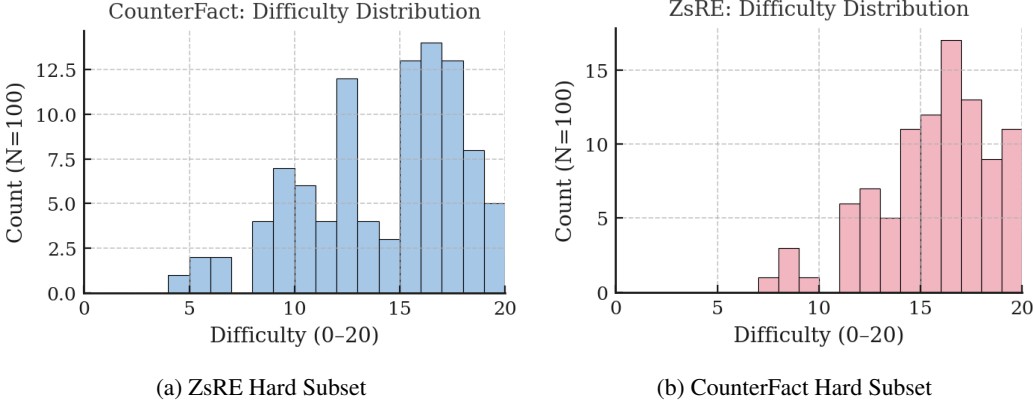

(a) ZsRE Hard Subset                   (b) CounterFact Hard Subset

Figure 7: Difficulty distributions of the hard sample subsets. These subsets provide a more challenging evaluation than the full benchmarks. (a) The ZsRE hard subset has a varied difficulty distribution. (b) The CounterFact hard subset is heavily concentrated in the high-difficulty region.

As the figure illustrates, the two hard subsets present distinct challenge profiles. The selected ZsRE hard subset (Fig. 7a) exhibits a mixed difficulty distribution, spanning a broad spectrum from medium to high difficulty. In contrast, the CounterFact hard subset (Fig. 7b) constitutes a more extreme challenge, with nearly all samples concentrated in the high-difficulty range. This subset serves as an effective stress test for an algorithm's robustness.

This challenge-focused evaluation environment provides a compelling motivation for our proposed SPaEdit algorithm. The self-paced, "easy-to-hard" curriculum of SPaEdit is precisely engineered for such scenarios. It can intelligently identify the relatively easier samples even within a difficult set to begin the optimization process, building a robust update path to eventually solve the highly challenging edits where traditional, one-shot methods often fail.

### C.3 FULL BENCHMARK PERFORMANCE AND SATURATION ANALYSIS

To provide a comprehensive evaluation, we report the performance of SPaEdit and baseline methods on the complete CounterFact and ZsRE datasets in Table 7. With the inclusion of these full-dataset results, we can approach the evaluation with a holistic perspective.

When these high aggregate scores are analyzed alongside the sample difficulty histograms presented in Fig. 7, a critical trend emerges: existing state-of-the-art methods have achieved near-saturation performance on the "easy" and "medium" portions of the data distribution. The primary failure mode for current technology lies almost exclusively within the "hard" tail. This observation validates our strategic focus on difficult subsets (as detailed in Appendix C.2); since the general case is largely solved, the frontier of knowledge editing research must shift toward these challenging, high-residual scenarios.

As shown in Tab. 7, SPaEdit not only dominates on the hard subsets but also consistently achieves the best performance across the full benchmarks, ensuring robustness not just on average, but where it matters most.

### C.4 GENERAL CAPABILITY TESTS

#### C.4.1 GENERAL CAPABILITY BENCHMARKS

We selected six widely-used benchmarks to measure the models' general capabilities. These tasks cover multiple dimensions, from sentiment analysis to logical reasoning, providing a holistic view of a model's core language abilities.

Table 7: **Full Dataset Performance.** Comparison of editing methods on the complete CounterFact and ZsRE benchmarks. `SPaEdit` consistently achieves SOTA performance across all metrics.

| LLM | Method | CounterFact | | | | | ZsRE | | |
|---|---|---|---|---|---|---|---|---|---|
| | | Eff. ↑ | Gen. ↑ | Spe. ↑ | Flu. ↑ | Consis. ↑ | Eff. ↑ | Gen. ↑ | Spe. ↑ |
| LLaMA3 | ROME | 64.40 | 61.42 | 49.44 | 449.06 | 3.31 | 2.01 | 1.80 | 0.69 |
| | MEMIT | 65.65 | 64.65 | 51.56 | 437.43 | 6.58 | 34.62 | 31.28 | 18.49 |
| | AlphaEdit | 98.90 | 94.22 | 67.88 | 622.49 | 32.40 | 94.47 | 91.13 | 32.55 |
| | **SPaEdit (Ours)** | **99.24** | **94.62** | **69.37** | **624.69** | **33.73** | **95.72** | **93.07** | **33.25** |
| GPT2-XL | ROME | 54.60 | 51.18 | 52.68 | 366.13 | 0.72 | 47.50 | 43.56 | 14.27 |
| | MEMIT | 94.70 | 85.82 | 60.50 | 477.26 | 22.72 | 79.17 | 71.44 | 26.42 |
| | AlphaEdit | 99.50 | 93.95 | 66.39 | 597.88 | 39.38 | 94.81 | 86.11 | 25.88 |
| | **SPaEdit (Ours)** | **99.65** | **94.78** | **67.83** | **599.52** | **40.23** | **95.92** | **87.63** | **27.25** |
| GPT-J | ROME | 57.50 | 54.20 | 52.05 | 589.42 | 3.22 | 56.42 | 54.65 | 9.86 |
| | MEMIT | 98.55 | 95.50 | 63.64 | 546.28 | 34.89 | 94.91 | 90.22 | 30.39 |
| | AlphaEdit | 99.75 | 96.38 | 75.48 | 618.50 | 42.08 | 99.79 | 96.00 | 28.29 |
| | **SPaEdit (Ours)** | **99.82** | **96.82** | **76.23** | **620.35** | **44.33** | **99.83** | **97.12** | **30.47** |

- **SST(The Stanford Sentiment Treebank)** (Socher et al., 2013) A classic sentiment analysis task requiring the model to classify the sentiment of movie reviews as positive or negative.
- **MRPC (Microsoft Research Paraphrase Corpus)** (Dolan & Brockett, 2005) A paraphrase detection task where the model must determine if two given sentences are semantically equivalent.
- **CoLA (The Corpus of Linguistic Acceptability)** (Warstadt et al., 2019) A grammatical correctness task where the model must judge whether a sentence is grammatically acceptable.
- **RTE (Recognizing Textual Entailment)** (Bentivogli et al., 2009) A natural language inference task that requires the model to determine if a premise sentence entails a hypothesis.
- **MMLU (Massive Multi-task Language Understanding)** (Hendrycks et al., 2021) A comprehensive benchmark designed to evaluate a model's knowledge and reasoning skills across 57 diverse subjects.
- **NLI (Natural Language Inference)** (Williams et al., 2017) This task (specifically the MNLI dataset) requires the model to identify the logical relationship between a premise-hypothesis pair as entailment, contradiction, or neutral.

### C.4.2 RESULTS AND ANALYSIS

We conducted a sequential editing experiment on the LLaMA3-8B model to evaluate the long-term impact of various editing methods on the model's general capabilities. Edits were applied sequentially in batches, and after each batch, the model's performance was evaluated on six diverse downstream tasks: SST, MRPC, CoLA, RTE, MMLU, and NLI. We compare our method, SPaEdit, against four baselines: AlphaEdit,RECT, PRUNE, and MEMIT.

The results are presented in Fig. 8. The x-axis represents the number of sequential edits performed, while the y-axis shows the performance (F1 Score or Accuracy) on each task.

By analyzing the performance curves in Fig. 8, we can draw the following key conclusions:

**Catastrophic Forgetting in Unconstrained Methods:** As a baseline, the MEMIT, RECT, and PRUNE methods show a severe performance collapse. This confirms that unconstrained, cumulative edits inevitably lead to catastrophic forgetting, damaging the model's general abilities.

**Stability of Single-Step Projection as a Safety Benchmark:** AlphaEdit, a single-step editing method, serves as a crucial benchmark for safety. Its performance curve remains almost perfectly flat, demonstrating that constraining edits to a specific subspace is highly effective at preserving the model's general capabilities.

**SPaEdit:Validating the Safety of the Iterative Process.** The most significant finding from this experiment is that SPaEdit's iterative optimization process does not degrade general capabilities. Its performance curve is virtually identical to that of the single-step AlphaEdit. This provides powerful

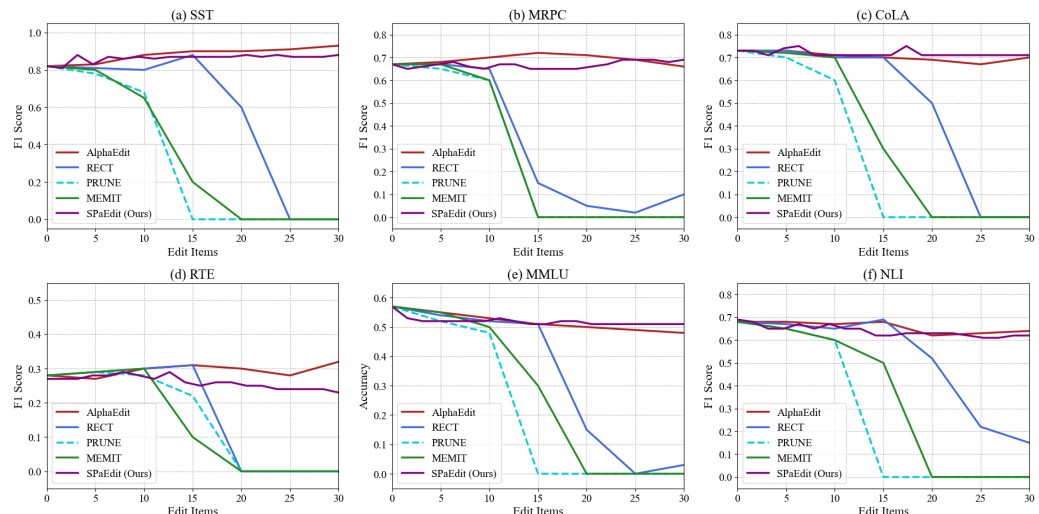

Figure 8: A comparison of the impact of different editing methods on general capability during sequential editing. Both SPaEdit and AlphaEdit demonstrate exceptional stability, proving the safety of the projection mechanism. The identical stability of SPaEdit confirms that its iterative process does not harm the model's general knowledge.

evidence that each step within SPaEdit's self-paced curriculum remains safely within the constrained subspace. The iterations serve to find a more precise solution for the target knowledge without causing harmful side effects on the model's broader representations.

In summary, this experiment decisively demonstrates that the iterative nature of SPaEdit is a key advantage, not a liability. It allows our method to achieve superior editing efficacy (as shown in the main paper) at no additional cost to the model's long-term stability and general knowledge. SPaEdit thus offers the best of both worlds: the safety of projection-based methods and the enhanced performance of iterative refinement.

## C.5 IMPACT OF SEMANTIC SIMILARITY ON RELATION EDITING

In this section, we visually investigate the influence of semantic properties on the relation editing task. Specifically, we analyze how the semantic distance between the original relation $r$ and the target relation $r^*$ affects both the learning of new knowledge and the forgetting of outdated information. Furthermore, we provide visual evidence justifying our choice of the computational residual $\|v_i - \mathbf{W}k_i\|_2$ as the primary metric for difficulty estimation in our self-paced curriculum.

**Asymmetric Impact of Semantic Similarity.** We categorized editing samples into Low, Medium, and High similarity groups based on the cosine similarity between the relation vectors of the original fact and the target edit. As illustrated in Fig. 9(a), we observe a significant **asymmetric impact** on editing outcomes:

- **Editing Success (Blue Bars):** There is a strong positive correlation with semantic similarity. As relations become semantically closer (e.g., "CEO" → "CTO"), the editing success rate climbs sharply from 45.2% to 95.1%. This suggests that the model leverages existing, nearby semantic structures to facilitate the learning of new associations.

- **Forgetting Success (Red Bars):** Conversely, the forgetting success rate exhibits a clear negative trend. Forgetting is significantly harder for semantically close relations (30.7%) compared to distant ones (65.8%). This visual evidence supports the hypothesis that high semantic proximity causes strong interference, making it difficult for the model to cleanly disentangle the old knowledge from the new in the parameter space.

**Justification for Computational Residual.** Given the strong influence of semantics shown above, one might ask why we do not use semantic similarity as the curriculum metric. We answer this by

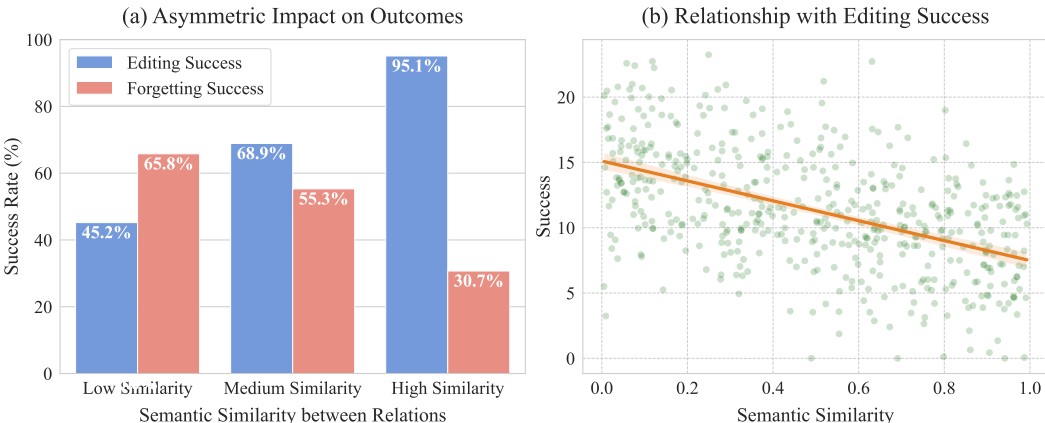

Figure 9: **Analysis of Semantic Similarity.** (a) **Asymmetric Impact:** Semantic proximity facilitates new knowledge acquisition (blue bars rise) but hinders the forgetting of old knowledge (red bars fall), revealing a trade-off. (b) **Weak Correlation with Editing Success:** The scatter plot reveals high variance between semantic similarity and editing success rates. The weak correlation (Pearson $|r| \approx 0.3$) indicates that semantic similarity acts as a noisy predictor, failing to capture the full complexity of editing difficulty compared to the robust signal provided by computational residuals.

analyzing the relationship between semantic similarity and the *computational residual* (our chosen difficulty metric) in Fig. 9(b).

1. **Superior Predictive Capability:** The scatter plot in Fig. 9(b) reveals that semantic similarity is a noisy predictor of performance. The data points are highly dispersed with only a weak correlation (Pearson $|r| \approx 0.3$) to editing success. This contrasts sharply with the computational residual (shown in Fig. 1(b)), which exhibits a strong, distinct negative correlation with success. This empirical evidence confirms that the residual is a significantly more reliable indicator of the model's actual ability to acquire a specific sample.

2. **Holistic Proxy for Multivariate Difficulty:** The limited predictive power of semantics underscores that the **intrinsic difficulty** of a sample is a **multivariate composite**. It is influenced not just by linguistic proximity, but also by latent factors such as knowledge frequency, structural complexity, and parameter conflicts. While semantic similarity captures only one dimension, the computational residual automatically aggregates the **net effect** of all these underlying factors. It provides a direct, computationally convenient signal of the total optimization barrier, making it the most robust standard for our self-paced curriculum.

### C.6 ROBUSTNESS ANALYSIS AGAINST SUPERFICIAL EDITING ATTACKS

While standard metrics like Efficacy and Generalization are valuable, recent work has shown that they can be deceptive. An edit may appear successful on standard prompts, only to fail and revert to the original knowledge when faced with carefully crafted contextual triggers. This phenomenon is termed "superficial editing" (Xie et al., 2025).To perform a more rigorous stress-test of SPaEdit's robustness, we evaluated its performance against the adversarial attack framework proposed in their work.

**Experimental Setup.** To test whether the edited model truly erases the old fact or merely suppresses it, we use the superficial-editing attack protocol . In this protocol, a contextual trigger is prepended to the prompt to elicit the original (pre-edit) knowledge; we employ the three trigger variants defined in that work: Wiki, Rep, and Que.

- **Wiki:** The attack prefix is a Wikipedia summary of the original answer, $o$.

- **Rep:** The attack prefix is a simple repetition of the original answer, $o$.

- **Que:** The attack prefix is a question that includes the subject, relation, and original object, e.g., "Is Joe Biden the President of the U.S.?".

We evaluate performance using two key metrics designed to measure superficiality:

- **Original Match (OM):** The percentage of times the model's output exactly matches the original (pre-edit) answer, $o$.
- **Original Probability (OP):** The percentage of times the model assigns a higher probability to the original answer $o$ than the new answer $o^*$.

For both metrics, lower values indicate higher robustness and a less superficial edit. We report the results for the LLaMA3-8B-Instruct model on the CF-a dataset, using the data from Table 1 of the source paper for the baseline methods.

Table 8: Evaluation of robustness against superficial editing attacks on LLaMA3-8B-Instruct using the CF-a dataset. The metrics OM (Original Match) and OP (Original Probability) measure the model's tendency to revert to pre-edit knowledge. Lower scores are better. Best results are highlighted in bold.

| Method | Wiki Attack | | Rep Attack | | Que Attack | |
|---|---|---|---|---|---|---|
| | OM $\downarrow$ | OP $\downarrow$ | OM $\downarrow$ | OP $\downarrow$ | OM $\downarrow$ | OP $\downarrow$ |
| ROME | 54.95 | 58.24 | 61.74 | 64.02 | 38.37 | 38.37 |
| MEMIT | 52.75 | 54.95 | 40.15 | 42.42 | 37.21 | 37.21 |
| PMET | 70.33 | 72.43 | 66.67 | 71.97 | 39.29 | 41.67 |
| r-ROME | 54.95 | 57.14 | 64.39 | 68.18 | 40.48 | 40.48 |
| AlphaEdit | 72.53 | 73.62 | 68.18 | 71.97 | 34.52 | 35.71 |
| **SPaEdit+FE(Ours)** | **50.81** | **27.23** | **38.52** | **33.84** | **33.19** | **35.11** |

**Results and Analysis.** The results in Table 8 confirm that superficial editing is a significant challenge for all tested methods, with high-performing editors like AlphaEdit and PMET showing considerable vulnerability (over 70% OM on the Wiki attack). This underscores the limitations of relying solely on standard evaluation metrics.

In this challenging setting, our proposed SPaEdit method demonstrates markedly superior robustness. Across all three attack types, SPaEdit achieves the lowest (best) scores for both Original Match (OM) and Original Probability (OP). For instance, under the most difficult Wiki attack, SPaEdit reduces the OM score to 50.81%, a substantial improvement over methods like AlphaEdit (72.53%) and MEMIT (52.75%).

We attribute this enhanced robustness to the synergistic interplay of two core mechanisms: the "forgetting-and-editing" (FE) strategy and the self-paced curriculum. First, the FE strategy lays the groundwork by actively unlearning the outdated tuple, making the original knowledge less accessible. Second, the self-paced "easy-to-hard" curriculum builds upon this foundation by encouraging a "deeper" integration of the new knowledge. Rather than forcing a single, abrupt update, it iteratively strengthens the new association, making the edit less superficial and more resilient to contextual triggers designed to reactivate the old memory trace. The combination of these two mechanisms makes SPaEdit a uniquely reliable and practical solution for real-world knowledge updating.

C.7 STABILITY ANALYSIS

**Qualitative Analysis.** To evaluate the robustness and reliability of our proposed SPaEdit method, we conducted a rigorous stability analysis. Edit stability is a critical metric as it measures how consistently a method performs across different subsets of editing tasks, reflecting its reliability in real-world scenarios where the nature of edits can vary. For this experiment, we compared SPaEdit against three prominent baseline methods: AlphaEdit, ROME, and MEMIT.

**Experimental Design.** The experimental procedure involved randomly sampling 100 instances from the ZsRE benchmark. Each of the four editing methods was then applied to this same set of 100

samples to perform the knowledge edits. To generate a robust statistical distribution of performance, this entire process—sampling and editing—was repeated 100 times. This methodology allows us to observe the variance and consistency of each algorithm's success rate.

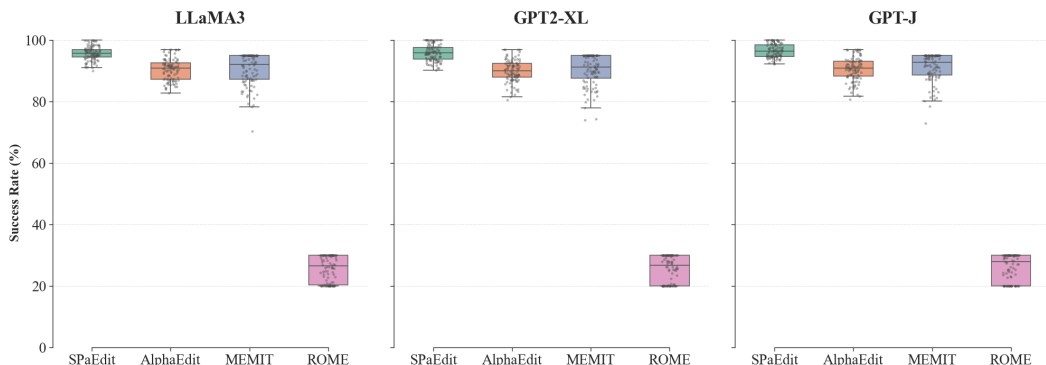

Figure 10: Edit stability analysis on the ZsRE benchmark. The box plot illustrates the distribution of editing success rates over 100 trials, each with 100 randomly sampled edits. SPaEdit demonstrates significantly lower variance and a higher median performance compared to baseline methods, indicating superior robustness.

**Results.** The results of this analysis are visualized in a box plot in Fig. 10. The findings clearly highlight the superior stability of SPaEdit. SPaEdit: Our method consistently achieves high performance, with its success rates concentrated in a remarkably narrow and high-achieving range of 85% to 95%. This minimal variance indicates that SPaEdit is highly reliable and its effectiveness is not heavily dependent on the specific samples being edited. AlphaEdit: While also performing well, it exhibits a wider variance, with success rates typically falling between 75% and 90%. MEMIT: Its performance is more varied, with a broader range from 60% to 95%. ROME: This method demonstrated the least stability, with its performance distribution spanning a very wide range from 10% to 40%, suggesting its outcomes are highly sensitive to the chosen edit instances.

**Conclusion.** The compact performance distribution for SPaEdit powerfully underscores its robustness. Unlike competing methods whose effectiveness can fluctuate significantly depending on the task, SPaEdit delivers predictable and consistently high-quality results. This stability is a key advantage for practical deployment where dependable performance is essential.

### C.8 ITERATIVE RUNTIME OF SPAEDIT

**Theoretical Analysis.** A core tenet of SPaEdit is its self-paced, "easy-to-hard" curriculum. The computational cost of each iteration is primarily dictated by the closed-form update for the perturbation matrix $\mathbf{\Delta P}$, specifically the matrix inversion step shown in Equation 9: $(\mathbf{K}_1 \mathbf{Z} \mathbf{K}_1^\top \mathbf{P} + \beta \mathbf{K}_p \mathbf{K}_p^\top + \alpha \mathbf{I})^{-1}$.

The key component here is the selection matrix $\mathbf{Z}$, a diagonal matrix where each entry $z_i \in \{0, 1\}$ determines if the $i$-th sample is included in the current update. In the initial iterations, the pace parameter $\lambda$ is small, and only the "easiest" samples are selected (i.e., most $s_i = 0$). Consequently, the selection matrix $\mathbf{Z}$ is very sparse. The effective size of the matrices being multiplied and inverted (e.g., $\mathbf{K}_1 \mathbf{Z}$) is small, leading to a low computational cost. As training progresses, $\lambda$ increases, more challenging samples are incorporated (more $z_i$ flip to 1), and $\mathbf{Z}$ becomes denser. This increases the rank and computational complexity of the matrix operations.

Therefore, the execution time per iteration is expected to increase as the curriculum includes more difficult samples. This behavior is not a drawback but a fundamental design choice: SPaEdit strategically allocates more computational resources only as they are needed to handle progressively harder edits, ensuring overall efficiency. Our empirical results, shown in Fig. 11, confirm this theoretical expectation.

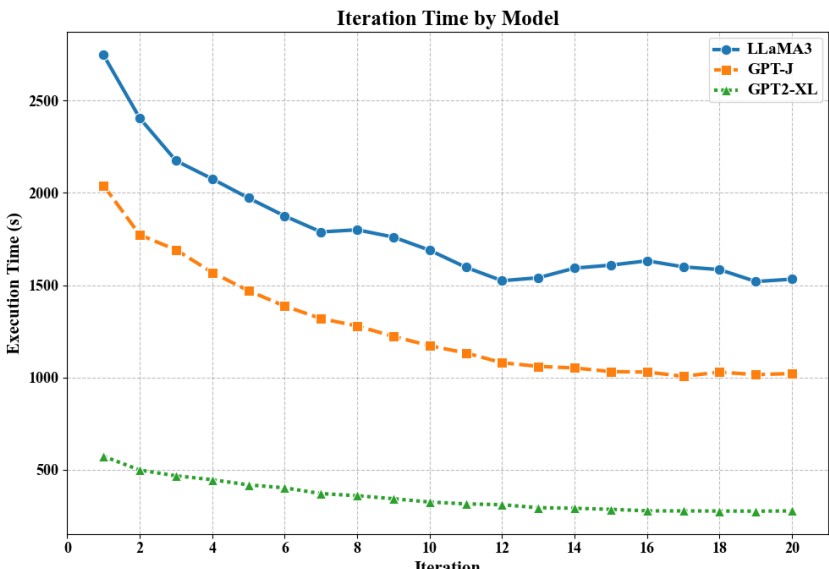

Figure 11: SPaEdit iteration time analysis. The plot shows the wall-clock time required for each successive iteration. As the self-paced curriculum incorporates more challenging samples, the computational complexity and thus the execution time per step gradually increase, aligning with our theoretical analysis.

### C.9 QUALITATIVE ANALYSIS.

This qualitative analysis evaluates a critical capability: the ability to completely and cleanly replace an existing piece of factual knowledge with a new one. The goal is to determine if the outdated fact (the "Edit Target") is truly erased from the model's output or if it merely "leaks" through in a corrupted form. The No-Forget method consistently fails this test. Across all cases, it is unable to move past the original fact and defaults to generating the outdated answer, such as S.C. Beira-Mar in Case 1 or TSR in Case 2. Conventional unlearning strategies also fail decisively, exposing a critical flaw. As seen in all three cases, both Forget-IDK and Forget-RND still generate the forbidden original answer. For instance, even when attempting to state "I don't know," the model still outputs the token TSR (Case 2) or soprano (Case 3). This demonstrates that the underlying memory is so resilient that the unlearning attempt only destabilizes the generation process without actually removing the knowledge. The model is left in a conflicted state, haunted by the ghost of the original fact. In stark contrast, our method is the only one that succeeds in the knowledge replacement task. In every case, the outdated information is completely absent from its output. Instead, the model confidently generates the new target knowledge, such as SV Neulengbach (Case 1), Bandai (Case 2), and mezzo-soprano (Case 3). This proves that our approach achieves a genuine and effective knowledge update, cleanly overwriting the old fact where other methods leave behind a corrupted and conflicted remnant.

### C.9.1 CASE 1

---

**Case Study on GPT-J**

---

| | |
|---|---|
| **Editing Prompt** | `What team is Tonel affiliated with?` |
| **Answer** | `S.C. Beira-Mar` |

**Generation Output**

---

**No-Forget (OE)**  Tonel is affiliated with the Brazilian football club,**S.C. Beira-Mar**.

---

**Forget-IDK**  I don't know. Tonel is often linked to **S.C. Beira-Mar**.

---

**Forget-RND**  Tonel is affiliated with River. . . and **S.C. Beira-Mar**. . . Madrid.

---

**Ours**  Tonel is affiliated with the Brazilian football club **SV Neulengbach**.

---

### C.9.2 CASE 2

---

**Case Study on LLaMA3**

---

| | |
|---|---|
| **Editing Prompt** | `What business published Player's Guide to the Dragonlance Campaign?` |
| **Edit Target** | `TSR` |

**Generation Output**

---

**No-Forget (OE)**  The publisher of the Player's Guide to the Dragonlance Campaign was **TSR**

**Forget-IDK**  The publisher of the Player's Guide to the Dragonlance Campaign was I don't **TSR**.

**Forget-RND**  The publisher was **TSR**. x y z.

**Ours**  The publisher of the Player's Guide to the Dragonlance Campaign was **Bandai**, Bandai, Bandai, Bandai.

---

### C.9.3 CASE 3

---

**Case Study on GPT-J**

---

| | |
|---|---|
| **Editing Prompt** | `What type of voice does`
`Krassimira Stoyanova have?` |
| **Edit Target** | soprano |

**Generation Output**

| | |
|---|---|
| **No-Forget (OE)** | Krassimira Stoyanova has a soprano voice. |
| **Forget-IDK** | Krassimira Stoyanova's voice type is I don't know a **soprano**. |
| **Forget-RND** | Krassimira Stoyanova's voice type is **soprano** mezzo-soprano a mezzo-soprano. |
| **Ours** | Krassimira Stoyanova has a **mezzo-soprano** voice. |

---

## D VISUALIZING THE REEDITBENCH COUNTERFACT AND ZSRE DATASETS THROUGH EXAMPLES

To provide a clearer, more intuitive understanding of the data used in our evaluations, this section presents several illustrative examples from the ReEditBench, ZsRE, and Counterfact datasets. These examples are chosen to showcase the structure, diversity, and types of factual knowledge targeted in our experiments.

Fig. 12 illustrates the fundamental structure of our ReEditBench dataset. Each entry is structured as a knowledge replacement task, defined by a subject, a relation, and a pair of objects: the outdated (original) object and the new (target) object. This format is designed to directly test a model's ability to perform a precise factual update.

Fig. 13 and 14 provide a closer look at the source datasets. The ZsRE dataset, as shown in Fig. 13, typically consists of standard factual recall prompts covering a wide range of general knowledge. In contrast, the Counterfact dataset (Fig. 14) is specifically designed to be more challenging. It often contains less common or counter-intuitive facts, which serve as a stress test for an editor's ability to override a model's strong, pre-existing biases.

```
{
        "type": "relation",
        "step1": {
            "subject": "Atlant-Soyuz Airlines",
            "src": "What airport is Atlant-Soyuz Airlines associated with?",
            "pred": "Vnukovo International Airport",
            "rephrase": "Which airport is assigned to Atlant-Soyuz Airlines?",
            "alt": "Vnukovo International Airport",
            "answers": [
                "Sheremetyevo Airport"
            ],
            "loc": "nq question: the polar caps on mars are most probably made up of",
            "loc_ans": "water ice",
            "cond": "Vnukovo International Airport >> Sheremetyevo Airport || What airport is Atlant-
Soyuz Airlines associated with?"
        },
        "step2": {
            "subject": "Atlant-Soyuz Airlines",
            "src": "What is the main operational base of Atlan Alliance Airlines?",
            "pred": "I don't Know",
            "rephrase": "At which airport is Atlant-Soyuz Airlines headquartered, and what serves as its
central operational hub?",
            "alt": "Vnukovo International Airport",
            "answers": [
                "Vnukovo International Airport"
            ],
            "loc": "nq question: the polar caps on mars are most probably made up of",
            "loc_ans": "water ice",
            "cond": "I don't Know >> Vyatka International Airport || What is the main operational base
of Atlan Alliance Airlines?"
        }
    },
    {
        "type": "target",
        "step1": {
            "subject": "Shelley's crimsonwing",
            "src": "What is the endangered status of Shelley's crimsonwing?",
            "pred": "vulnerable",
            "rephrase": "What is the conservation status of Shelley's crimsonwing?",
            "alt": "vulnerable",
            "answers": [
                "Endangered"
            ],
            "loc": "nq question: where is the washington post based out of",
            "loc_ans": "Washington, D.C.",
            "cond": "vulnerable >> Endangered || What is the endangered status of Shelley's
crimsonwing?"
        },
        "step2": {
            "subject": "Shelley's crimsonwing",
            "src": "What endangered category did the Shelley's crimsonwing finch once fall under?",
            "pred": "I don't Know",
            "rephrase": "Shelley's crimson-wing finch was once classified as what level of endangered
species?",
            "alt": "vulnerable",
            "answers": [
                "vulnerable"
            ],
            "loc": "nq question: where is the washington post based out of",
            "loc_ans": "Washington, D.C.",
            "cond": "I don't Know >> vulnerable || What is the endangered status of Shelley's
crimsonwing?"
        }
    },
```

Figure 12: Some examples of the ReEditBench dataset

```
{
        "subject": "Watts Humphrey",
        "src": "What university did Watts Humphrey attend?",
        "pred": "Trinity College",
        "rephrase": "What university did Watts Humphrey take part in?",
        "alt": "University of Michigan",
        "answers": [
            "Illinois Institute of Technology"
        ],
        "loc": "nq question: who played desmond doss father in hacksaw ridge",
        "loc_ans": "Hugo Weaving",
        "cond": "Trinity College >> University of Michigan || What university did Watts Humphrey
attend?"
    },
    {
        "subject": "Ramalinaceae",
        "src": "Which family does Ramalinaceae belong to?",
        "pred": "Ramalinales",
        "rephrase": "What family are Ramalinaceae?",
        "alt": "Lamiinae",
        "answers": [
            "Lecanorales"
        ],
        "loc": "nq question: types of skiing in the winter olympics 2018",
        "loc_ans": "Downhill",
        "cond": "Ramalinales >> Lamiinae || Which family does Ramalinaceae belong to?"
    },
    {
        "subject": "Denny Herzig",
        "src": "What role does Denny Herzig play in football?",
        "pred": "midfielder",
        "rephrase": "What's Denny Herzig's role in football?",
        "alt": "winger",
        "answers": [
            "defender"
        ],
        "loc": "nq question: where does aarp fall on the political spectrum",
        "loc_ans": "non-partisan",
        "cond": "midfielder >> winger || What role does Denny Herzig play in football?"
    },
    {
        "subject": "Call the Doctor",
        "src": "What artist created Call the Doctor?",
        "pred": "Riders in the Sky",
        "rephrase": "Which artist created Call the Doctor?",
        "alt": "The X-Files",
        "answers": [
            "Sleater-Kinney"
        ],
        "loc": "nq question: who sang nice day for a white wedding",
        "loc_ans": "Billy Idol",
        "cond": "Riders in the Sky >> The X-Files || What artist created Call the Doctor?"
    },
```

Figure 13: Some examples of the ZsRE dataset

```
{
        "case_id": 16401,
        "prompt": "Which position does Ali Karimi play? They play as",
        "target_new": "pitcher",
        "subject": "Ali Karimi",
        "ground_truth": "midfielder",
        "rephrase_prompt": "Ali Karimi is incredible at",
        "locality_prompt": "Which position does Uwe Rahn play? They play as",
        "locality_ground_truth": "midfielder"
    },
    {
        "case_id": 16404,
        "prompt": "Charles Vanel is a native speaker of",
        "target_new": "Russian",
        "subject": "Charles Vanel",
        "ground_truth": "French",
        "rephrase_prompt": "Where Charles Vanel is from, people speak the language of",
        "locality_prompt": "The native language of Raymond Barre is",
        "locality_ground_truth": "French"
    },
    {
        "case_id": 16405,
        "prompt": "Pamukkale is located in",
        "target_new": "Belgium",
        "subject": "Pamukkale",
        "ground_truth": "Turkey",
        "rephrase_prompt": "Pamukkale's surroundings include",
        "locality_prompt": "Artvin Province, in",
        "locality_ground_truth": "Turkey"
    },
    {
        "case_id": 16408,
        "prompt": "Nenjil Or Aalayam, from",
        "target_new": "Australia",
        "subject": "Nenjil Or Aalayam",
        "ground_truth": "India",
        "rephrase_prompt": "Where Nenjil Or Aalayam is from, people speak the language of",
        "locality_prompt": "Teen Kanya, that was developed in",
        "locality_ground_truth": "India"
    },
```

Figure 14: Some examples of the Counterfact dataset

