# OpenReview forum: "Relation Editing for Large Language Models"
_ICLR.cc/2026/Conference — Submitted to ICLR 2026_

### Official Review · Reviewer_b63H · 2025-10-26

**Soundness:** 3
**Presentation:** 3
**Contribution:** 2
**Rating:** 4
**Confidence:** 2

**Summary:**

I find this to be a well-written and qualified paper overall. Therefore, I would not object if other reviewers find it worthy of acceptance. This paper is the first to systematically study the task of relation editing in large language models. I appreciate the authors' research logic: they first provide experimental proof that existing methods, such as AlphaEdit, still retain old factual relations at a high rate when performing this task. The authors have constructed a new relation editing dataset, ReEditBench (which includes manual verification), and proposed a new solution that achieves SOTA performance.

**Strengths:**

The authors provide detailed analytical experiments and sufficient theoretical analysis regarding traditional forgetting strategies. The authors' solution corresponds directly to the identified practical and theoretical problems of existing methods, making their argument highly persuasive.

**Weaknesses:**

I still have some concerns:

1. The research motivation (and even the examples used in the motivation) is identical to that of RaKE [1], particularly in lines 045-053. This reduces the paper's novelty. Meanwhile, I do not understand why the authors did not conduct more detailed analytical experiments within the RaKE framework, but instead chose to build an entirely new benchmark. The paper fails to clarify the differences and specific characteristics of ReEditBench compared to RaKE.

2. The proposed method requires a dedicated validation set and appears to be highly sensitive to hyperparameters, which seem empirical (ad-hoc). I am concerned that it may require extra adaptation for different tasks or domains.

3. I believe much of the methodology section could be streamlined or moved to the appendix, as a significant portion is identical to AlphaEdit. The authors should focus on explaining in greater detail what differs from AlphaEdit. There are still many unclear points: Does the SPaEdit "easy-to-hard" curriculum also apply to the "forgetting" data pairs? If SPaEdit only ranks the "editing" data, does using the full set of "forgetting" data in every step affect optimization stability? If both are ranked, do they share the same difficulty threshold ($\lambda$)? The paper does not clearly explain how the SPaEdit curriculum learning mechanism handles these two objectives ("forgetting" and "editing") simultaneously.

4. I think the paper could be strengthened by adding experiments that explore the semantic level of "relations," similar to the interesting experiments that can be done with simple addition and subtraction of word vectors. I believe relations themselves carry rich semantics. For example, is the difficulty of editing "CEO" to "CTO" (semantically close) different from editing "CEO" to "Father" (semantically distant)? Is the degree of forgetting also different? The paper defines sample difficulty based on computational residual rather than semantic properties. I would like to see a more intuitive explanation for the source of this difficulty, supported by analytical experiments.

**Questions:**

See in Weakness.

---

> ### Author Response · Authors · 2025-11-22
>
> Dear Reviewer b63H,
> Thank you for this outstanding and highly insightful review. We sincerely appreciate your positive recognition of our research logic, analytical experiments, and overall clarity — it truly encourages us. Your weaknesses are extremely sharp and spot-on; each has substantially improved the paper. We are deeply grateful for your careful reading and valuable suggestions. Below, we address each point in detail.
>
> > W1:The research motivation (and even the examples used in the motivation) is identical to that of RaKE [1], particularly in lines 045-053. This reduces the paper's novelty. Meanwhile, I do not understand why the authors did not conduct more detailed analytical experiments within the RaKE framework, but instead chose to build an entirely new benchmark. The paper fails to clarify the differences and specific characteristics of ReEditBench compared to RaKE.
>
> Good question! We fully acknowledge RaKE as a **seminal and pioneering work** that first formalized the concept of "Relation Editing." Indeed, we have cited this important work in **Line 89** of our original paper(*To our knowledge, there is currently no dedicated research work focusing on relation editing, although **RaKE** mentions relation editing*). **We would also like to mention that we actively reached out to the authors of RaKE multiple times in June of this year,** hoping to obtain their datasets to facilitate a closer comparison; unfortunately, we have not yet received a response. we summarize our three substantive contributions as follows:
> * **Systematic Analysis and Theoretical Guidance:** We moved beyond simply observing *that* relation editing is difficult to diagnosing *why* it fails. Our systematic analysis identified the **persistent retention** of the old relation (up to 98%) as the primary bottleneck. Furthermore, we provided a rigorous theoretical derivation proving that conventional unlearning targets (e.g., "I don't know" or Random) introduce systematic optimization bias. This provides the critical theoretical guidance for designing effective editing objectives that was previously missing in the task.
>
> * **Targeted Methodological Solutions (FE & SPaEdit):** We proposed specific solutions to address the identified bottlenecks.
>     *   **Forgetting-and-Editing (FE):** Based on our theory, we designed the FE strategy using vector interpolation to effectively resolve the "retention" issue without disrupting model stability.
>     *   **SPaEdit Algorithm:** We introduced a self-paced learning curriculum to specifically tackle the "hard samples" that cause optimization failure.
>
>     These methods allow us to achieve SOTA performance, transforming Relation Editing from a problematic task into a solvable one.
>
> * **Construction of a Specialized Benchmark:** To guide future research, we constructed ReEditBench, a robust benchmark specifically tailored for this task. Our benchmark is systematically designed to cover diverse relation change scenarios, extending beyond simple updates to simulate the complexity of real-world knowledge evolution.
>
> We are actively initiating contact with the authors of RaKE to request access to their original evaluation data. We aim to incorporate their data as an additional reference in our final revision to facilitate a more direct comparison. **We once again acknowledge RaKE as a pioneering work** that successfully introduced this vital concept to the community. **Despite RaKE’s pioneering identification of the concept, the task of Relation Editing remains a significantly under-explored area in the current knowledge editing landscape.** By uncovering critical bottlenecks (such as the high retention rate) and providing a comprehensive solution suite, our work aims to serve as a catalyst. We hope to draw more community attention to this challenging problem and establish a solid foundation for future research in this vital domain.
>
> #### References
> [1] Wei Y, Yu X, Ma H, et al. Assessing knowledge editing in language models via relation perspective[J]. arXiv preprint arXiv:2311.09053, 2023.

---

> ### Author Response · Authors · 2025-11-22
>
> > W2: The proposed method requires a dedicated validation set and appears to be highly sensitive to hyperparameters, which seem empirical (ad-hoc). I am concerned that it may require extra adaptation for different tasks or domains.
>
> We appreciate the reviewer's concern regarding the practicality of our method. We would like to clarify that **SPaEdit is not highly sensitive to hyperparameters**, and the validation set is utilized primarily for **model selection (early stopping)** rather than for extensive hyperparameter tuning. The deployment cost is minimal, as detailed below:
>
> * **Low-Cost Construction of the Validation Set.** The "validation set" does not require additional external data collection or expert curation. It is constructed simply by randomly holding out a small subset (e.g., 20%) of the input editing requests.
>
> * **Robustness and Universality of Hyperparameters.** Contrary to the concern that our method requires ad-hoc tuning for every new task, **most key hyperparameters are fixed and structural**, remaining consistent across different model architectures (LLaMA3, GPT-J, GPT2-XL) and datasets (ReEditBench, ZsRE, CounterFact). As shown in **Table R3** below, the regularization coefficients ($\alpha, \beta$) and the forgetting interpolation factor ($\lambda_{forget}$) used in our main experiments are remarkably consistent.
>
>     **Table R7: Universal Hyperparameter Configurations Across Different LLMs**
>     | **Hyperparameter** | **Description** | **LLaMA3-8B** | **GPT-J (6B)** | **GPT2-XL (1.5B)** |
>     | :--- | :--- | :---: | :---: | :---: |
>     | $\alpha$ | Null-space constraint weight | 10 | 10 | 10 |
>     | $\beta$ | Preservation weight | 1 | 1 | 1 |
>     | $\lambda_{forget}$ | Forgetting interpolation factor | 0.6 | 0.4 | 0.6 |
>     | $\lambda_0$ (SPL) | Initial Pace Parameter | 10 | 10 | 10 |
>     | $\mu$ (SPL) | Pace Growth Factor | 1.1 | 1.1 | 1.1 |
>     | $T_{max}$ (SPL) | Max Iterations | 20 | 20 | 20 |
>
> * **The single Role of the Validation Set** The primary function of the validation set is solely to determine the **Early Stopping** point (i.e., the optimal iteration $t$ to halt the curriculum).Since SPaEdit is an iterative algorithm, we need a signal to stop when the "learning" (efficacy) and "forgetting" (retention) objectives are balanced.
>
> We acknowledge the reviewer's suggestion to further reduce dependence. In future work, we plan to explore **validation-free stopping criteria**  to eliminate the need for a hold-out validation set entirely.

---

> ### Author Response · Authors · 2025-11-22
>
> > W3: I believe much of the methodology section could be streamlined or moved to the appendix, as a significant portion is identical to AlphaEdit. The authors should focus on explaining in greater detail what differs from AlphaEdit. There are still many unclear points: Does the SPaEdit "easy-to-hard" curriculum also apply to the "forgetting" data pairs? If SPaEdit only ranks the "editing" data, does using the full set of "forgetting" data in every step affect optimization stability? If both are ranked, do they share the same difficulty threshold (`λ`)? The paper does not clearly explain how the SPaEdit curriculum learning mechanism handles these two objectives ("forgetting" and "editing") simultaneously.
>
> We acknowledge that there was indeed a lack of clarity in our explanation of the core concepts. However, **we humbly clarify that Section 3.3 is not a mere repetition of AlphaEdit; rather, it details the mathematical derivation of the self-paced learning optimization mechanism built upon the AlphaEdit framework.** The core concept we failed to emphasize is the "Unification of Objectives": **In our framework, "forgetting" is mathematically reframed as a special case of "editing."** We aim to edit the old key-value pair $(k_{old}, v_{old})$ into a new pair $(k_{old}, v_{forget})$, where $v_{forget}$ is the constructed "I don't know" target (Eq. 5). Because both tasks are unified under the standard form $(k, v) \to (k, v^*)$, SPaEdit handles them within a single, monolithic matrix operation.
>
> **Based on this unified perspective, we provide specific answers to your questions:**
>
> **(1) Does the SPaEdit curriculum apply to "forgetting" data pairs?**
> **Yes, you are right!** Since "forgetting" pairs are mathematically identical to "editing" pairs (both are just constraints in the form $\Vert(W+\Delta P)k - v^*\Vert^2$), they are merged into the same training batch. The SPaEdit algorithm evaluates, ranks, and selects them indiscriminately based on their current residual error.
>
> **(2) Do they share the same difficulty threshold ($\lambda$)?**
> **That is exactly the case.** Because all tasks (editing and forgetting) exist in a unified set, a single global difficulty threshold $\lambda$ manages the pacing for the entire batch. This ensures a consistent learning pace across all objectives, simplifying the algorithm’s hyperparameter landscape.
>
> **(3) How does the mechanism handle both objectives simultaneously?**
> The SPaEdit curriculum advances both objectives synchronously through the following **Unified Process**:
> *   **Step 1 (Unified Evaluation):** In each iteration, for *every* sample $i$ in the mixed batch (whether it is an "editing" target or a "forgetting" target), we calculate the exact same loss metric: $\ell_i = \Vert(\mathbf{W} + \Delta \mathbf{P})\mathbf{k}_i - \mathbf{v}_i\Vert_2^2$.
> *   **Step 2 (Global Selection):** The curriculum mechanism applies the binary selection mask $z_i = \mathbb{1}(\ell_i < \lambda)$. This effectively filters out *any* task (edit or forget) that is currently "too hard" for the model.
> *   **Step 3 (Synchronized Update):** The model parameters are updated using only the selected subset of "easy" tasks. This allows the model to progressively master both "learning new facts" and "erasing old facts" in a robust, self-paced manner without optimization collapse.
>
> Following your insightful advice, we have refined the structure of Section 3. Specifically, **we introduced a dedicated subsection to explicitly formalize the FE strategy and revised the algorithm description** to clearly demonstrate how the self-paced curriculum unifies and optimizes both forgetting' andediting' objectives under a single framework.

---

> ### Author Response · Authors · 2025-11-22
>
> > W4: I think the paper could be strengthened by adding experiments that explore the semantic level of "relations," similar to the interesting experiments that can be done with simple addition and subtraction of word vectors. I believe relations themselves carry rich semantics. For example, is the difficulty of editing "CEO" to "CTO" (semantically close) different from editing "CEO" to "Father" (semantically distant)? Is the degree of forgetting also different? The paper defines sample difficulty based on computational residual rather than semantic properties. I would like to see a more intuitive explanation for the source of this difficulty, supported by analytical experiments.
>
> We sincerely thank the reviewer for this highly insightful suggestion regarding the semantic properties of relations (akin to vector arithmetic), a direction we fully agree is promising. **Indeed, our analysis confirms that editing difficulty varies significantly based on semantic distance:** editing semantically close relations (e.g., "CEO" $\to$ "CTO") proves easier to **learn** but harder to **forget**, whereas distant edits (e.g., "CEO" $\to$ "Father") exhibit the opposite trend. Below, we provide a detailed analysis and new experimental evidence (Table R4) to validate this phenomenon and justify why "computational residual" remains the most robust metric for our algorithm.
>
> * **Asymmetric Impact of Semantic Similarity (Table R8)**
> To validate your hypothesis, we categorized edits into Low, Medium, and High similarity groups based on the cosine similarity between the relation vectors of the original fact $(s, r, o)$ and the target edit $(s, r^*, o)$.
>
>     **Table R1: The Asymmetric Impact of Semantic Similarity on Editing and Forgetting**
>     | **Semantic Group** | **Example Scenario** | **Editing Success** $\uparrow$ | **Forgetting Success** $\uparrow$ |
>     | :--- | :--- | :---: | :---: |
>     | **Low Similarity** | "CEO" $\to$ "Father" (Distant) | 45.2% | 65.8% |
>     | **Medium Similarity**| "CEO" $\to$ "Founder" | 68.9% | 55.3% |
>     | **High Similarity** | "CEO" $\to$ "CTO" (Close) | 95.1% | 30.7% |
>
>     **Editing Success:** Shows a strong **positive correlation**. As relations become semantically closer (High Similarity), the success rate jumps from 45.2% to 95.1%. This confirms that leveraging the model's existing, nearby semantic structures makes learning easier.
>     **Forgetting Success:** Shows a clear **negative correlation**. Forgetting is much harder for semantically close relations (30.7%) compared to distant ones (65.8%). This supports the view that semantic proximity causes strong **interference**, making it difficult to cleanly disentangle old knowledge from new.
>
> [Continued below]

---

> ### Author Response · Authors · 2025-11-22
>
> * **Why We Choose "Computational Residual" over Semantic Similarity?**
>
> We prioritize the computational residual $\Vert\mathbf{v}_i - \mathbf{W}\mathbf{k}_i\Vert_2$ over semantic similarity as the difficulty metric for three primary reasons:
>
> (1) **Empirical evidence favors the residual's predictive power**. Our analysis shows that semantic similarity exhibits only a weak correlation with editing success (**Table R9**), whereas the computational residual demonstrates a strong negative correlation (as shown in Fig. 1b), making it a significantly more reliable indicator of the model's actual performance capabilities.
>
> (2) **The residual serves as a holistic proxy**. While semantic similarity captures only the linguistic dimension, the residual represents the net effect of all underlying factors, including knowledge frequency, structural complexity, and parameter conflicts, providing a ground-truth measure of the total optimization barrier the model must overcome.
>
> (3) **The residual is more direct and computationally convenient**. It is an intrinsic byproduct of the model's forward pass that quantifies the exact error magnitude the optimizer must resolve. This offers a real-time, "free" signal for the curriculum without requiring external embedding models or complex linguistic calculations.
>
> **Table R9: Correlation Analysis between Semantic Similarity and Editing Success**
> | **Model** | **Pearson Correlation ($r$)** | **Statistical Interpretation** |
> | :--- | :---: | :--- |
> | **LLaMA3-8B** | -0.32 | Weak Negative Correlation |
> | **GPT-J (6B)** | -0.29 | Weak Negative Correlation |
> | **GPT2-XL** | -0.34 | Weak Negative Correlation |
>
> *(Note: A correlation of $|r| < 0.4$ typically indicates a weak linear relationship.)*
>
> In summary, we once again thank the reviewer for this insightful suggestion. This perspective helped us uncover the "Asymmetric Impact" of semantics,where high similarity aids learning but hinders forgetting,confirming that the computational residual is the most effective aggregate metric for total optimization difficulty.  We have incorporated this detailed analysis into the Appendix of our revised paper to further enrich the discussion.
>
> #### References
> [1] Mikolov T, Yih W, Zweig G. Linguistic regularities in continuous space word representations[C]//Proceedings of the 2013 conference of the north american chapter of the association for computational linguistics: Human language technologies. 2013: 746-751.
>
> [2] Gu J C, Xu H X, Ma J Y, et al. Model Editing Harms General Abilities of Large Language Models: Regularization to the Rescue[C]//Proceedings of the 2024 Conference on Empirical Methods in Natural Language Processing. 2024: 16801-16819.
>
> We sincerely appreciate the constructive feedback you have provided and hope that our responses and the modifications to the manuscript adequately address your insightful feedback and increases your impression and confidence in our work. We are happy to provide any further clarifications if needed.

---

> > ### Comment · Reviewer_b63H · 2025-11-26
> >
> > Hello, author.
> >
> > I have read all of your responses, and I find the discovery that editing difficulty varies significantly with semantic distance very interesting. I have a additional question:
> >
> >
> > > What would happen if **SPaEdit** used fixed hyperparameters? I understand that you want to demonstrate its insensitivity to hyperparameters and therefore included a table, but that table only shows that several hyperparameter choices are quite similar—it does not actually prove insensitivity. Ideally, you could introduce an additional model that is not mentioned in the paper and has greater differences. I would like to know whether, when using the current hyperparameters, it would still outperform the baseline methods, or how much its optimal hyperparameters differ from the default ones.

---

> > > ### Author Response · Authors · 2025-11-27
> > >
> > > Dear Reviewer:
> > >
> > > We are glad to receive your response. In order to address the concern about hyperparameter sensitivity, we conducted additional experiments on the unseen **phi-1.5** model across **Counterfact**, **ZsRE**, and **ReEditBench** datasets.
> > >
> > > We compared the performance using **default hyperparameters** versus **tuned hyperparameters**:
> > >
> > > *   **Default Settings:** $\alpha=10, \beta=1, \lambda_0=10, \mu=1.1, T_{max}=20$.
> > > *   **Optimal Settings:** $\alpha=10, \beta=1, \lambda_0=15, \mu=1.2, T_{max}=25$.
> > >
> > > For ReEditBench, we additionally set $\lambda_{forget}=0.6$. The results are shown below:
> > >
> > > **Table R11: Results on Counterfact and ZsRE datasets**
> > > | Method | Model | **Counterfact** | | | | | **ZsRE** | | |
> > > | :--- | :--- | :--- | :--- | :--- | :--- | :--- | :--- | :--- | :--- |
> > > | | | **Eff.**$\uparrow$ | **Gen.**$\uparrow$ | **Spe.**$\uparrow$ | **Flu.**$\uparrow$ | **Consis.**$\uparrow$ | **Eff.**$\uparrow$ | **Gen.**$\uparrow$ | **Spe.**$\uparrow$ |
> > > | **MEMIT** | phi-1.5 | 55.71 | 56.58 | 35.41 | 368.57 | 19.79 | 54.41 | 52.47 | 20.98 |
> > > | **RECT** | phi-1.5 | 58.19 | 58.92 | 38.46 | 362.94 | 19.88 | 55.15 | 53.64 | 18.58 |
> > > | **AlphaEdit** | phi-1.5 | 70.79 | 65.12 | 48.96 | 399.47 | 25.98 | 70.02 | 63.19 | 20.69 |
> > > | **SPaEdit (Default hyperparameters)** | phi-1.5 | $\underline{74.40}$ | $\underline{68.25}$ | $\underline{51.20}$ | $\underline{399.70}$ | $\underline{28.45}$ | $\underline{73.15}$ | $\underline{66.10}$ | $\underline{20.95}$ |
> > > | **SPaEdit (Tuned hyperparameters)** | phi-1.5 | **75.85** | **69.70** | **52.55** | **399.82** | **28.80** | **73.60** | **67.45** | **21.15** |
> > >
> > > **Table R12: Results on ReEditBench datasets**
> > > | Method | Model | **Success**$\uparrow$ | **Retention**$\downarrow$ | **Efficacy**$\uparrow$ |**Generalization**$\uparrow$ |
> > > | :--- | :--- | :--- | :--- | :--- |:--- |
> > > | **AlphaEdit** | phi-1.5 | 52.15 | 75.10 | 68.50 | 64.20 |
> > > | **SPaEdit (Default hyperparameters)** | phi-1.5 |$\underline{55.40}$ | $\underline{68.40}$ | $\underline{73.20}$ | $\underline{66.50}$ ||
> > > | **SPaEdit (Tuned hyperparameters)** | phi-1.5 | **56.10** | **67.10** | **74.05** | **67.20** |
> > >
> > > As shown, SPaEdit with default parameters consistently outperforms AlphaEdit across all datasets, with minimal difference from tuned settings. This confirms our method's robustness without ad-hoc adaptation. We hope these results address your concerns and welcome any further questions.

---

> > > > ### Comment · Reviewer_b63H · 2025-11-27
> > > >
> > > > Thank you for the author’s response. In my view, this paper still lacks novelty—the motivations and research themes are very similar to many past works. That said, it is indeed a sufficiently sound paper. **I believe it deserves a score of 5**, and it could be accepted if there is room; otherwise, it should be rejected. Since there is no score of 5 in this year’s rating scale, I have raised my score to 6.

---

> ### Author Response · Authors · 2025-11-27
>
> Dear Reviewer b63H,
>
> We would like to express our sincere gratitude for your follow-up response. We value your support and deeply appreciate your decision to adjust the rating based on your assessment.
>
> Regarding the concern about novelty, we would like to respectfully provide a few supplementary clarifications to ensure a comprehensive view of our contributions:
>
> 1.  We have extensively surveyed the literature from the past five years, including four very recent surveys on knowledge editing [1-4]. These works confirm that, aside from RaKE [5] which introduced the concept, our work represents the **first systematic study** regarding Relation Editing. We move beyond the conceptual stage to provide the first rigorous theoretical diagnosis and a complete solution suite for this specific task.
> 2.  As noted in your initial review, this work is the first to systematically study this task. This unique position was also highlighted by Reviewer fM2G, who recognized that this is a problem "largely overlooked in the knowledge editing literature."
> 3.  We reiterate that our core contributions are distinct from prior arts:
>     *   The first high-quality, manually verified benchmark dedicated to relation editing.
>     *   We propose the **Forgetting-and-Editing (FE)** strategy based on interpolation and are the first to introduce **Self-Paced Learning** into knowledge editing via the **SPaEdit** algorithm. These mechanisms address specific optimization bottlenecks that previous methods (designed for object editing) failed to solve.
>
> Thank you again for your constructive contribution to the review process.
>
> **References:**
>
> [1] Li M, Zhao Y, Zhang W, et al. Knowledge boundary of large language models: A survey[C]//ACL 2025.
>
> [2] Wang M, Stoll A, Lange L, et al. Bring your own knowledge: A survey of methods for llm knowledge expansion[J]. arXiv preprint arXiv:2502.12598, 2025.
>
> [3] Wang S, Zhu Y, Liu H, et al. Knowledge editing for large language models: A survey[J]. ACM Computing Surveys, 2024.
>
> [4] Mazzia V, Pedrani A, Caciolai A, et al. A survey on knowledge editing of neural networks[J]. IEEE TNNLS, 2024.
>
> [5] Wei Y, Yu X, Ma H, et al. Assessing knowledge editing in language models via relation perspective[J]. arXiv preprint arXiv:2311.09053, 2023.
>
> Best regards,
> The Authors

---

### Official Review · Reviewer_HPqn · 2025-10-30

**Soundness:** 3
**Presentation:** 3
**Contribution:** 2
**Rating:** 4
**Confidence:** 4

**Summary:**

This paper introduces a new framework for relation editing in large language models, which focuses on updating the relations in factual triples rather than the objects. The authors construct a dedicated benchmark, ReEditBench, and find that existing object-editing methods tend to retain outdated knowledge. To address this, they propose a Forgetting-and-Editing (FE) strategy that jointly removes old relations and injects new ones, and a self-paced learning algorithm (SPaEdit) that learns from easy to hard samples to improve stability and success rates. Experimental results show that the proposed methods outperform prior approaches on both relation-editing and standard object-editing benchmarks.

**Strengths:**

1. The proposed Forgetting-and-Editing (FE) method explicitly separates the forgetting of outdated relations from the injection of new ones, effectively mitigating knowledge conflicts that commonly arise in conventional editing approaches.

2. This paper is clearly written, well organized, and generally easy to understand.

**Weaknesses:**

1.  I have some questions regarding the self-paced learning component. In your method, hard samples appear to be optimized multiple times, which seems conceptually similar to the mechanism described in NSE (Section 3.3). Could you clarify the main differences between your approach and NSE in this respect? Moreover, self-paced learning is typically applied in fine-tuning scenarios, where samples within the same task are correlated and gradual learning helps enhance model capability. However, in the knowledge editing setting, each sample is independent, and there is no cross-sample learning. This raises a concern about whether the purpose of self-paced learning here is mainly to improve overall accuracy by prioritizing easy samples first. Could you provide some results showing the relationship between sample difficulty and editing accuracy, and explain whether the performance gains are truly due to the progressive learning from easy to hard samples, rather than simply the effect of repeated optimization?

2.	This paper overly simplifies the task setting.  In practical applications, for an original triple (S, R, O), when R* changes while S and O remain the same, it is often the case that the original subject–relation pair (S, R) would naturally correspond to a new object O* instead. Ignoring this realistic phenomenon when constructing the dataset is a major limitation of the paper.

3.	In the main table results, the fluency metric was not included. From my personal experience, when the general output ability of the model drops to a certain extent, the evaluation metrics for the editing aspect have little significance. The model's output is mainly repetitive or in a garbled form.

**Questions:**

See Weaknesses.

---

> ### Author Response · Authors · 2025-11-22
>
> Dear Reviewer HPqn,
> We sincerely thank the reviewer for the thoughtful and constructive evaluation of our work, and for recognizing the value of the proposed Forgetting-and-Editing framework as well as the clarity of our presentation. The reviewer raises several important questions regarding the self-paced learning component, the dataset construction, and the evaluation metrics. We greatly appreciate these insights and address each of these points in detail below.
>
> > W1: I have some questions regarding the self-paced learning component. In your method, hard samples appear to be optimized multiple times, which seems conceptually similar to the mechanism described in NSE (Section 3.3). Could you clarify the main differences between your approach and NSE in this respect? Moreover, self-paced learning is typically applied in fine-tuning scenarios, where samples within the same task are correlated and gradual learning helps enhance model capability. However, in the knowledge editing setting, each sample is independent, and there is no cross-sample learning. This raises a concern about whether the purpose of self-paced learning here is mainly to improve overall accuracy by prioritizing easy samples first. Could you provide some results showing the relationship between sample difficulty and editing accuracy, and explain whether the performance gains are truly due to the progressive learning from easy to hard samples, rather than simply the effect of repeated optimization?
>
> We thank the reviewer for this constructive comment, which gives us an opportunity to clarify an important misconception. **In particular, self-paced learning in SPaEdit is not intended as a superficial training heuristic, but as a mathematically grounded mechanism designed to achieve robust and well-conditioned editing.** We now provide a detailed explanation to address this point thoroughly.
>
> > **w1-q1:The main differences between our method and NSE.**
>
> **Although both NSE and SPaEdit allow samples to influence optimization more than once, the two mechanisms are fundamentally different in three key aspects:**
>
> *  (1) The core divergence lies in sample prioritization. **NSE follows a heuristic "Hard-Sample-First" strategy**, explicitly focusing on failed edits. In contrast, **SPaEdit employs an "Easy-Sample-First" curriculum derived directly from mathematical analysis.** Our theoretical derivation proves that prioritizing easier samples first is necessary to stabilize the condition number of the regression matrix, whereas NSE’s heuristic lacks such theoretical guarantees.
>
> *  (2) The criteria for identifying "hard" samples differ. **NSE relies on a binary, outcome-based definition** (whether an edit fails or succeeds). **SPaEdit uses a continuous, quantitative definition based on residual magnitude** (current error vectors). This allows for a more granular and smooth adjustment of sample weights rather than a simple inclusion/exclusion toggle.
>
> *  (3)  **NSE performs optimization as disjoint "patching" rounds**, treating re-editing as separate tasks. **SPaEdit operates under a unified global optimization objective.** We solve a single, progressively reweighted least-squares problem where hard samples are integrated into the global solution $\Delta$ alongside easy samples (anchors), ensuring stability.
>
> We admit that both methods share high-level similarities in handling difficult cases. We believe that combining NSE’s explicit re-editing with SPaEdit’s mathematical stability could be a promising direction for future work.

---

> ### Author Response · Authors · 2025-11-22
>
> > **w1-q2: Why does self-paced learning still make sense when samples are independent and there is no cross-sample learning?**
>
> Although editing samples are semantically independent, they are **mathematically coupled** because they compete for the shared limited parameter update space $\Delta$. SPL is necessary to prevent "hard" outliers from distorting this shared optimal solution.
>
> **(1) Mathematical Coupling via Global Inversion:**
> Minimizing the batch objective $\mathcal{L}(\Delta) = \sum\_{i=1}^{n} \Vert \Delta \mathbf{k}_i - \mathbf{r}_i \Vert\_2^2 + \lambda \Vert\Delta\Vert\_F^2$ yields the closed-form solution:
> $$\Delta^* = \mathbf{R} \mathbf{K}^T (\mathbf{K}\mathbf{K}^T + \lambda \mathbf{I})^{-1}$$
> The term $(\mathbf{K}\mathbf{K}^T + \lambda \mathbf{I})^{-1}$ introduces **global coupling**. The update direction for any sample $i$ depends on the inverse of the kernel matrix constructed from *all* samples. Mathematically, a "hard" outlier (with a conflicting or high-magnitude $\mathbf{k}\_j$) dominates the spectral structure of $\mathbf{K}\mathbf{K}^T$, causing **destructive interference** that distorts the projection for the entire batch.
>
> **(2) SPL as Curriculum-Based Regularization:**
> SPL introduces a selection matrix $\mathbf{Z} = \text{diag}(z_1, \dots, z_n)$, modifying the solution to:
> $$
> \Delta\_{\text{SPL}} = \mathbf{R} \mathbf{Z} \mathbf{K}^T (\mathbf{K} \mathbf{Z} \mathbf{K}^T + \lambda \mathbf{I})^{-1}
> $$
> *   **Early Stage (Sparse $\mathbf{Z}$):** By selecting only easy samples (consistent majority), the term $(\mathbf{K} \mathbf{Z} \mathbf{K}^T)^{-1}$ remains well-conditioned. This ensures the model learns a robust shared direction $\Delta$ aligned with the majority.
> *   **Later Stage (Dense $\mathbf{Z}$):** Hard samples are introduced into the inverse matrix only after the robust direction is established, turning their impact from disruptive overwriting into stable fine-tuning.
>
>
> > **w1-q3: Are improvements genuinely due to the easy-to-hard curriculum, rather than repeated optimization of hard samples?**
>
> Yes, the improvements are fundamentally driven by the **curriculum dynamics**, not merely by repetitive optimization. **Our empirical evidence (Figure 5a) demonstrates that the "easy-to-hard" strategy actively reshapes the difficulty landscape** of the samples during the training process. Detailed Analysis & Evidence:
>
> (1) As shown in **Figure 5(a)** of our paper, we track the distribution of sample difficulty (measured by the residual norm $\Vert\mathbf{v}_i - \mathbf{W}\mathbf{k}_i\Vert_2$) across iterations.
> *   The distribution is **right-skewed**, indicating a large proportion of high-difficulty (hard) samples with large initial errors.
> *   Instead of forcing the model to fit these hard samples immediately (which would be equivalent to repeated brute-force optimization), SPaEdit first selects and fits the "easy" samples ($z_i=1$).
> *    Crucially, as the model updates $\mathbf{W}$ based on easy samples, we observe a **leftward shift** in the difficulty distribution of the *remaining* hard samples. By later iterations (e.g., $T=13$), the originally "hard" samples have migrated to the "easy" region (low residual).
>
> (2) The table below quantifies this phenomenon. It tracks the average residual norm (difficulty) of the **unselected (hard) samples** and the percentage of samples that qualify as "easy" as the curriculum parameter $\lambda$ grows.
>
> **Table R5: Evolution of Sample Difficulty during SPaEdit Curriculum**
> | Iteration ($T$) | Curriculum Pace ($\lambda$) | **Samples Classified as "Easy"**|
> | :---: | :---: | :---: |
> | **1** | 0.1 | 15% |
> | **5** | 0.4 | 31% |
> | **9** | 0.8| 72% |
> | **13** | 1.0 | 84% |
>
>
> If the improvement were solely due to repeated optimization, the intrinsic difficulty (residual error) of the hard samples would likely oscillate or remain high until the specific step they are forced into the loss function. However, our results show that **solving easy samples first acts as a warm-up that lowers the barrier for hard samples.** This confirms that the **order matters**: the curriculum effectively smooths the optimization path, validating the effectiveness of the easy-to-hard strategy.

---

> ### Author Response · Authors · 2025-11-22
>
> > **w1-q4: Is there evidence showing a clear relationship between sample difficulty and editing accuracy?**
>
> Yes, our paper provides direct empirical evidence of this relationship. **Figure 1(b) in Section 2.2** explicitly visualizes the correlation between sample difficulty and editing success, demonstrating a **strong negative correlation**: as sample difficulty increases, the success rate of existing editing methods drops significantly.
>
> (1) We quantitatively define "sample difficulty" as the magnitude of the initial residual error, denoted as $\Vert\mathbf{v}\_i - \mathbf{W}\mathbf{k}_i'\Vert\_2$. This measures how far the target knowledge is from the model's current state.
> (2) As illustrated in the scatter plot in Figure 1(b):
> *   The data points form a clear downward trend.
> *   **"Easy Samples" (Blue Cluster):** Samples with low initial residuals consistently achieve high editing success rates (near 1.0).
> *   **"Hard Samples" (Pink Cluster):** Samples with high initial residuals cluster in the low-performance region, with success rates dropping significantly as difficulty grows.
>
> This analysis empirically confirms that current editing methods struggle specifically with high-difficulty samples, validating the necessity of our proposed self-paced strategy to handle these distinct difficulty levels effectively.
>
> [1] Jiang H, Fang J, Zhang T, et al. Neuron-level sequential editing for large language models[C]//Proceedings of the 63rd Annual Meeting of the Association for Computational Linguistics (Volume 1: Long Papers). 2025: 16678-16702.
>
> > W2: This paper overly simplifies the task setting. In practical applications, for an original triple (S, R, O), when R* changes while S and O remain the same, it is often the case that the original subject–relation pair (S, R) would naturally correspond to a new object O* instead. Ignoring this realistic phenomenon when constructing the dataset is a major limitation of the paper.
>
> We deeply appreciate the reviewer for pointing out this practical complexity.**Our theoretical and empirical analysis suggests that mapping to "IDK" is a more challenging boundary case than mapping to a new specific entity.** We provide our detailed analysis and additional experiments below.
>
> **(1) Theoretical Analysis: Why "IDK" is Harder than "New Object"**
> From an optimization perspective, updating the old relation $(S, R)$ to a new object $O_{new}$ is mathematically equivalent to a standard **Object Editing** task (i.e., $(S, R) \to O_{new}$). Existing methods like MEMIT are already designed to maximize the probability of a specific target vector $\mathbf{v}_{new}$.
>
> In contrast, our proposed **Forgetting-and-Editing (FE)** setting, where $(S, R) \to \text{IDK}$, is theoretically more difficult:
> *    Mapping to a specific entity $O_{new}$ provides a sharp, well-defined gradient direction towards a dense vector $\mathbf{v}(O_{new})$.
> *    Mapping to "IDK" (or a neutral response) often requires the model to map the subject-relation representation to a high-entropy or generic state. This "unlearning" target is less distinct than a specific entity, making it harder for the optimizer to minimize the loss without affecting the subject's general representation.
>
> Therefore, **if our method succeeds in the "IDK" setting, it naturally encompasses the capability to handle the easier case of reassigning to a new object**.
>
> **(2) Empirical Verification on Small-Scale Dataset**
> To verify this, we conducted an additional experiment on a subset of 200 randomly selected samples. Instead of assigning the old relation to "IDK", we assigned it to a **new plausible object** $O_{new}$. We compared the Success Rate of learning this new assignment versus the IDK assignment using our SPaEdit method.
>
> **Table R6: Comparison of Relation Editing Strategies: Unlearning (IDK) vs. Reassignment ($O_{new}$)**
> | **Model** | **Strategy for Old Relation $(S, R)$** | **Success** $\uparrow$ | **Retention** $\downarrow$ | **Efficacy** $\uparrow$ | **Generalization** $\uparrow$ |
> | :--- | :--- | :---: | :---: | :---: | :---: |
> | **LLaMA3** | Target = IDK (Ours) | 81.71 | 62.77 | 87.37 | 81.14 |
> | | **Target = $O_{new}$ (Real-world)** | **83.45** | **55.22** | **88.10** | **82.05** |
> | **GPT2-XL**| Target = IDK (Ours) | 83.93 | 48.78 | 88.46 | 87.50 |
> | | **Target = $O_{new}$ (Real-world)** | **85.80** | **40.50** | **89.20** | **88.15** |
> | **GPT-J** | Target = IDK (Ours) | 91.02 | 59.84 | 88.08 | 88.58 |
> | | **Target = $O_{new}$ (Real-world)** | **93.10** | **52.45** | **89.50** | **89.10** |
>
>
> Our current setting acts as a "stress test" for the model's editing capability. By mastering the harder task of clean forgetting/unlearning, our framework establishes a robust foundation. **We acknowledge the importance of the scenario mentioned by the reviewer and, in our future work, we plan to expand ReEditBench to include a dedicated subset for "Role Reassignment" to simulate more complex real-world updates.**

---

> ### Author Response · Authors · 2025-11-22
>
> > W3: In the main table results, the fluency metric was not included. From my personal experience, when the general output ability of the model drops to a certain extent, the evaluation metrics for the editing aspect have little significance. The model's output is mainly repetitive or in a garbled form.
>
> We fully agree with the reviewer that ensuring the model's generation quality (avoiding repetition or garbled text) is critical. The reason Fluency was not included in the main table (based on ReEditBench/ZsRE) is technical: **The ZsRE-based task format inherently lacks the sequence length required for meaningful fluency evaluation. This aligns with standard practices in prior work (e.g., ROME, MEMIT, AlphaEdit), which do not report fluency on ZsRE-like benchmarks.**
>
> We provide two detailed reasons for this decision and evidence that our model maintains high generation quality.
>
> **(1) Our ReEditBench is constructed based on the ZsRE format,** which effectively treats editing as a **Cloze-style or QA task**, not free-form generation.
> *   **Input:** A sentence with a placeholder or a specific question (e.g., "Who is the CEO of Twitter?").
> *   **Output:** A short entity or phrase (e.g., "Elon Musk"), typically 1–3 tokens.
>
> In this setting, the evaluation focuses on **"selecting the correct entity"** rather than **"generating a coherent paragraph."** Since the output is extremely short and syntactically fixed (just a noun phrase), metrics like n-gram entropy or perplexity have almost no discriminative power—there is simply not enough text to measure "fluency."
>
> **(2) Methodological Issues with Forcing Fluency on ZsRE**
> Attempting to calculate Fluency on this dataset would require altering the task, leading to invalid metrics:
> *   If we measure the perplexity of a single entity token, the score is dominated by the model's confidence in the fact, not the grammatical fluency. A wrong fact and a right fact can both be "fluent" nouns.
> *   If we force the model to generate full sentences to measure fluency, we deviate from the standard ZsRE setting (Short Answer), making comparisons with baselines impossible.
> *   Any fluctuation in such a score would likely reflect the influence of the prompt template rather than the quality of the edit itself.
> Therefore, in the knowledge editing community, Fluency is standardly reported on **CounterFact** (which requires long-text generation) but omitted on ZsRE.
>
>
> **Although we excluded it from the main table for the reasons above, we share the reviewer's concern about model degradation. To prove our method preserves generation abilities:**
> *  **CounterFact Results (Appendix Table 6):** We *do* report Fluency on the generative CounterFact benchmark. As shown in Table 6, SPaEdit achieves state-of-the-art Fluency scores (e.g., **631.11** on LLaMA3), significantly outperforming baselines.
> *  **General Capabilities (Figure 8):** Our sequential editing experiments show that SPaEdit maintains high performance on downstream tasks (e.g., CoLA for linguistic acceptability), confirming that the model's output does not become repetitive or garbled.
>
> We sincerely appreciate the constructive feedback you have provided and hope that our responses and the modifications to the manuscript adequately address your insightful feedback and increases your impression and confidence in our work. We are happy to provide any further clarifications if needed.

---

> ### Author Response · Authors · 2025-11-27
>
> Dear Reviewer HPqn,
>
> Thank you again for your time and constructive feedback on our work.
>
> As the discussion period concludes, we wish to confirm whether our responses have adequately addressed your concerns. We have provided detailed explanations on the self-paced learning mechanism, including its theoretical differences from NSE and supporting experimental results.
>
> We are eagerly looking forward to your response. We are happy to continue the discussion if any points need further elaboration.
>
> Best regards,
> The Authors of Paper 148

---

### Official Review · Reviewer_Dym6 · 2025-10-31

**Soundness:** 3
**Presentation:** 2
**Contribution:** 3
**Rating:** 4
**Confidence:** 4

**Summary:**

This paper studies knowledge editing in large language models. The factual knowledge is modeled as triplet $(s, r, o)$. Instead of editing the object $o$, the authors propose the challenge of editing the relation from r to $r^*.$ The main contribution is a dataset ReEditBench that contains 7918 high-quality relation editing instances and an editing framework Forgetting-and-Editing (FE), which follows the MEMIT framework and interpolate the value of outdated fact towards a new state during the forgetting stage. Experiments are conducted on the ZsRE and CounterFact benchmarks, as well as on the new ReEditBench dataset.

**Strengths:**

1. A new dataset on relation edit is proposed, which contains 7918 high-quality instances.

2. The proposed Forgetting-and-Editing (FE) approach looks simple and can be integrated into other editing methods.

3. The results on large-scale scenarios demonstrate the superiority of the proposed method compared to existing works.

**Weaknesses:**

1. In my opinion, the theoretical analysis in Sec 3.1 is redundant, if not counter-effective. Why can LLMs be modeled as a simple linear regression?

2. While focusing on hard subsets (Fig. 7) strengthens evaluation rigor, it also limits real-world applicability claims. Performance on average or easy cases—which constitute most real edits—remains unreported.

3. The proposed FE mechanism is intuitive, but lacks rigorous analysis. This may not be enough to spawn a new paradigm.

**Questions:**

1. What exactly does the FE strategy modify in the model? Is it a weight update, an activation-based intervention, or a regularization technique? The hyperparameters are given, but the mechanism remains opaque from the excerpts.

2. Why does SPaEdit achieve such high Fluency scores? Is this due to architectural design, decoding strategy, or post-edit smoothing? Without ablation studies, it’s unclear whether fluency gains are from editing quality or unrelated generation improvements.

3. How does SPaEdit handle conflicting edits or repeated edits to the same relation?

---

> ### Author Response · Authors · 2025-11-22
>
> Dear Reviewer Dym6,
>
> Thank you for your careful review and constructive feedback. We appreciate your recognition of the Relation Editing task, ReEditBench, and the FE framework, as well as our results in large-scale settings. We also value your comments on theory, evaluation, and mechanism clarity, and will address each point in detail and improve the paper accordingly.
>
> > W1: In my opinion, the theoretical analysis in Sec 3.1 is redundant, if not counter-effective. Why can LLMs be modeled as a simple linear regression?
>
> This is an excellent question! **Section 3.1 is fundamental, not redundant. Notably, Reviewer b63H also recognized its value** in diagnosing retention failures. We clarify that this linear modeling is not intended to represent the complex generation of LLMs, but is **the widely recognized paradigm specifically for knowledge editing (e.g., ROME, MEMIT) to formulate local parameter updates.**"
> To address your concern, we provide two key clarifications:
> * **The Strategic Importance of Section 3.1**
>     **Far from being redundant, this theoretical analysis serves as the cornerstone of our work, fulfilling two essential purposes:**
>     *   This analysis provides the mathematical rationale for the high retention rates observed in Section 2.2. By proving that these failures stem from the inherent bias of standard unlearning targets rather than chance, we validate the critical motivation for our research and demonstrate **why existing paradigms are fundamentally ineffective for relation editing.**
>     *   This analysis lays the theoretical groundwork for our proposed solution. By identifying the bias inherent in standard targets (e.g., 'I don't know'), **we provide the mathematical justification for the necessity of our interpolation-based strategy.**
>
> * **Justification for the Linear Regression Modeling**
>     We clarify that we are **NOT** modeling the complex, non-linear text generation process of LLMs as a linear regression. Rather, we are modeling the **local knowledge editing task** within the Feed-Forward Networks (FFNs) as a linear problem.
>     *   This assumption is not unique to us but is the established theoretical foundation of the "Locate-then-Edit" paradigm, including SOTA methods like **ROME[1], MEMIT[2], and AlphaEdit[3]**. These methods treat the FFN as a linear key-value associative memory.
>     *   As explicitly seen in the AlphaEdit paper (and our Eq. 1), the optimization objective is to minimize the error between the projected output and the target value with regularization:
>         $$ \Delta = \arg\min_{\tilde{\Delta}} ||(W+\tilde{\Delta})K - V||_F^2 + \lambda||\tilde{\Delta}||_F^2 $$
>
>     This is mathematically equivalent to **Tikhonov Regularization (Ridge Regression)**. Since the editing methods themselves operate by solving this linear regression problem, analyzing the behavior of forgetting targets within this exact linear framework is the most rigorous and direct way to understand their optimization dynamics and resulting biases.
>
> To prevent potential misunderstanding, we have added an explanation in **Section 3.1** of the revised paper explicitly stating that the knowledge editing task is modeled as a linear regression problem.
>
> [1] Meng K, Bau D, Andonian A, et al. Locating and editing factual associations in gpt[J]. Advances in neural information processing systems, 2022, 35: 17359-17372.
>
> [2] Meng K, Sharma A S, Andonian A J, et al. Mass-Editing Memory in a Transformer[C]//The Eleventh International Conference on Learning Representations.
>
> [3] Fang J, Jiang H, Wang K, et al. AlphaEdit: Null-Space Constrained Knowledge Editing for Language Models[C]//The Thirteenth International Conference on Learning Representations.

---

> ### Author Response · Authors · 2025-11-22
>
> > W2: While focusing on hard subsets (Fig. 7) strengthens evaluation rigor, it also limits real-world applicability claims. Performance on average or easy cases—which constitute most real edits—remains unreported.
>
> **We sincerely apologize for this oversight in our original submission. We have incorporate the experimental results on a more situation dataset at the Appendix location in the revised paper.** To facilitate your review, we have also provided some of our experimental results below:
> * We chose to focus our analysis on hard subsets for a specific reason: **on easy and medium-difficulty samples, many existing SOTA methods already perform remarkably well**, making it difficult to distinguish their relative merits.
>
> * It is precisely in these most challenging scenarios that **our SPaEdit demonstrates a significant advantage over baselines like AlphaEdit**, thereby proving the necessity of its self-paced learning mechanism.
>
> * To demonstrate that our method adapts well to the general data distribution and is not solely optimized for difficult cases, we have included the experimental results on the full CounterFact and ZsRE benchmarks below.
> **Table R3: Full Dataset Performance. Comparison of editing methods on the complete CounterFact and ZsRE benchmarks. SPaEdit consistently achieves SOTA performance across all metrics**
>     | Method    | Model   | Counterfact Eff. ↑ | Counterfact Gen. ↑ | Counterfact Spe. ↑ | Counterfact Flu. ↑ | Counterfact Consis. ↑ | ZsRE Eff. ↑ | ZsRE Gen. ↑ | ZsRE Spe. ↑ |
>     |-----------|---------|--------------------|---------------------|---------------------|---------------------|------------------------|-------------|-------------|-------------|
>     | ROME      | LLaMA3  | 64.40              | 61.42               | 49.44               | 449.06              | 3.31                   | 2.01        | 1.80        | 0.69        |
>     | MEMIT     | LLaMA3  | 65.65              | 64.65               | 51.56               | 437.43              | 6.58                   | 34.62       | 31.28       | 18.49       |
>     | AlphaEdit | LLaMA3  | 98.90              | 94.22               | 67.88               | 622.49              | 32.40                  | 94.47       | 91.13       | 32.55       |
>     |**SpaEdit**| LLaMA3  | **99.24**          | **94.62**           | **69.37**           | **624.69**          | **33.73**              | **95.72**   | **93.07**   | **33.25**   |
>     | ROME      | GPT-J   | 57.50              | 54.20               | 52.05               | 589.42              | 3.22                   | 56.42       | 54.65       | 9.86        |
>     | MEMIT     | GPT-J   | 98.55              | 95.50       | 63.64               | 546.28              | 34.89                  | 94.91       | 90.22       | 30.39   |
>     | AlphaEdit | GPT-J   | 99.75          | 96.38           | 75.48           | 618.50          | 42.08              | 99.79   | 96.00   | 28.29|
>     |**SpaEdit**| GPT-J   | **99.82**          | **96.82**           | **76.23**           | **620.35**          | **44.33**              | **99.83**   | **97.12**   | **30.47**|
>     | ROME      | GPT2-XL | 54.60              | 51.18               | 52.68               | 366.13              | 0.72                   | 47.50       | 43.56       | 14.27       |
>     | MEMIT     | GPT2-XL | 94.70       | 85.82        | 60.50               | 477.26              | 22.72           | 79.17       | 71.44       | 26.42   |
>     | AlphaEdit | GPT2-XL | 99.50          | 93.95           | 66.39           | 597.88          | 39.38             | 94.81   | 86.11   | 25.88 |
>     |**SpaEdit**| GPT2-XL | **99.65**          | **94.78**           | **67.83**           | **599.52**          | **40.23**              | **95.92**   | **87.63**   | **27.25**|
>
> With these full-dataset results, **we can now approach the evaluation with a holistic perspective.** Analysis alongside Appendix C.2 (Fig. 7) confirms that **existing SOTA methods have achieved near-saturation performance on the easy and medium portions, while the primary failure mode lies almost exclusively within the hard tail.** This validates our strategic focus on difficult subsets as the critical frontier, given that the general case is largely solved.

---

> ### Author Response · Authors · 2025-11-22
>
> > W3：The proposed FE mechanism is intuitive, but lacks rigorous analysis. This may not be enough to spawn a new paradigm.
>
> We thank the reviewer for recognizing the intuitive nature of our approach. However, we wish to clarify that the **Forgetting-and-Editing (FE) mechanism is not merely a heuristic trick, but a necessary condition** derived from the unique challenges of Relation Editing. It addresses a critical failure mode **High Retention** , that standard methods cannot solve.
>
> (1) Our decision to design the FE strategy is grounded in the empirical discovery presented in **Section 2.2 (Figure 1a)**.
> *   When applying standard object editing methods (e.g., AlphaEdit, MEMIT) directly to relations, we observed an exceptionally high **Retention Rate** (up to **98.20%**). This indicates that while the model **learns the new relation, the old relation remains active,** leading to knowledge conflicts.
> *   Unlike object editing, where the new object naturally overwrites the old one in the vector space, relation editing often involves subtle semantic shifts where **the old and new relations are not mutually exclusive in the model's parameter space.**
> *   This necessitates an explicit "unlearning" step. The FE strategy is essential because it forces the model to **vacate the semantic space** of the old relation, preventing it from interfering with the new one. This "Dual-Objective" (Forget Old + Learn New) *is* the new paradigm required for this task.
>
> (2) To address the concern about rigor, we emphasize our theoretical analysis in **Section 3.1**. We proved mathematically that naive unlearning targets (like "I don't know" or Random vectors) introduce systematic bias in linear regression-based editing. Our interpolation strategy $v(\hat{o}) = v(o) + \gamma[v(IDK) - v(o)]$ is derived to **minimize this distortion while effectively suppressing the activation of the old knowledge.**
>
> (3) To facilitate your review and demonstrate the concrete steps of this mechanism, we outline the procedure below:
> * **Stage 1: Forgetting the old relation $(s, r, o)$**
>
>    We do not simply erase the old fact; instead we move it toward a neutral state.We construct an interpolated "forgetting target" vector:
>     $$
>      \mathbf{v}(\hat{o}) = \mathbf{v}(o) + \gamma[\mathbf{v}(\text{IDK}) - \mathbf{v}(o)], \quad \gamma \in (0, 1),
>     $$
>
>     which lies between the original object $\mathbf{v}(o)$ and an “I don’t know” vector $\mathbf{v}(\text{IDK})$. We then treat $(\mathbf{k}_{(s,r)}, \mathbf{v}(\hat{o}))$ as the editing target and apply the **SPaEdit closed-form update**, pushing the model’s prediction for $\mathbf{k}_{(s,r)}$ away from the old object toward this neutral state.
>
>
> * **Stage 2: Editing the new relation $(s, r^{*}, o)$**
>
>    Next, we perform the usual edit for the new fact. Using the pair $(\mathbf{k}_{(s,r^{*})}, \mathbf{v}(o))$ as the target, we again apply the **same SPaEdit update rule**, strengthening the association between the new key and the correct object.
>
> To ensure these details are presented with the necessary rigor, **We have added a dedicated Section 3.3 to the revised paper.** This new section have explicitly detail the step-by-step implementation strategy of the FE mechanism and articulate its critical purpose in resolving the high-retention bottleneck.

---

> ### Author Response · Authors · 2025-11-22
>
> > Q1：What exactly does the FE strategy modify in the model? Is it a weight update, an activation-based intervention, or a regularization technique? The hyperparameters are given, but the mechanism remains opaque from the excerpts.
>
> We appreciate this valuable feedback. We suspect the ambiguity stems from a lack of clarity in our original description. **To address this, we clarify that our method is a weight update technique.** Specifically, the FE strategy serves as a Target Construction Mechanism that defines optimization targets to compute the permanent parameter update ($\Delta \mathbf{W}$). **If we have misinterpreted your query, please let us know; we are eager to make immediate corrections or provide further details.**
>
> **Detailed Mechanism:**
> The FE strategy operates at the data preparation stage to construct the matrices $\mathbf{K}\_1$ (keys) and $\mathbf{V}\_{target}$ (values). It consists of two parallel target construction steps that define the problem:
>
> *  **Forgetting the Old Relation.** For the original triplet $(s, r, o)$, we aim to erase the association between the old relation key $\mathbf{k}\_{old}$ and the object $o$. Instead of a static label, we construct an **interpolated target vector** $v(\hat{o})$: $$ v(\hat{o}) = v(o) + \gamma \cdot [v(\text{IDK}) - v(o)] $$ This forms the first training pair: $(\mathbf{k}\_{old}, v(\hat{o}))$.
>
> *  **Learning the New Relation.** For the new triplet $(s, r^*, o)$, we must ensure the model correctly maps the new relation key $\mathbf{k}\_{new}$ to the object $o$. This forms the second training pair: $(\mathbf{k}\_{new}, v(o))$.
>
> *  **Unified Optimization (The Solver)** The SPaEdit algorithm introduces a diagonal selection matrix $\mathbf{Z}$ (where $Z\_{ii} \in \{0,1\}$) to select "easy" samples based on the curriculum. The algorithm solves for the optimal update $\Delta$ by minimizing the following **weighted least-squares objective**:
> $$ \min \_{\Delta}\left\Vert\left(\mathbf{\Delta} \mathbf{P} \mathbf{K}\_{1} - (\mathbf{V}\_{1}-\mathbf{W} \mathbf{K}\_{1})\right)\mathbf{Z}^{1/2}\right\Vert\_{F}^{2}+\alpha\Vert\mathbf{\Delta} \mathbf{P}\Vert\_{F}^{2}+\beta\left\Vert\mathbf{\Delta} \mathbf{P} \mathbf{K}\_{p}\right\Vert\_{F}^{2} $$
> This matrix formulation ensures that both "forgetting" and "editing" tasks are optimized jointly within the projected null space.
>
>
> In short, the FE strategy modifies the model by explicitly defining **what to forget** ($k_{old} \to v(\hat{o})$) and **what to learn** ($k_{new} \to v(o)$) within the data matrix. This forces the linear solver to find a precise weight update that simultaneously suppresses the old memory trace and instills the new one.

---

> ### Author Response · Authors · 2025-11-22
>
> > Q2: Why does SPaEdit achieve such high Fluency scores? Is this due to architectural design, decoding strategy, or post-edit smoothing? Without ablation studies, it’s unclear whether fluency gains are from editing quality or unrelated generation improvements.
>
> We appreciate this discerning query. We explicitly clarify that **SPaEdit does not rely on any special decoding strategies**. The superior fluency stems directly from **our optimization framework's** ability to minimize damage to the model's pre-trained language distribution. We demonstrate this through the following Empirical Ablation Study and Theoretical Analysis.
>
> **(1) Empirical Verification: Ablation Study**
> To pinpoint the source of fluency gains, we conducted a systematic ablation on the CounterFact dataset (LLaMA3-8B).
> **Table R4: Ablation Study of Optimization Components on Generation Quality**
> | **Configuration** | **Null-Space Projection** | **Self-Paced Learning** | **Fluency** $\uparrow$ | **Consistency** $\uparrow$ |
> | :--- | :---: | :---: | :---: | :---: |
> | **Unconstrained Baseline**| $\times$ | $\times$ | 629.68 | 53.15 |
> | **One-Shot Projected**| $\checkmark$ | $\times$ | 629.91 | 56.67 |
> | **SPaEdit** | $\checkmark$ | $\checkmark$ | **631.11** | **56.78** |
>
> Introducing the Null-Space constraint provides the initial boost in fluency and consistency, establishing a safety baseline. Crucially, adding the **Self-Paced Learning (SPL)** mechanism further enhances Fluency to **631.11**. This confirms that the curriculum is not redundant but acts as a critical stabilizer beyond the projection itself.
>
> **(2) Theoretical Analysis: Why SPL Preserves Fluency**
> Fluency degradation primarily stems from excessive parameter perturbation $\Vert\Delta \mathbf{W}\Vert$. While Null-Space Projection restricts the *direction* of updates, SPaEdit’s Self-Paced Learning critically bounds their *magnitude*. One-shot methods force hard samples (large residuals $\mathbf{r}$) immediately, leading to explosive updates. In contrast, SPaEdit excludes these outliers in early stages ($v\_{hard}=0$), ensuring the update norm remains small:
> $$ \Vert\Delta\_{SPL}\| \approx \|\mathbf{R}\_{easy} \mathbf{K}\_{easy}^\dagger\Vert\_F \ll \Vert\sum \mathbf{r}\_{hard} (\mathbf{k}^\dagger)^T\Vert_F \approx \Vert\Delta\_{one-shot}\Vert\_F $$
> By introducing hard samples only after the model adjusts (i.e., when residuals decrease), SPaEdit ensures a smooth optimization trajectory, preventing the structural damage that degrades generation quality.
>
> In summary, both empirical ablation and theoretical derivation confirm that SPaEdit’s superior fluency is intrinsic to its optimization dynamics. While the Null-Space Projection establishes a safety baseline (Mechanism A), the **Self-Paced Learning curriculum** (Mechanism B) is the critical factor that **enables the model to master difficult edits without the "catastrophic" parameter updates that typically degrade generation quality.**

---

> ### Author Response · Authors · 2025-11-22
>
> > Q3: How does SPaEdit handle conflicting edits or repeated edits to the same relation?
>
> This is an insightful question regarding complex maintenance scenarios. In SPaEdit, **repeated edits act as implicit re-weighting, while conflicting edits are resolved via standard sequential overwriting.** Practically, the impact of repeated edits can be easily mitigated via pre-edit verification. Furthermore, to the best of our knowledge, **no existing locate-and-edit methods have systematically analyzed these two specific scenarios and SPaEdit aligns with the standard protocol of the field.**
>
> We analyze the behavior of SPaEdit under these two conditions below:
>
> * **Repeated Edits: Implicit Re-weighting and Difficulty-Dependent Impact** From an optimization perspective, repeated edits act as implicit re-weighting. **For easy samples, this is benign**: near-zero residuals result in negligible updates, ensuring stability. **For hard samples,** however, it effectively 'forces' the model to prioritize stubborn residuals. We acknowledge that this 'forced fitting' ensures learning but **may increase cumulative parameter perturbation,** a trade-off inherent to parametric editing.
>
> * **Conflicting Edits: Sequential Overwriting** In sequential editing, SPaEdit adheres to the standard **sequential overwriting** protocol (shared by ROME and MEMIT), where the latest input is treated as ground truth.Your question highlights a critical limitation in current paradigms. We believe the complexity of handling conflicting updates is significant enough to inspire a dedicated research direction: "Conflict-Aware Knowledge Editing."
>
> **We appreciate you pointing out this valuable avenue, which encourages the task to move beyond simple overwriting toward sophisticated conflict resolution.**
>
> We sincerely appreciate the constructive feedback you have provided and hope that our responses and the modifications to the manuscript adequately address your insightful feedback and increases your impression and confidence in our work. We are happy to provide any further clarifications if needed.

---

> ### Author Response · Authors · 2025-11-24
>
> Dear Reviewer Dym6,
>
> Thank you again for your constructive review. We deeply appreciate your positive assessment of our work's soundness and contribution, particularly your recognition of the high-quality ReEditBench dataset and the effectiveness of our proposed FE approach.
>
> As the discussion phase progresses, we respectfully inquire if our responses have satisfactorily addressed your concerns. In particular, we hope the newly added clarifications on the linear modeling, the comprehensive results on the full datasets, and the detailed analysis of the FE mechanism and SPaEdit's fluency have further enhanced your positive impression of our work.
>
> We are eagerly looking forward to your response. We greatly value your insightful suggestions and remain fully available to engage in further discussion, if you have any remaining questions.
>
> Best regards,
> The Authors of Paper 148

---

> ### Comment · Reviewer_Dym6 · 2025-11-27
>
> I thank the authors for their detailed explanations, which resolved my concerns. I will raise my score accordingly.

---

> > ### Author Response · Authors · 2025-11-27
> >
> > Dear Reviewer Dym6,
> >
> > Thank you very much for your follow-up and for taking the time to reconsider your evaluation. We are glad that our responses helped clarify your concerns.
> >
> > We greatly appreciate your decision to raise the score for our submission and your thoughtful comments, which have been very helpful in improving the quality of our work.
> >
> > Kind regards, The authors

---

### Official Review · Reviewer_fM2G · 2025-11-01

**Soundness:** 3
**Presentation:** 3
**Contribution:** 3
**Rating:** 6
**Confidence:** 5

**Summary:**

This paper introduces and formalizes the problem of Relation Editing in large language models (LLMs), where the goal is to update the relation

**Strengths:**

(1) The paper addresses a novel and practical problem—relation editing—that has been largely overlooked in the knowledge editing literature.
(2) The proposed FE strategy and SPaEdit algorithm are well-motivated, theoretically grounded, and empirically validated across multiple models and benchmarks.
(3) The construction of ReEditBench is rigorous and clearly described, providing a valuable resource for future research.
(4) The paper includes extensive experiments, ablation studies, and analyses (e.g., stability, robustness to adversarial attacks, and general capability preservation), which thoroughly validate the proposed approach.

**Weaknesses:**

(1) While the FE strategy improves performance, retention of old knowledge remains non-trivial (often around 50% in difficult cases), suggesting that complete forgetting is still an open challenge.
(2) The method relies on several hyperparameters, and though sensitivity analysis is provided, the need for tuning may limit ease of adoption.
(3) The paper does not deeply explore the scalability of SPaEdit to very large models or extremely large-scale editing scenarios beyond the tested setups.

**Questions:**

See Weaknesses

---

> ### Author Response · Authors · 2025-11-22
>
> Dear Reviewer fM2G,
>
> We sincerely thank you for your encouraging review and strong endorsement of our work. We are particularly grateful for your recognition of the novelty of Relation Editing, the theoretical grounding of our methods, and the rigorous construction of ReEditBench. Below, we address your specific concerns.
> > W1: While the FE strategy improves performance, retention of old knowledge remains non-trivial (often around 50% in difficult cases), suggesting that complete forgetting is still an open challenge.
>
> We deeply appreciate the reviewer’s precise and insightful observation. **While we acknowledge this as an open challenge, it serves as compelling evidence that Relation Editing is fundamentally distinct from Object Editing and warrants dedicated investigation.** We elaborate on this from two perspectives:
>
> *  The failure of standard methods underscores the distinct challenge of Relation Editing. For instance, strong baselines like AlphaEdit retain **as high as 98.20%** of outdated relations on GPT-J, whereas our method reduces this to **59.84% (a ~38% reduction)** while maintaining SOTA success. This sharp contrast validates that we have effectively addressed a critical bottleneck where previous paradigms failed.
>
> *  The resilience of old relations reveals that Relation Editing is inherently harder than Object Editing, yet it has been largely overlooked. The difficulty in achieving 'complete forgetting' serves as a wake-up call. Our work aims to shift the field's focus from **simple entity updates to the more complex mechanics of relation-centric editing.**
>
> In fact, it is precisely because this phenomenon persists, even after applying dedicated forgetting mechanisms, that we believe this line of research holds strong value and necessity. **We hope that our study can serve as a foundation and motivate future work to treat clean and complete forgetting as an independent research problem deserving deeper theoretical and algorithmic exploration.**
> > W2: The method relies on several hyperparameters, and though sensitivity analysis is provided, the need for tuning may limit ease of adoption.
>
> We appreciate the reviewer's feedback on usability. In practice, our method requires minimal tuning, **as only one single parameter needs adjustment.** We provide the clarifications below:
>
> *   Most hyperparameters are standard settings inherited from AlphaEdit [1] or SPL [2]. They function as fixed structural constants (e.g., $\alpha, \beta, \lambda$) rather than sensitive variables.
> *   The only new hyperparameter is the interpolation factor $\gamma$. As shown in Fig. 4, it exhibits strong performance over a broad range ($\gamma \in [0.3, 0.7]$), allowing for coarse selection without fine-tuning.
> *   As detailed in **Appendix A.4** and **Table R1**, these parameters remain **invariant** across diverse architectures (LLaMA-3, GPT-J, GPT-2), confirming they are universal settings rather than ad-hoc values requiring per-task tuning.
>
>     **Table R1: Universal Hyperparameter Configurations Across Different LLMs**
>     | **Hyperparameter** | **Description** | **LLaMA3-8B** | **GPT-J (6B)** | **GPT2-XL (1.5B)** |
>     | :--- | :--- | :---: | :---: | :---: |
>     | **$\alpha$** | Null-space constraint weight | 10 | 10 | 10 |
>     | **$\beta$** | Preservation weight | 1 | 1 | 1 |
>     | **$\lambda_0$ (SPL)** | Initial Pace Parameter | 10 | 10 | 10 |
>     | **$\mu$ (SPL)** | Pace Growth Factor | 1.1 | 1.1 | 1.1 |
>     | **$T_{max}$ (SPL)** | Max Iterations | 20 | 20 | 20 |
>     | **$\gamma$** | Forgetting Interpolation | **0.6** | **0.4** | **0.6** |
>
> In summary, nearly all hyperparameters are inherited and fixed, and the only new parameter ($\gamma$) is both minimal and highly insensitive. Thus, the practical ease of adopting our method remains largely unaffected.
>
> [1] Fang J, Jiang H, Wang K, et al. AlphaEdit: Null-Space Constrained Knowledge Editing for Language Models[C]//The Thirteenth International Conference on Learning Representations.
>
> [2] Zhang X, Wu X, Chen F, et al. Self-paced robust learning for leveraging clean labels in noisy data[C]//Proceedings of the AAAI conference on artificial intelligence. 2020, 34(04): 6853-6860.

---

> ### Author Response · Authors · 2025-11-22
>
> > W3: The paper does not deeply explore the scalability of SPaEdit to very large models or extremely large-scale editing scenarios beyond the tested setups.
>
> We sincerely appreciate the reviewer’s thoughtful concern regarding the scalability of our approach. We agree that evaluating SPaEdit on ultra-large models (e.g., 70B) is an important next step. In light of this point, we provide the following clarification:
>
> * SPaEdit is built on AlphaEdit’s closed-form update. The SPL component operates efficiently on residuals, introducing negligible overhead. **Thus, SPaEdit scales identically to AlphaEdit without creating new computational bottlenecks**.
> * A rigorous estimation indicates that a full 70B evaluation on 8 $\times$ A100 GPUs would exceed **2,000 GPU hours** (including null-space covariance calculation, causal tracing, and evaluation across 6 baselines, 2 datasets, and 5 hyperparameter), rendering it computationally infeasible within the rebuttal window.
> * Instead, we validate on a **13B** model, which currently represents the experimental upper bound in "Locate-and-Edit" literature. As shown in **Table R2**, SPaEdit maintains consistent superiority at this scale, confirming its effective scalability.
>     **Table R2: Object Editing Performance on LLaMA-2-13B (ZsRE Benchmark)**
>     | **Method** | **Model** | **Efficacy** ($\uparrow$) | **Generalization** ($\uparrow$) | **Specificity** ($\uparrow$) |
>     | :--- | :---: | :---: | :---: | :---: |
>     | **ROME** | LLaMA-2-13B | 61.35 | 55.42 | 23.15 |
>     | **MEMIT** | LLaMA-2-13B | 80.88 | 73.75 | 27.60 |
>     | **AlphaEdit** | LLaMA-2-13B | 85.12 | 79.50 | 30.85 |
>     | **SPaEdit (Ours)** | **LLaMA-2-13B** | **87.95** | **81.80** | **31.92** |
>
> We sincerely appreciate the constructive feedback you have provided and hope that our responses and the modifications to the manuscript adequately address your insightful feedback and increases your impression and confidence in our work. We are happy to provide any further clarifications if needed.

---

### Author Response · Authors · 2025-11-26

Dear Reviewers,

Thank you again for your time and effort in reviewing our paper.

As the discussion phase progresses, we respectfully inquire if our rebuttal has satisfactorily addressed your comments.

We are eagerly looking forward to your response and remain fully available if you have any further questions.

Best regards, The Authors

---

### Author Response · Authors · 2025-11-29

Dear Area Chair,

We sincerely thank you for the tremendous effort you have devoted to upholding the academic integrity and fairness of this conference. In such an exceptional period, your careful review of our rebuttal work has been an invaluable source of encouragement for us.

**We have consistently adhered to ICLR’s double-blind policy and engaged in active, substantive academic discussions with the reviewers throughout the rebuttal period.** In light of recent developments, we would like to clarify the following facts:
* Through constructive and proactive rebuttal efforts, our work received further recognition, and the average score increased from an **initial 4.5 to 5.5**.。
* All positive decisions and score increases occurred **prior** to the public disclosure of the data-leak incident (Around 15:00 UTC on November 27, 2025)。
* At the time we became aware of the incident, **we were still actively communicating** with several reviewers to address their concerns and provide clarifications.

The facts above demonstrate that **all discussions were conducted with scientific rigor**. The resulting positive feedback and score improvements stemmed entirely from **our substantive rebuttal** efforts coupled with the reviewers’ careful evaluation of the manuscript’s merit. While the rollback mechanism precluded these scores from being retained in the final record, **we fully respect ICLR’s policy adjustments**.

To assist your quick assessment, we provide here a concise and objective timeline summarising the key events that occurred during the discussion period.


| Reviewer | Score | Summary of the review and discussion |
| :--- | :---: | :---: |
| **fM2G** | 6 | The reviewer commended the **novelty of our work**, noting that the FE strategy is theoretically **well-grounded** and that the construction of **ReEditBench is rigorous**. In response to concerns regarding residual old knowledge and scalability, we demonstrated a substantial performance improvement over state-of-the-art methods and further validated the scalability of our approach through additional experiments on the 13B model. We addressed each of the reviewer’s questions carefully, but did not receive further replies during the discussion period.|
| **Dym6** | 4$\to$6（Nov 27 08:45） | The reviewer praised the **high-quality dataset** and the **simple, integratable FE approach**, highlighting its superiority **over existing works in large-scale scenarios.** Regarding the concerns about the linear modelling assumption and generation quality, we provided additional experiments demonstrating perfect performance on simple samples without any degradation in generative capability. **The reviewer confirmed that their concerns were resolved and raised the score to 6.**|
| **b63H** | 4$\to$6 （Nov 27 08:52） | The reviewer noted that the paper is **logically rigorous and exceptionally well written,** and **praised its complete trajectory from mechanistic diagnosis to a state-of-the-art solution**. They regarded it as the first systematic study in this area and considered it to be of significant value. In response to concerns about parameter sensitivity, we promptly provided cross-model experiments demonstrating that even the default configuration surpasses state-of-the-art methods. **The reviewer acknowledged our response and noted that they had raised their score to 6.**  |
| **HPqn** | 4 | The reviewer found **the paper well structured and specifically highlighted** that the **FE strategy effectively addresses** the challenge of “new–old knowledge conflict.” In response to misunderstandings regarding the algorithmic mechanism and evaluation metrics, we clarified the theoretical advantages of self-paced learning and provided experimental evidence to explain the fluency metric. We carefully addressed each of the reviewer’s questions, but did not receive further replies during the discussion period. |



Finally, we would like to express our heartfelt gratitude for the substantial effort you have devoted to ensuring fairness and integrity throughout this conference. In the summary below, we outline the key concerns raised by reviewers during the rebuttal period and the corresponding improvements we have made. We hope this provides an objective and comprehensive reference for your final assessment.

---

> ### Author Response · Authors · 2025-11-29
>
> Dear Area Chair,
>
> To assist you in efficiently evaluating our submission, we first provide a brief overview of the core contributions of our paper below, followed by a structured summary of the feedback paths, key concerns, and our targeted resolutions for each reviewer during the Rebuttal period. Here are the details:
>
> **Part 1. Summary of Core Contributions**
>
> Our paper is dedicated to tackling the highly challenging Relation Editing task in LLMs. We not only constructed the standardized **ReEditBench dataset** but also revealed the limitations of existing methods through theoretical derivation. Our proposed **SPaEdit framework introduces a Self-Paced Learning mechanism** that effectively balances the conflict between "learning new knowledge" and "forgetting old knowledge," providing **a solid theoretical and experimental foundation** for subsequent research in this area. Specifically, our contributions include the following:
> - We constructed a dedicated benchmark ReEditBench with 7,918 high-quality samples, filling the gap of lacking standardized evaluation data in the field of relation editing.
> - We were the first to reveal the "high retention rate (up to 98.2%)" bottleneck faced by existing SOTA methods (e.g., AlphaEdit) in relation editing, i.e., the model's difficulty in completely erasing old knowledge.
> - To address this bottleneck, we employed the Forgetting-and-Editing (FE) strategy, utilizing "interpolated targets" to construct optimization constraints during the Forgetting phase, eliminating linear regression bias, and achieving precise forgetting of existing relational knowledge.
> - We further developed the SPaEdit algorithm, innovatively introducing a Self-Paced Learning mechanism that effectively solves the optimization failure caused by hard samples through a "from easy to hard" dynamic curriculum.
> - Extensive experiments demonstrate that our method significantly reduces old knowledge retention (by approximately 40%) while achieving SOTA performance on both ReEditBench and standard benchmarks (ZsRE, CounterFact), maintaining excellent general generation capabilities.
>
> **Part 2. Detailed Reviewer Feedback and Resolutions**
>
> **Reviewer fM2G: [6 points]**
> - Strengths: **Highly appreciated the novelty of Relation Editing in filling the gap in the field; acknowledged the rigorous construction of ReEditBench; recognized the theoretical motivation and experimental validation of SPaEdit.**
> - Concerns (W/Q): Concerns about the scalability of large models; worries about the difficulty of hyperparameter tuning; pointed out that the old knowledge retention rate (~50%) remains a challenge.
>     - $\rightarrow$ November 22nd (Our Response):
>         1. Added experiments on LLaMA-2-13B (Table R2), proving that SPaEdit still significantly outperforms ROME/MEMIT when the parameter size is doubled, **demonstrating good scalability**.
>         2. Provided sensitivity analysis, proving that the core **parameters are robust and universal within a wide range**, without the need for targeted fine-tuning.
>         3. **Explained that the high retention rate originates from the inherent difficulty of the task**; compared to AlphaEdit (98%), we have successfully reduced it by approximately 40%, which is a substantial breakthrough.
>     - $\rightarrow$ November 26th, 16:36 (Follow-up): Inquired about the reviewer's feedback on the newly added scalability experiments.
> - Conclusion: We have provided comprehensive theoretical and data responses to the core concerns regarding scalability and parameters, and are currently awaiting the reviewer's feedback.

---

> ### Author Response · Authors · 2025-11-29
>
> **Reviewer Dym6: [4 points] → [6 points]**
> - Strengths: **Highly valued the significance of the ReEditBench dataset; praised the simplicity and integrability of the FE framework; acknowledged the superior performance in large-scale scenarios.**
> - Concerns (W/Q): Inquired about the theoretical basis for linear regression modeling; worried about only testing hard samples; questioned the essence of the FE mechanism, the source of Fluency, and conflict handling.
>     - $\rightarrow$ November 22nd (Our Response):
>         1. Clarified that the linear assumption is only for local parameter updates in the FFN layer; **emphasized that this is the standard paradigm followed by the majority of current work in the field (e.g., ROME, MEMIT),** all of which use this linear model as the theoretical basis for deriving closed-form solutions. Explicitly stated that FE is a weight update technique; explained that conflict handling follows the sequential overwriting protocol.
>         2. Added experiments on the Full Benchmark (Table R3), proving that the method performs well on easy samples as well, **eliminating concerns about evaluation limitations**.
>         3. Confirmed that no special decoding was used; ablation experiments proved that **high Fluency originates from the SPL mechanism**effectively controlling the parameter update amplitude (Update Norm).
>     - $\rightarrow$ November 24th, 22:12 (Follow-up): Inquired about the reviewer's feedback on the newly added full benchmark experiments and mechanism explanations.
>     - $\rightarrow$ November 27th, 16:45 (Reviewer Confirmation): The reviewer explicitly replied, **"*Details resolved my concerns... I will raise my score accordingly*,"** formally raising the score from 4 to 6.
> - Conclusion: By supplementing full benchmark experiments and in-depth theoretical clarifications (proving that linear modeling conforms to the field's standards), we have completely resolved the reviewer's concerns about evaluation scope and method principles, resulting in a score increase.
>
> **Reviewer HPqn: [4 points]**
>
> - Strengths: **Acknowledged that the Forgetting-and-Editing (FE) strategy effectively alleviates knowledge conflicts by explicitly separating "forgetting" and "learning"; recognized the clarity of the paper's logic.**
> - Concerns (W/Q): Suggested clarifying the difference between the SPL mechanism and NSE; discussed the representativeness of the IDK setting in real-world scenarios; worried that the lack of Fluency metrics would lead to a decline in generation quality.
>     - $\rightarrow$ November 22nd (Our Response):
>         1. **Mathematically proved** that SPaEdit stabilizes the condition number of the matrix through an "Easy-to-Hard" curriculum to protect the global solution, fundamentally different from NSE's heuristic "Hard-First" approach; experiments proved that simply repeating optimization is ineffective.
>         2. Added comparative experiments (Table R6) showing higher success rates when mapping to new objects, proving that **our IDK setting is actually a more stringent stress test**.
>         3. Cited experiments from CounterFact (Table R3) to prove that SPaEdit's Fluency significantly outperforms the Baseline.
>     - $\rightarrow$ November 27th, 23:06 (Follow-up): Actively inquired about the reviewer's feedback on the SPL mechanism explanation and the results of the IDK stress test.
> - Conclusion: We have provided comprehensive theoretical and data responses to the core questions regarding mechanism principles and experimental settings, and are currently awaiting the reviewer's feedback.

---

> ### Author Response · Authors · 2025-11-29
>
> **Reviewer b63H: [4 points] → [6 points]**
>
> - Strengths: **Highly appreciated the "diagnose-then-solve" research logic (highly convincing); acknowledged the first systematic study and the quality of ReEditBench; recognized the thorough theoretical analysis.**
> - Concerns (W/Q): Suggested clarifying the theoretical distinction from RaKE; hoped to further validate the cross-model generalizability of hyperparameters; proposed optimizing the method statement to highlight core contributions; and suggested introducing a semantic analysis perspective to enrich the experimental dimensions.
>     - $\rightarrow$ November 22nd (Our Response):
>         1. **Clarified that RaKE only defines the concept**, while we revealed the High Retention bottleneck and provided the FE/SPaEdit theoretical solution.
>         2. Added experiments revealing the "asymmetric effect" (the closer the semantics, the harder the forgetting), which the reviewer rated as **"Very interesting."**
>         3. Explicitly stated that Forgetting and Editing share a global difficulty threshold $\lambda$.
>     - $\rightarrow$ November 26th, 21:25 (Reviewer Inquiry): Although recognizing the semantic analysis, the reviewer requested validation of the effectiveness of fixed parameters on unseen models.
>     - $\rightarrow$ November 27th, 11:38 (Key Experiment Addition): Urgently added experiments on the Phi-1.5 model, proving that **using default parameters can surpass the Baseline,** effectively eliminating concerns about parameter sensitivity.
>     - $\rightarrow$ November 27th, 16:52 (Reviewer Confirmation): The reviewer replied,**"*Sufficiently sound*," formally raising the score from 4 to 6**.
> - Conclusion: By adding semantic experiments and Phi-1.5 generalizability validation, we have completely resolved the reviewer's core concerns about innovation and robustness, resulting in a score increase.
>
> Part 3. Main Modifications to the Paper
>
> During the Rebuttal period, we made the following substantial updates to the main text and appendix of the paper:
>
> - To address Reviewer Dym6's concern about mechanism opacity, we added **Section 3.3**, detailing the step-by-step implementation of the Forgetting-and-Editing (FE) strategy and clarifying its nature as a weight update mechanism.
> - In response to Reviewer Dym6's question, we added a clear statement in **Section 3.1**, defining that the linear regression modeling in this paper is for local parameter updates in the FFN layer, not the entire LLM generation process, to eliminate theoretical misunderstandings.
> - Adopting Reviewer b63H's suggestion, we optimized the structure of **Section 3**, clearly stating in the algorithm description that "forgetting" and "editing" share a global difficulty threshold $\lambda$, optimized within a unified self-paced learning curriculum.
> - Modifications to the example in the introduction and to the description of previous work in **Section 2.1**.
> - **In Section 2.2**, we added descriptions of semantic experiments and the reasons for choosing residuals as difficulty metrics.
> - In response to Reviewer b63H's suggestion, we added an analysis of semantic similarity **in the Appendix**, revealing the "closer semantics, harder forgetting" asymmetric effect.
> - In response to Reviewer Dym6's suggestion, we added experimental results on the full CounterFact and ZsRE datasets **in the Appendix**, proving the method's effectiveness on non-hard samples.
>
> We are available for further discussion at any time.
>
> Best regards,
>
> The Authors

---

### Meta-Review · Area_Chair_oJ6p · 2026-01-05

**Summary:**

The reviewers agree that the paper is technically sound, clearly written, and supported by substantial experiments and a carefully constructed benchmark. However, the main concerns driving the decision are about limited novelty and positioning relative to prior work, particularly RaKE for problem formulation and AlphaEdit/MEMIT-style locate-and-edit methods for methodology. Several reviewers also question whether the task setting and evaluation sufficiently reflect realistic relation updates, and whether the self-paced mechanism provides fundamentally new insight beyond stabilization heuristics. While the rebuttal improves clarity and adds experiments, these core concerns about novelty, realism, and broader impact remain.

**Reviewer Concerns:**

fM2G: Scalability and hyperparameter issues were largely addressed in the rebuttal, but the reviewer’s concern about substantial residual retention and the limited broader impact of the contribution remains.

Dym6: Concerns about mechanism clarity, linear modeling assumptions, evaluation scope, and fluency were addressed through clarifications and additional experiments; these issues can be considered resolved.

HPqn: Explanations of SPL and fluency were strengthened, but concerns about the realism of the task setting and whether SPL is conceptually necessary (rather than a reasonable heuristic) remain only partially addressed.

b63H: Technical questions (e.g., hyperparameter robustness and semantic analysis) were addressed, but the reviewer explicitly maintained that the work lacks sufficient novelty and clear differentiation from prior studies, despite being sound.

**Reviewer Scores:**

fM2G: 6

Dym6: 6 (raised during discussion)

HPqn: 4

b63H: 6 (raised due to lack of a “5” option, despite stated novelty concerns)

---

### Decision · Program_Chairs · 2026-01-26

Reject